# Reduced progranulin increases tau and α-synuclein inclusions and alters mouse tauopathy phenotypes via glucocerebrosidase

Hideyuki Takahashi [1], Sanaea Bhagwagar [1,2], Sarah H. Nies [1,3], Hongping Ye[4], Xianlin Han [4,5], Marius T. Chiasseu[1], Guilin Wang[6], Ian R. Mackenzie[7] & Stephen M. Strittmatter [1] ✉

Comorbid proteinopathies are observed in many neurodegenerative disorders including Alzheimer's disease (AD), increase with age, and influence clinical outcomes, yet the mechanisms remain ill-defined. Here, we show that reduction of progranulin (PGRN), a lysosomal protein associated with TDP-43 proteinopathy, also increases tau inclusions, causes concomitant accumulation of α-synuclein and worsens mortality and disinhibited behaviors in tauopathy mice. The increased inclusions paradoxically protect against spatial memory deficit and hippocampal neurodegeneration. PGRN reduction in male tauopathy attenuates activity of β-glucocerebrosidase (GCase), a protein previously associated with synucleinopathy, while increasing glucosylceramide (GlcCer)-positive tau inclusions. In neuronal culture, GCase inhibition enhances tau aggregation induced by AD-tau. Furthermore, purified GlcCer directly promotes tau aggregation in vitro. Neurofibrillary tangles in human tauopathies are also GlcCer-immunoreactive. Thus, in addition to TDP-43, PGRN regulates tau- and synucleinopathies via GCase and GlcCer. A lysosomal PGRN−GCase pathway may be a common therapeutic target for age-related comorbid proteinopathies.

Many neurodegenerative disorders are characterized by abnormal accumulation of protein aggregates in the brain. Alzheimer's disease (AD) and frontotemporal lobar degeneration (FTLD) are characterized by tau and/or TAR DNA-binding protein 43 (TDP-43) inclusions while α-synuclein (α-syn) accumulates in the brain of Parkinson's disease (PD) and dementia with Lewy bodies (DLB) patients[1]. In addition to these primary proteinopathies defining the diseases, it is also known that additional proteinopathies often accumulate as co-pathologies in many neurodegenerative diseases. For example, accumulation of α-syn is frequently seen in AD cases. These concomitant proteinopathies increase with age and affect clinical course, therefore have implications for clinical trials targeting only a single disease-associated protein. Global decline in machinery to maintain cellular proteostasis, cross-seeding, and genetic factors are hypothesized to account for the

[1]Cellular Neuroscience, Neurodegeneration, Repair, Departments of Neurology and of Neuroscience, Yale University School of Medicine, New Haven, CT 06536, USA. [2]College of Arts and Sciences, University of Pennsylvania, Philadelphia, PA, USA. [3]Graduate School of Cellular and Molecular Neuroscience, University of Tübingen, D-72074 Tübingen, Germany. [4]Barshop Institute for Longevity and Aging Studies, University of Texas Health Science Center At San Antonio, San Antonio, TX 78229, USA. [5]Department of Medicine, University of Texas Health Science Center At San Antonio, San Antonio, TX 78229, USA. [6]Department of Molecular Biophysics and Biochemistry, School of Medicine, Yale University, New Haven, CT 06520, USA. [7]Department of Pathology, University of British Columbia and Vancouver General Hospital, Vancouver, BC, Canada. ✉e-mail: stephen.strittmatter@yale.edu

comorbidities, but the exact molecular mechanisms are not fully understood[2–4].

Progranulin (PGRN), encoded by the *GRN* gene in humans, is a secreted and lysosomal glycoprotein that plays an important role in lysosomal homeostasis. PGRN is mainly produced by neurons and microglia in the CNS and is involved in several neurodegenerative disorders associated with lysosomal dysfunction[5–7]. While heterozygous *GRN* mutations are a frequent cause of familial FTLD[8,9], rare homozygous *GRN* mutations cause the lysosomal storage disorder neuronal ceroid lipofuscinosis[10]. Several *GRN* polymorphisms are associated with increased risk for Gaucher disease, a common lysosomal storage disorder[11].

Although PGRN was initially linked to FTLD with TDP-43 inclusions, *GRN* mutations are also reported in a substantial number of AD and PD patients[12–19]. In addition, genetic studies have suggested that common *GRN* variants increase risk for AD and PD[20–25]. A *GRN* AD risk variant is associated with increased cerebrospinal fluid tau levels[26]. Accumulation of tau and/or α-syn in addition to TDP-43 inclusions is observed in several FTLD patients with different *GRN* mutations[16,27–32]. In preclinical models, we and others have found an increase in tau pathology in *Grn*[−/−] mice injected with AAV-human P301L tau and P301L tau transgenic mice with PGRN reduction[26,33]. These studies suggest that lysosomal PGRN regulates not only TDP-43 but also other proteinopathies, especially tau- and synucleinopathy, and therefore may be a common therapeutic target for multiple proteinopathies in neurodegenerative diseases. However, the mechanisms by which PGRN regulates other proteinopathies and whether PGRN reduction affects their symptoms are currently not clear.

In this study, to gain insights into the relationship between PGRN and tauopathy, we analyze the PS19 tauopathy mouse model overexpressing human P301S 1N4R tau[34] on PGRN haploinsufficient and complete null backgrounds[35]. PS19 mice develop tau hyperphosphorylation, neurofibrillary tangle-like inclusions, gliosis, neuronal loss, and brain atrophy especially in the hippocampus, amygdala, piriform cortex, and entorhinal cortex[34,36–39]. In addition, PS19 mice display behavioral abnormalities, including cognitive impairments and motor deficits[34,37,40,41]. Therefore, these mice are useful in examining tau pathology and tau-mediated neurodegeneration although interanimal variability and sex differences are also reported[36,38,42–44]. Here, we show that both complete loss and haploinsufficiency of PGRN increase tau inclusions, cause concomitant accumulation of α-syn, and worsen mortality and disinhibited behaviors in PS19 mice. Reduction of PGRN paradoxically protects against a spatial memory deficit and hippocampal neurodegeneration and transcriptomic change in PS19 mice. We find that PGRN reduction with tauopathy significantly impairs activity of β-glucocerebrosidase (GCase), a protein previously associated with synucleinopathy[45,46], while increasing tau inclusions that are immunoreactive for GCase substrate glucosylceramide (GlcCer). In neuronal culture, GCase inhibition promotes tau aggregation induced by AD brain-derived tau fibrils. In vitro studies show that purified GlcCer directly accelerates tau aggregation. Neurofibrillary tangles in human tauopathy brains are also immunoreactive for GlcCer. Thus, our study reveals unexpected role of GlcCer in tau aggregation and demonstrates that PGRN regulates formation of tau and α-syn inclusion via GCase to alter symptoms and neurodegeneration. A lysosomal PGRN–GCase pathway may have therapeutic potential in comorbid proteinopathies.

## Results

### PGRN reduction and tauopathy synergistically decrease body weight and survival rate and worsen disinhibited behavior

To assess effects of PGRN reduction on tauopathy, we crossed PS19 and *Grn*[−/−] mice and generated littermates of 6 genotypes (WT, *Grn*[+/−], *Grn*[−/−], PS19, PS19 *Grn*[+/−], and PS19 *Grn*[−/−] mice). We first monitored their body weights and found a significant decrease in body weight of PS19

*Grn*[−/−] mice, but not the other genotypes, compared to WT mice at 7.5 months of age (Fig. 1a). A significant decrease in the body weight was observed in PS19 mice at 9 months of age. Death due to hindlimb paresis observed in PS19 mice[34,37] was also significantly increased in PS19 *Grn*[−/−] mice compared to WT mice at 10 months of age (Fig. 1b).

Open field tests with the remaining cohort at 10–11 months of age showed significant increases in permanence time and travelled distance in center area of the field for PS19 *Grn*[−/−] mice, but not the other genotypes, compared to WT mice, while total distance travelled was similar between the 6 genotypes, suggesting that PS19 *Grn*[−/−] mice exhibit disinhibited behavior (Fig. 1c). In line with the results of open field test, in elevated plus maze (EPM), PS19 *Grn*[−/−] mice, but not the other genotypes, spent more time in the open arms, compared to WT mice. Consistent with previous studies[40,47], PS19 and *Grn*[−/−] mice also entered the open arms more frequently than WT mice, but the frequency was further increased in PS19 *Grn*[−/−] mice. In addition, a greater travelled distance in the open arms was observed for PS19 *Grn*[+/−] mice (Fig. 1d). Similar results were also observed when only male animals were analyzed (Supplementary Fig. 1a). These results suggest that PGRN reduction and tauopathy synergistically worsen disinhibited behavior.

### PGRN reduction ameliorates memory impairment and hippocampal neurodegeneration in PS19 mice

To assess spatial learning and memory of the 6 genotypes, we also tested the same cohorts in Morris water maze (MWM). Consistent with previous studies[40,41], PS19 mice showed significant learning and memory impairments in the MWM paradigm (Fig. 1e, f). Unexpectedly, in learning trials, the latency to find the platform in PS19 *Grn*[+/−] or PS19 *Grn*[−/−] mice was indistinguishable from that of WT mice in trial blocks 4 and 6. In a probe trial after the learning trials, PS19 *Grn*[+/−] or PS19 *Grn*[−/−] mice spent significantly more time in the target quadrant than average of all other quadrants, suggesting improved memory retention in these mice (Fig. 1f). The latency was similar between the 6 genotypes in the visible platform trial (Fig. 1g). Similar results in MWM were also observed when only male animals were analyzed (Extended Data Fig. 1b, c). Thus, PGRN reduction improved a memory impairment while exacerbating disinhibited behavior in PS19 mice.

To provide insight into the basis for the behavioral results, we assessed neurodegeneration and pathological changes in the brains of the 6 genotypes. Previous studies reported a significant difference between male and female in neurodegeneration and tau pathology in PS19 mice[36,38,42,43]. Thus, subsequent analyses were performed using only male animals at 9–12 months of age unless otherwise noted. We first analyzed PGRN levels in PS19 mice using immunohistochemistry and found an increase in PGRN immunoreactivity, which was primarily associated with microgliosis, in the brain of PS19 mice. A reduction and loss of PGRN immunoreactivity were confirmed in PS19 *Grn*[+/−] and PS19 *Grn*[−/−] mice, respectively (Supplementary Fig. 2).

We then investigated whether PGRN reduction affects brain atrophy, enlarged posterior lateral ventricle, and hippocampal neurodegeneration in PS19 mice[34,36–39]. Strikingly, PGRN reduction significantly attenuated atrophy of the hippocampus and piriform/entorhinal cortex of PS19 mice in a gene dosage-dependent manner (Fig. 1h, i). A significant increase in the posterior lateral ventricle was observed in PS19 mice compared to *Grn*[+/−] or *Grn*[−/−] mice but the increase was absent in PS19 *Grn*[−/−] mice (Fig. 1h, i). Hippocampal CA1 pyramidal neuronal and dentate gyrus (DG) granule cell layers were thinner in PS19, but not in PS19 *Grn*[+/−], or PS19 *Grn*[−/−] mice compared to WT mice (Supplementary Fig. 1d, e). In contrast, there was no significant difference in cresyl-violet-positive cell density in the ventral thalamus of the 6 genotypes, revealing regional specificity of the brain atrophy and neurodegeneration (Supplementary Fig. 3). These results demonstrate an unexpected protective role for PGRN reduction in tau-mediated brain atrophy and hippocampal neurodegeneration.

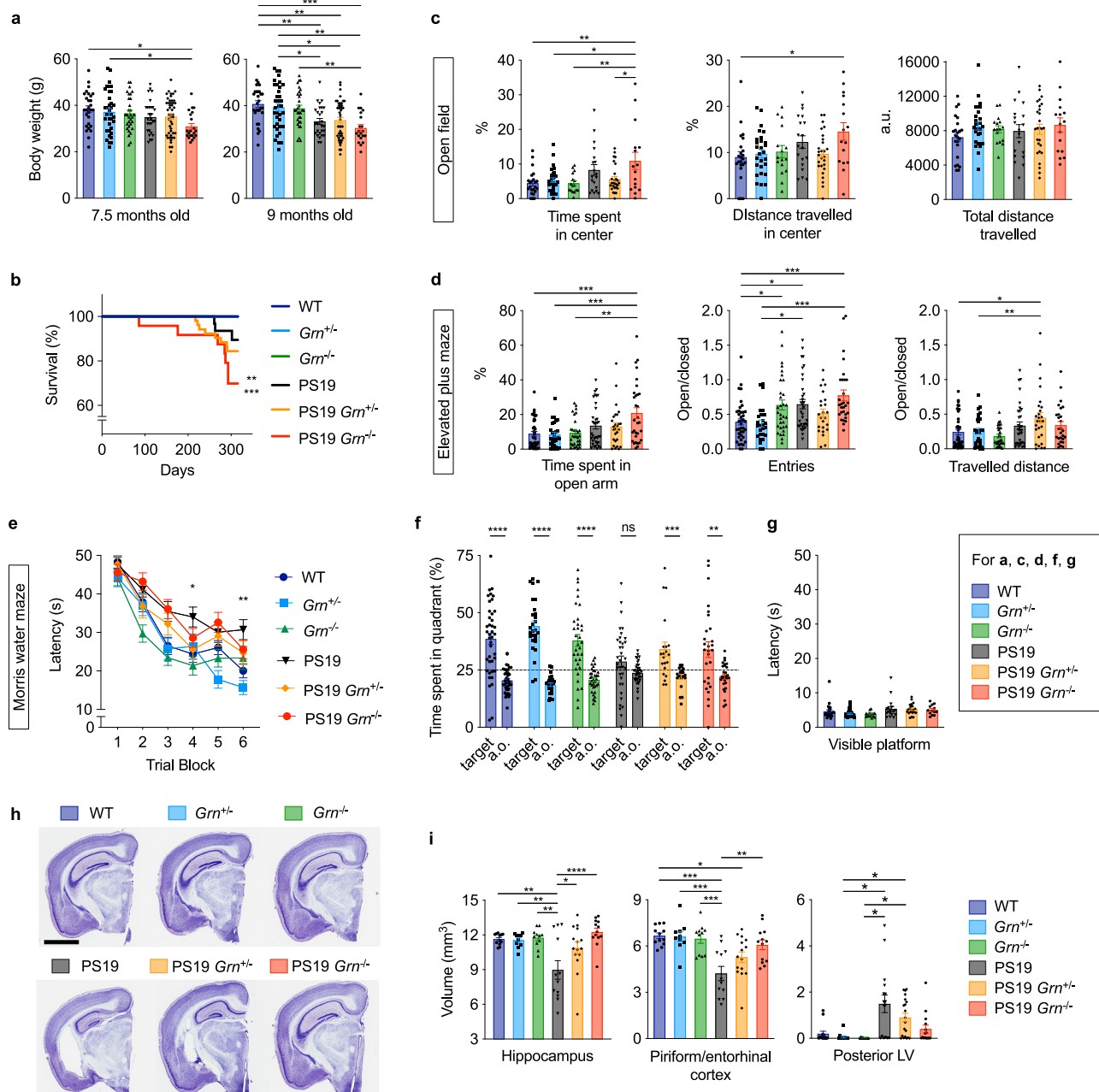

## PGRN reduction attenuates hippocampal transcriptomic changes in PS19 mice

To further characterize the protective role for PGRN reduction in neurodegeneration at transcriptomic levels, we performed single-nucleus RNA sequencing (snRNA-seq) with nuclei isolated from hippocampi of the 6 genotypes with three biological replicates (18 mice, 185,974 nuclei). We used 7-month-old male animals for this analysis to gain insights into earlier changes, rather than late secondary effects due to neurodegeneration. At 7 months of age, no significant hippocampal brain atrophy, neuronal loss, or microgliosis was observed in PS19 mice while accumulation of phospho-tau was detectable (Supplementary Figs. 4, 5). RNA expression profiles of all nuclei were visualized by Uniform Manifold Approximation and Projection (UMAP) and grouped into 13 cell type clusters based on expression of known marker genes (Fig. 2a, b, and Supplementary Fig. 6a). Fractions of each cell type cluster were not significantly different between the 6 genotypes (Fig. 2c), confirming no significant neurodegeneration or gliosis in PS19 mice at this age.

Differential expression analysis in each cluster identified multiple differentially-expressed genes (DEGs) in PS19 nuclei compared to WT, particularly in DG granule cells (Ex_DG), CA1 pyramidal (Ex_CA1), GABAergic (GABA), astrocytes, and oligodendrocytes cell type clusters. In addition, we found significant gene overlaps between DEGs from PS19 versus WT and from PS19 versus PS19 with PGRN reduction for several cell type clusters including excitatory neurons, GABAergic neurons, and oligodendrocytes (Supplementary Fig. 6c). Across cell types, substantially fewer DEGs were detected for non-PS19 mice when comparing $Grn^{+/-}$ and $Grn^{-/-}$ to WT (Supplementary Fig. 6d, e). Overall, the results show that mutant tau causes transcriptomic changes in PS19 neurons and glia, and these are attenuated by PGRN reduction. In fact, a transcriptome-wide rescue effect[48] was also observed in most cell types of PS19 mice with PGRN reduction (Fig. 2d). Please note that a slope of 1 in these correlation plots reflects full rescue of DEGs. Furthermore, enrichment analysis focusing on GABAergic and oligodendrocyte clusters revealed that PGRN reduction significantly decreases enriched terms identified from DEGs in PS19 versus WT

**Fig. 1 | PGRN reduction worsens mortality and disinhibition while improving a memory deficit and hippocampal atrophy in PS19 mice. a** Body weight of 6 genotypes (WT, $Grn^{+/-}$, $Grn^{-/-}$, PS19, PS19 $Grn^{+/-}$, and PS19 $Grn^{-/-}$ mice) at 7.5 and 9 months of age. Mean ± SEM, $n = 28$ mice (7.5-month-old WT), $n = 29$ mice (9-month-old WT), $n = 35$ mice (7.5-month-old $Grn^{+/-}$), $n = 36$ mice (9-month-old $Grn^{+/-}$), $n = 31$ mice ($Grn^{-/-}$), $n = 28$ mice (7.5-month-old PS19), $n = 27$ mice (9-month-old PS19), $n = 47$ mice (7.5-month-old PS19 $Grn^{+/-}$), $n = 46$ mice (9-month-old PS19 $Grn^{+/-}$), $n = 21$ mice (7.5-month-old PS19 $Grn^{-/-}$), $n = 20$ mice (9-month-old PS19 $Grn^{-/-}$), *$p = 0.0134$ (7.5 months old, WT vs. PS19 $Grn^{-/-}$), *$p = 0.0448$ (7.5 months old, $Grn^{+/-}$ vs. PS19 $Grn^{-/-}$), **$p = 0.0081$ (9 months old, WT vs. PS19), **$p = 0.0047$ (9 months old, WT vs. PS19 $Grn^{+/-}$), ***$p = 0.0002$ (9 months old, WT vs. PS19 $Grn^{-/-}$), *$p = 0.0397$ (9 months old, $Grn^{+/-}$ vs. PS19), *$p = 0.0277$ (9 months old, $Grn^{+/-}$ vs. PS19 $Grn^{+/-}$), **$p = 0.0013$ (9 months old, $Grn^{+/-}$ vs. PS19 $Grn^{-/-}$), **$p = 0.0040$ (9 months old, $Grn^{-/-}$ vs. PS19 $Grn^{-/-}$); One-way ANOVA with Tukey's post hoc test. **b** Survival curves of 6 genotypes. Mean ± SEM, $n = 57$ mice (WT), $n = 57$ mice ($Grn^{+/-}$), $n = 30$ mice ($Grn^{-/-}$), $n = 31$ mice (PS19), $n = 53$ mice (PS19 $Grn^{+/-}$), $n = 24$ mice (PS19 $Grn^{-/-}$), **$p = 0.0015$ (WT or $Grn^{-/-}$ vs. PS19 $Grn^{-/-}$), ***$p = 0.0001$ ($Grn^{+/-}$ vs. PS19 $Grn^{-/-}$); Pairwise log-rank test. These were the only $p$-values that were less than 0.0033 of Bonferroni-corrected threshold, which equals 0.05 divided by 15 comparisons. **c** Open field test of 6 genotypes at 10–11 months of age showing time spent in the center zone (%), distance travelled in the center zone (%), and total distance travelled during the test. Mean ± SEM, $n = 23$ mice (WT), $n = 28$ mice ($Grn^{+/-}$), $n = 16$ mice ($Grn^{-/-}$), $n = 19$ mice (PS19), $n = 27$ mice (PS19 $Grn^{+/-}$), $n = 16$ mice (PS19 $Grn^{-/-}$), **$p = 0.0035$ (time spent in center, WT vs. PS19 $Grn^{-/-}$), *$p = 0.0133$ (time spent in center, $Grn^{+/-}$ vs. PS19 $Grn^{-/-}$), **$p = 0.0096$ (time spent in center, $Grn^{-/-}$ vs. PS19 $Grn^{-/-}$), *$p = 0.0242$ (time spent in center, PS19 $Grn^{+/-}$ vs. PS19 $Grn^{-/-}$), *$p = 0.0318$ (distance travelled in center, WT vs. PS19 $Grn^{-/-}$); One-way ANOVA with Tukey's post hoc test. **d** Elevated plus maze (EPM) of 6 genotypes at 10–11 months of age showing time spent in the open arms (%), ratio of number of entries into the open versus closed arms, and ratio of distance travelled in the open versus closed arms of the maze. Mean ± SEM, $n = 41$ mice (WT), $n = 28$ mice ($Grn^{+/-}$), $n = 34$ mice ($Grn^{-/-}$), $n = 37$ mice (PS19), $n = 27$ mice (PS19 $Grn^{+/-}$), $n = 31$ mice (PS19 $Grn^{-/-}$), ***$p = 0.0002$ (time spent in open arms, WT vs. PS19 $Grn^{-/-}$), ***$p = 0.0005$ (time spent in open arms, $Grn^{+/-}$ vs. PS19 $Grn^{-/-}$), **$p = 0.0012$ (time spent in open arms, $Grn^{-/-}$ vs. PS19 $Grn^{-/-}$), *$p = 0.0471$ (entries to open arms, WT vs. $Grn^{-/-}$), *$p = 0.0272$ (entries to open arms, WT vs. PS19), ***$p = 0.0002$ (entries to open arms, WT vs. PS19 $Grn^{-/-}$), *$p = 0.0317$ (entries to open arms, $Grn^{+/-}$ vs. PS19), ***$p = 0.0004$ (entries to open arms, $Grn^{+/-}$ vs. PS19 $Grn^{-/-}$), *$p = 0.0316$ (distance travelled in open arms, WT vs. PS19 $Grn^{+/-}$), **$p = 0.0046$

(distance travelled in open arms, $Grn^{-/-}$ vs. PS19 $Grn^{-/-}$); One-way ANOVA with Tukey's post hoc test. **e** Morris water maze (MWM) learning trial of 6 genotypes at 10–11 months of age. Spatial learning is plotted as latency to find hidden platform. Mean ± SEM, $n = 41$ mice (WT), $n = 28$ mice ($Grn^{+/-}$), $n = 34$ mice ($Grn^{-/-}$), $n = 35$ mice (PS19), $n = 21$ mice (PS19 $Grn^{+/-}$), $n = 28$ mice (PS19 $Grn^{-/-}$). In the trial block 4 and 6, the PS19 group differed from the WT group, whereas none of the other groups differed from the WT group, as shown in figure. *$p = 0.0192$ (WT vs. PS19), *$p = 0.0042$ (WT vs. PS19); One-way ANOVA with Dunnett's post hoc test. **f** MWM probe trial of 6 genotypes at 10–11 months of age. Percentage of time spent in the target quadrant and averaged time spent in all other (a.o.) quadrants. Mean ± SEM, $n = 41$ mice (WT), $n = 28$ mice ($Grn^{+/-}$), $n = 34$ mice ($Grn^{-/-}$), $n = 35$ mice (PS19), $n = 21$ mice (PS19 $Grn^{+/-}$), $n = 28$ mice (PS19 $Grn^{-/-}$), ****$p = 0.000000006$ (WT), ****$p = 0.000000000002$ ($Grn^{+/-}$), ****$p = 0.0000002$ ($Grn^{-/-}$), **$p = 0.0021$, ***$p = 0.0010$; Two-tailed unpaired $t$ test with Welch's correction. Additionally, two-tailed one sample $t$ test showed that the mean time of PS19 mice, but not the others, in the target quadrant is not different from random chance performance of 25% (dashed line) ($p = 0.139$). **g** Visible platform trial of 6 genotypes at 10–11 months of age. Mean ± SEM, $n = 22$ mice (WT), $n = 26$ mice ($Grn^{+/-}$), $n = 16$ mice ($Grn^{-/-}$), $n = 16$ mice (PS19), $n = 21$ mice (PS19 $Grn^{+/-}$), $n = 13$ mice (PS19 $Grn^{-/-}$). **h** Representative images of Nissl staining of sections from 6 genotypes (WT, $Grn^{+/-}$, $Grn^{-/-}$, PS19, PS19 $Grn^{+/-}$, and PS19 $Grn^{-/-}$ mice). Bar, 2 mm. **i** Volumes of the hippocampus, piriform/entorhinal cortex, and posterior lateral ventricle (LV) in 6 genotypes at 9–12 months of age. Mean ± SEM, $n = 12$ mice (WT), $n = 10$ mice ($Grn^{+/-}$), $n = 11$ mice ($Grn^{-/-}$), $n = 12$ mice (PS19), $n = 15$ mice (PS19 $Grn^{+/-}$), $n = 13$ mice (PS19 $Grn^{-/-}$) for the hippocampus and piriform/entorhinal cortex, and $n = 14$ mice (WT), $n = 12$ mice ($Grn^{+/-}$), $n = 13$ mice ($Grn^{-/-}$), $n = 16$ mice (PS19), $n = 20$ mice (PS19 $Grn^{+/-}$), $n = 15$ mice (PS19 $Grn^{-/-}$) for the posterior LV, **$p = 0.0018$ (hippocampus, WT vs. PS19), **$p = 0.0050$ (hippocampus, $Grn^{+/-}$ vs. PS19), **$p = 0.0013$ (hippocampus, $Grn^{-/-}$ vs. PS19), *$p = 0.0382$ (hippocampus, PS19 vs. PS19 $Grn^{+/-}$), ****$p = 0.00005$ (hippocampus, PS19 vs. PS19 $Grn^{-/-}$), *$p = 0.0338$ (piriform/entorhinal cortex, WT vs. PS19 $Grn^{+/-}$), ***$p = 0.0001$ (piriform/entorhinal cortex, $Grn^{+/-}$ vs. PS19), ***$p = 0.0002$ (piriform/entorhinal cortex, $Grn^{-/-}$ vs. PS19), **$p = 0.0023$ (piriform/entorhinal cortex, PS19 vs. PS19 $Grn^{-/-}$), *$p = 0.0249$ (posterior LV, $Grn^{+/-}$ vs. PS19), *$p = 0.0238$ (posterior LV, $Grn^{+/-}$ vs. PS19 $Grn^{+/-}$), *$p = 0.0240$ (posterior LV, $Grn^{-/-}$ vs. PS19), *$p = 0.0226$ (posterior LV, $Grn^{-/-}$ vs. PS19 $Grn^{+/-}$); One-way ANOVA with Tukey's post hoc test or Kruskal-Wallist test with Dunn's post hoc test (for the posterior LV). Source data are provided as a Source Data file.

(Fig. 2e). Recent single cell RNA-seq analyses have identified disease-associated microglia (DAM) and astrocytes (DAA) signatures in mouse models of AD[49,50]. Upregulated genes in PS19 microglia and astrocytes partly overlapped with those upregulated in DAM and DAA (Fig. 2f, g). Importantly, PGRN reduction also decreased expression of these genes (Fig. 2f, g). Together, PGRN reduction attenuated neuronal and glial transcriptomic changes in the hippocampus of PS19 mice.

## PGRN reduction alters tau pathology pattern and increases tau inclusions

The rescue of brain atrophy and global gene expression in PS19 $Grn^{+/-}$ and PS19 $Grn^{-/-}$ hippocampi suggests an effect of PGRN reduction on tau pathology. Therefore, we next examined tau pathology of PS19 mice with PGRN reduction by immunostaining using AT8 (p-S202/T205 tau) antibody. Previous studies have established four types of AT8 staining patterns (type1-4) in PS19 hippocampus that are correlated with hippocampal atrophy[36,39,51]. We reproduced this correlation in our cohort, irrespective of $Grn$ genotypes (Fig. 3a, b). Intriguingly, while PGRN reduction decreases the percentage of type 4 in PS19 mice, there was an increase in type 2, which is characterized by tangle-like cell body staining, in PS19 $Grn^{+/-}$ and PS19 $Grn^{-/-}$ mice (Fig. 3c). Consequently, we observed a significant decrease in AT8 mean intensity and AT8-positive area in the hippocampus of PS19 $Grn^{-/-}$ mice (Fig. 3e, f). The decrease in AT8 mean intensity and AT8-positive area was also found in the piriform/entorhinal cortex, while tangle-like inclusions were frequently observed (Fig. 3d–f). Indeed, high-magnification confocal image analysis revealed an increase in AT8-positive tau inclusions in the CA2 + CA3 and amygdala regions of PS19 $Grn^{+/-}$ and

PS19 $Grn^{-/-}$ mice, although the increase did not reach statistical significance in PS19 $Grn^{+/-}$ mice (Fig. 3g, h). Immunoblot analysis using anti-total tau antibody of the formic-acid-soluble fraction showed that the increase in tau inclusions was due to a shift in subcellular distribution without an overall change of insoluble total tau amount in PS19 mice with PGRN reduction (Supplementary Fig. 7e). These results suggest that PGRN reduction decreases overall p-tau area while accelerating formation of concentrated tangle-like cell body tau inclusions in PS19 brains, which is partly consistent with previous studies using different tauopathy models[26,33].

## Tau phosphorylation and gliosis are not associated with increased tau inclusions in PS19 mice with PGRN reduction

We next sought to elucidate the mechanisms underlying increased tau inclusions in PS19 mice with PGRN reduction. We first examined the status of tau phosphorylation in the cortex of these animals. Immunoblot analysis using anti-total tau and phospho-tau antibodies (p-S199/S202, p-S202/T205, p-S356, and p-S396/S404) of the RAB- and RIPA-soluble fractions revealed no significant changes in total tau and phospho-tau levels in the cortex of PS19 mice with PGRN reduction (Supplementary Fig. 7a–d).

Several recent studies have suggested an important role of microglial activation/neuroinflammation in driving tau pathology[39,52–54]. PGRN has also been implicated in microglial activation[5,6,55]. Therefore, we next examined whether microgliosis is associated with increased tau inclusions in PS19 mice with PGRN reduction using microglial markers Iba1 and CD68. Immunohistochemistry showed an increase in Iba1-immunoreactivity in PS19

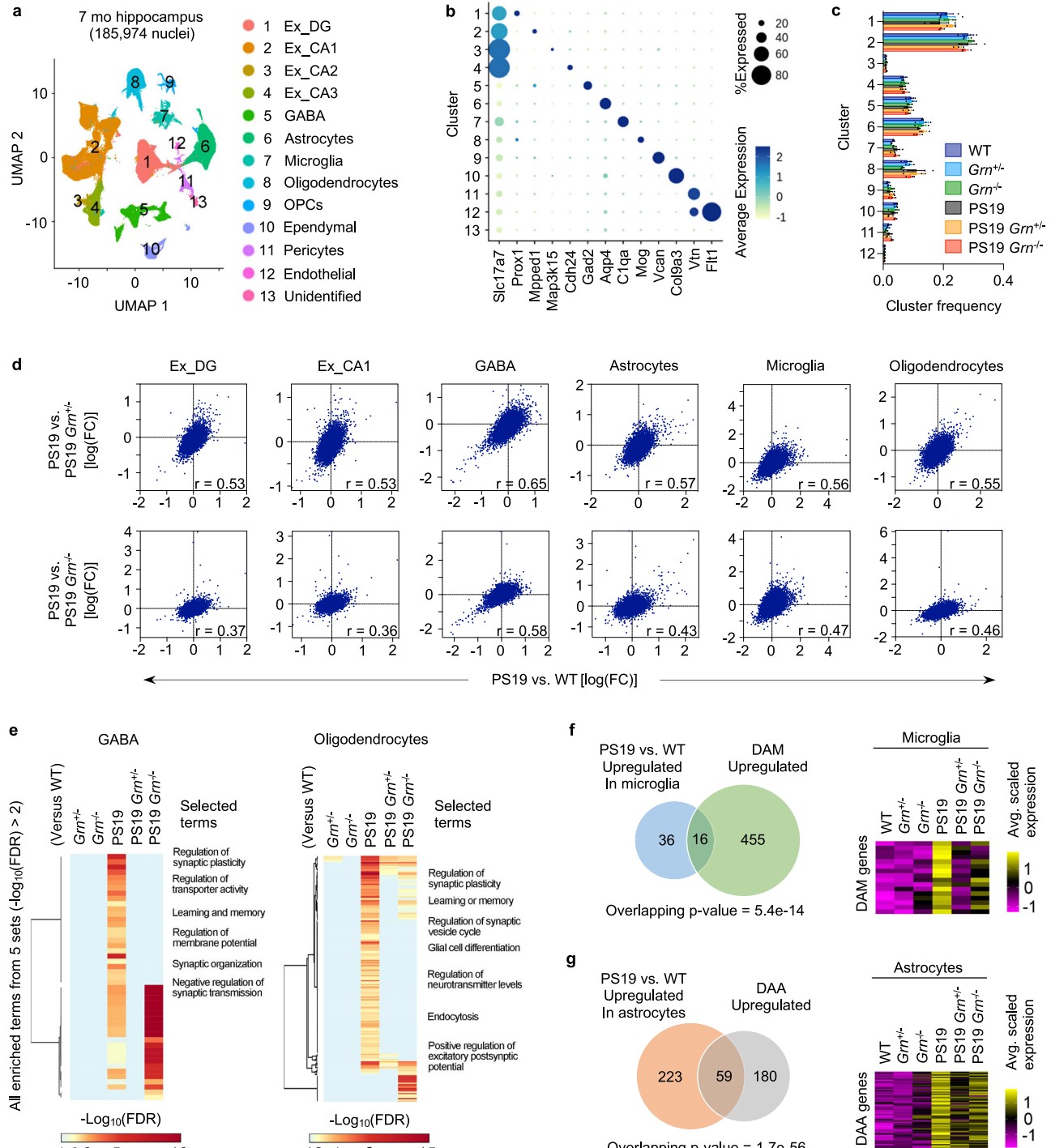

**Fig. 2 | PGRN reduction attenuates transcriptomic changes in PS19 mice.**
**a** UMAP plot showing 13 cell type clusters of 185,974 single-nucleus RNA profiles from hippocampi of 6 genotypes (WT, *Grn*⁺/⁻, *Grn*⁻/⁻, PS19, PS19 *Grn*⁺/⁻, and PS19 *Grn*⁻/⁻ mice) with three biological replicates at 7 months of age. **b** Dot plot showing the percent of cells expressing marker genes and average expression across all cell type clusters. **c** Quantification of frequency of each cluster in 6 genotypes. Mean ± SEM, *n* = 3 mice per genotype. No significant difference was observed between 6 genotypes in each cluster (One-way ANOVA). **d** Transcriptome-wide rescue/exacerbation plots showing the relationship between log(FC)s of genes in PS19 vs. WT and PS19 vs. PS19 *Grn*⁺/⁻ or PS19 *Grn*⁻/⁻ mice in the indicated cell clusters. Genes that express >1% of all nuclei in either genotype were used. The correlation coefficients (r) are indicated in the plots. All correlations were significant (*p* (two-tailed) <1e-15), although adjustments were not made for multiple comparisons.

**e** Heatmaps showing values of −log10(FDR) of all enriched terms identified from 5 comparisons (*Grn*⁺/⁻, *Grn*⁻/⁻, PS19, PS19 *Grn*⁺/⁻, or PS19 *Grn*⁻/⁻ vs. WT) (defined by −log₁₀(FDR) > 2) in the same 5 comparisons in GABA and Oligodendrocytes cell type clusters. Selected terms are highlighted. **f** Effects of PGRN reduction on tau-induced expression of DAM-associated genes. Venn diagram shows the overlap between DEGs in PS19 microglia (vs. WT) and DAM-associated genes. Heatmap shows expression of the overlapping genes in microglia of 6 genotypes. Overlapping *p*-values was calculated by one-sided Fisher's exact test. **g** Effects of PGRN reduction on tau-induced expression of DAA-associated genes. Venn diagram shows the overlap between DEGs in PS19 astrocytes (vs. WT) and DAA-associated genes. Heatmap shows expression of the overlapping genes in astrocytes of 6 genotypes. Overlapping *p*-values was calculated by one-sided Fisher's exact test.

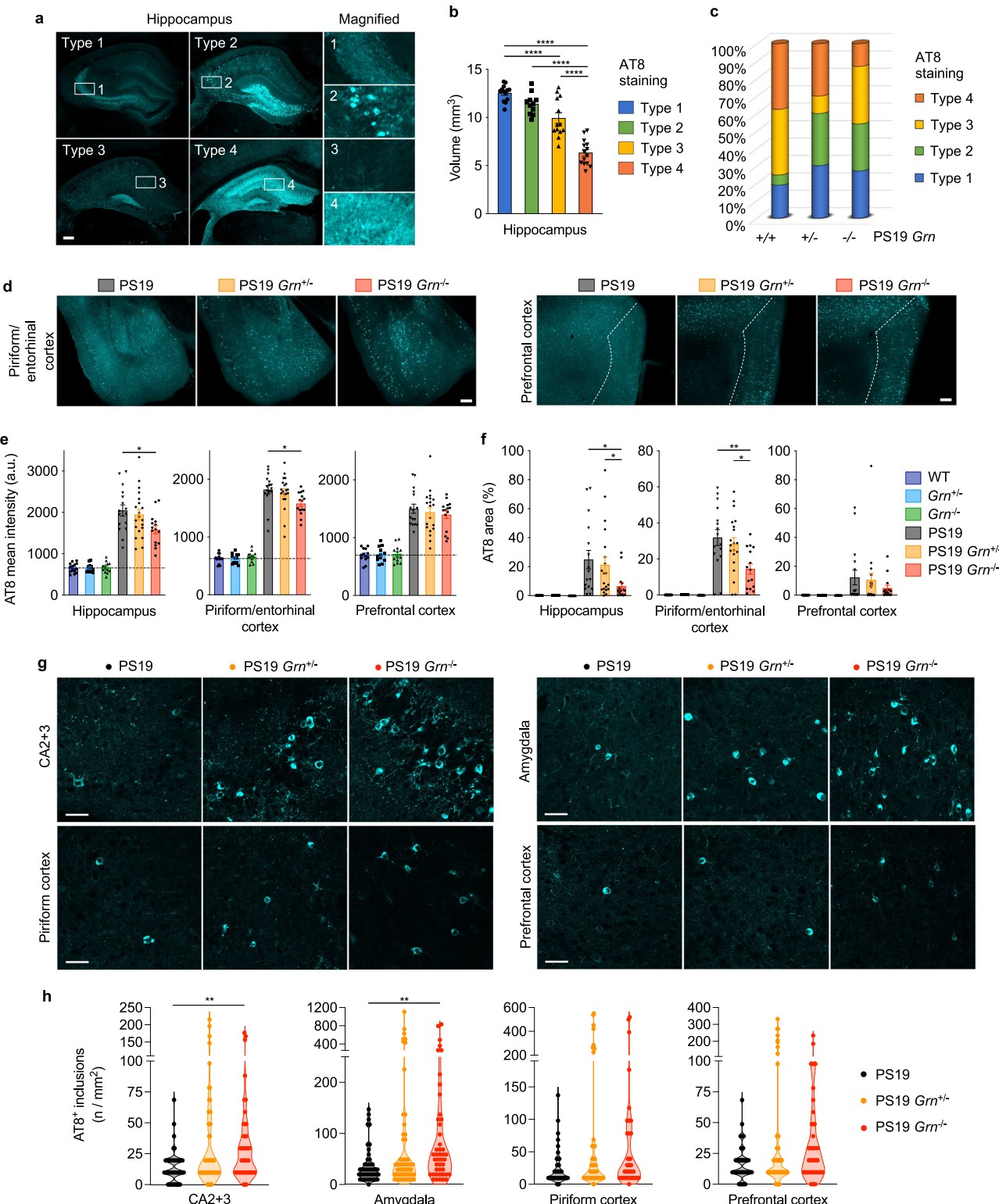

brains but no significant difference between PS19, PS19 *Grn*⁺/⁻ and PS19 *Grn*⁻/⁻ brains (Fig. 4a, b, and Supplementary Fig. 2b–e). Similar to our previous studies using the APP/PS1 cerebral Aβ-amyloidosis mouse model[26], CD68-immunoreactivity within Iba1 area was significantly increased in PS19 *Grn*⁻/⁻ mice. However, the increase was not observed in PS19 *Grn*⁺/⁻ mice (Fig. 4c). Critically, the CD68 levels were not significantly correlated with AT8 staining types in the hippocampus in our cohorts (Fig. 4d). We also examined astrocytosis of the 6 genotypes using GFAP staining. Similar to the Iba1 staining, GFAP

immunoreactivity was increased in PS19 brains, but it was not significantly affected by PGRN reduction (Supplementary Fig. 8). Thus, neither tau phosphorylation nor glial marker changes were correlated with increased tau inclusions in PS19 mice with PGRN reduction.

### PGRN reduction attenuates GCase activity and increases GlcCer-positive tau inclusions in PS19 mice

We then considered whether PGRN reduction cell-autonomously affects tau inclusions in neurons. Co-staining with anti-PGRN and

**Fig. 3 | PGRN reduction alters AT8 staining pattern and increases tau inclusions in PS19 mice. a** Representative images of hippocampal four distinct AT8 staining types. Bar, 200 μm. **b** Association of hippocampal atrophy with four AT8 staining types. Mean ± SEM, $n = 51$ mice, ****$p = 0.0001$ (Type 1 vs. Type 3), ****$p < 1e\text{-}15$ (Type 1 vs. Type 4), ****$p = 0.00000000005$ (Type 2 vs. Type 4), ****$p = 0.0000002$ (Type 3 vs. Type 4); One-way ANOVA with Tukey's post hoc test. **c** Distribution of the AT8 staining types in 3 genotypes (PS19, PS19 Grn+/-, and PS19 Grn-/- mice) at 9–12 months of age. $n = 15$–20 mice per genotype. **d** Representative images of AT8 staining in the piriform/entorhinal and prefrontal cortex of PS19, PS19 $Grn^{+/-}$, and PS19 $Grn^{-/-}$ mice. Bar, 200 μm. **e** AT8 mean intensity of the hippocampus, piriform/entorhinal cortex, and prefrontal cortex of 6 genotypes (WT, $Grn^{+/-}$, $Grn^{-/-}$, PS19, PS19 $Grn^{+/-}$, and PS19 $Grn^{-/-}$ mice) at 9–12 months of age. Mean ± SEM, $n = 14$ mice (WT), $n = 12$ mice ($Grn^{+/-}$), $n = 13$ mice ($Grn^{-/-}$), $n = 16$ mice (PS19), $n = 20$ mice (PS19 $Grn^{+/-}$), $n = 15$ mice (PS19 $Grn^{-/-}$), *$p = 0.0376$ (hippocampus), *$p = 0.0378$ (piriform/entorhinal cortex); One-way ANOVA with Tukey's post hoc test (in PS19, PS19 $Grn^{+/-}$, and PS19 $Grn^{-/-}$ mice). **f** AT8-positive area of the hippocampus, piriform/entorhinal cortex, and prefrontal cortex of 6 genotypes at 9–12 months of age. Mean ± SEM, $n = 14$ mice (WT), $n = 12$ mice ($Grn^{+/-}$), $n = 13$ mice ($Grn^{-/-}$), $n = 16$ mice (PS19), $n = 20$ mice (PS19 $Grn^{+/-}$), $n = 15$ mice (PS19 $Grn^{-/-}$), *$p = 0.0108$ (hippocampus, PS19 vs. PS19 $Grn^{-/-}$), *$p = 0.0404$ (hippocampus, PS19 $Grn^{+/-}$ vs. PS19 $Grn^{-/-}$), **$p = 0.0069$ (piriform/entorhinal cortex, PS19 vs. PS19 $Grn^{-/-}$), *$p = 0.0286$ (piriform/entorhinal cortex, PS19 $Grn^{+/-}$ vs. PS19 $Grn^{-/-}$); One-way ANOVA with Tukey's post hoc test for the piriform/entorhinal cortex and Kruskal-Wallis test with Dunn's post hoc test for the other regions (in PS19, PS19 $Grn^{+/-}$, and PS19 $Grn^{-/-}$ mice). **g** Representative confocal images of AT8 staining in the CA2 + 3, amygdala, piriform cortex, and prefrontal cortex regions of PS19, PS19 $Grn^{+/-}$ or PS19 $Grn^{-/-}$ mice. Bar, 50 μm. **h** Quantification of the number of AT8-positive inclusions in the CA2 + 3, amygdala, piriform cortex, and prefrontal cortex of 3 genotypes (PS19, PS19 $Grn^{+/-}$, and PS19 $Grn^{-/-}$ mice) at 9–12 months of age. Mean ± SEM, $n = 15$–20 mice per mouse, 3 ROIs per mouse, **$p = 0.0043$ (CA2 + 3, PS19 vs. PS19 $Grn^{-/-}$), **$p = 0.0023$ (amygdala, PS19 vs. PS19 $Grn^{-/-}$); Kruskal-Wallis test with Dunn's post hoc test. Source data are provided as a Source Data file.

AT8 antibodies revealed no incorporation of PGRN into tau inclusions in PS19 mice (Supplementary Fig. 9a). Thus, it seems unlikely that PGRN prevents formation of tau inclusions by directly binding to tau. Interestingly, we observed fewer PGRN-positive puncta consistent with lysosomes in neurons with AT8-positive tau inclusions compared to ones without the inclusions (Supplementary Fig. 9a), prompting a consideration of the role of lysosomes in formation of tau inclusions.

Recent studies have reported physical interaction between PGRN and β-glucocerebrosidase (GCase), a lysosomal enzyme that cleaves the glycosphingolipids glucosylceramide (GlcCer) and glucosyl-sphingosine (GlcSph). In addition, PGRN deficiency was shown to result in destabilization and/or reduced activity of GCase[56–58]. Interestingly, mice with reduced GCase activity have been reported to develop tau pathology[59–61]. We confirmed both the PGRN-GCase interaction using co-immunoprecipitation (co-IP) assay with HEK293T cells (Supplementary Fig. 9b, c) and the reduced GCase activity using 5-month-old $Grn^{-/-}$ cortices (Supplementary Fig. 9d). Thus, we next examined whether GCase and GlcCer/Sph are involved in increased formation of tau inclusions upon PGRN reduction. Immunohistochemistry with anti-GlcCer and GlcSph antibodies revealed that tau inclusions in PS19 mice were consistently immunoreactive for GlcCer but not GlcSph (Fig. 5a, e and Supplementary Fig. 9e, f), suggesting a potential involvement of GlcCer in tau inclusion formation. Importantly, specificity of the anti-GlcCer antibody has been previously demonstrated[62,63] and the antibody has been commonly used for immunostaining[64–67]. We also confirmed an increase in GlcCer and GlcSph immunoreactivity in the brains of mice treated with GCase inhibitor conduritol B epoxide (CBE) (Supplementary Fig. 9g–i). GCase activity assay using cortical lysates showed that while there was no difference in GCase activity between WT and PS19 brains at 3 months of age, total GCase activity was slightly but significantly decreased in PS19 brains at 10 months of age (Fig. 5b). In addition, total GCase and GBA1-specific activity of PS19 brains were further attenuated in both PS19 $Grn^{+/-}$ and PS19 $Grn^{-/-}$ brains (Fig. 5c). Consistent with the reduced GCase activity, immunohistochemistry showed gene dosage-dependent increases in GlcCer-positive tau inclusions and integrated GlcCer intensity within tau inclusions in PS19 mice with PGRN reduction (Fig. 5d, e). Nearly all AT8 inclusions were also immunoreactive for GlcCer in PS19, PS19 $Grn^{+/-}$ and PS19 $Grn^{-/-}$ brains, 94.0% (322 out of 343), 98.2% (2006 out of 2043), 99.7% (2015 out of 2020), respectively. The GlcCer-positive tau inclusions in PS19 mice with PGRN reduction were also consistently labeled by MC1, a conformation-dependent antibody specific for PHF-tau[68,69], suggesting pathological conformation of tau in the inclusions (Supplementary Fig. 10).

To determine levels of GlcCer and galactosylceramide (GalCer) species, we further performed SFC-MS/MS analysis using lipid extracts from cortical lysates of 6 genotypes. We found that PGRN reduction increased levels of several GlcCer species and total GlcCer

in WT and PS19 cortices. (Fig. 5f, g and Supplementary Fig. 11). Importantly, PGRN reduction did not cause any significant changes in GalCer species, which are also cerebrosides but are not substrates of GCase (Fig. 5f, g and Supplementary Fig. 11). We also examined levels of other 253 lipid species of 14 general lipid classes as well as bis(-monoacylglycerol)phosphate (BMP) and GM1 ganglioside, both of which have been recently shown to be affected by PGRN deficiency[70,71], with multidimensional mass spectrometry-based shotgun lipidomic analysis (Supplementary Fig. 12a). Even without false discovery rate adjustment, only a few lipid species were significantly increased in $Grn^{-/-}$ or PS19 $Grn^{-/-}$ cortex compared to WT or PS19, respectively, and none of them was increased in both $Grn^{-/-}$ and PS19 $Grn^{-/-}$ cortex (Supplementary Fig. 12b, c). Specifically, we did not observe an increase of GM1 ganglioside species that were detected in a previous study[71] which assessed mice at a more advanced age. Consistent with previous studies[70,71], there was a significant decrease in several species and total levels of BMP in both $Grn^{-/-}$ and PS19 $Grn^{-/-}$ cortices (Supplementary Fig. 12b, c, d and 13a). However, the decrease in BMP was not seen in $Grn^{+/-}$ and PS19 $Grn^{+/-}$ cortex. In addition, immunohistochemical analysis showed no accumulation of BMP in tau inclusions in PS19 mice (Supplementary Fig. 13b), excluding a possibility of a direct involvement of BMP in tau aggregation. We also found that several species of sulfatides were decreased in both $Grn^{-/-}$ and PS19 $Grn^{-/-}$ cortices, although total sulfatide levels were not significantly decreased in $Grn^{-/-}$ and PS19 $Grn^{-/-}$ cortices (Supplementary Fig. 13a). Together, these results suggest that PGRN reduction impairs GCase activity and that an increase in GlcCer accumulation may promote formation of tau inclusions in PS19 mice.

## PGRN deficiency alters subcellular localization of GCase but not saposin C

The decrease in GCase activity in PGRN-deficient cortices was significant but limited in magnitude, so we considered whether PGRN reduction may also affect trafficking of GCase and/or saposin C (Sap C), an activator of GCase, to the lysosome thereby causing an additional functional contribution to the increase in GlcCer levels. To test this hypothesis, we isolated lysosome-enriched fractions from 6-month-old WT and $Grn^{-/-}$ cortices using Optiprep density gradient centrifugation (Supplementary Fig. 14a, b). After the centrifugation, we found two lysosomal enzymes-enriched fractions (fractions #2 and #3) in the cortex of WT mice (Supplementary Fig. 14c, f). Interestingly, PGRN deficiency increases accumulation of the lysosomal markers in fractions #3 and/or #4 (Supplementary Fig. 15c, d, f), and the #3/#2 ratio of the lysosomal markers, but not other organelle markers, was significantly increased in $Grn^{-/-}$ cortices (Supplementary Fig. 14e). Given the previous studies showing an increase in the size and number of lysosomes by PGRN deficiency[47], these results suggest that enlarged

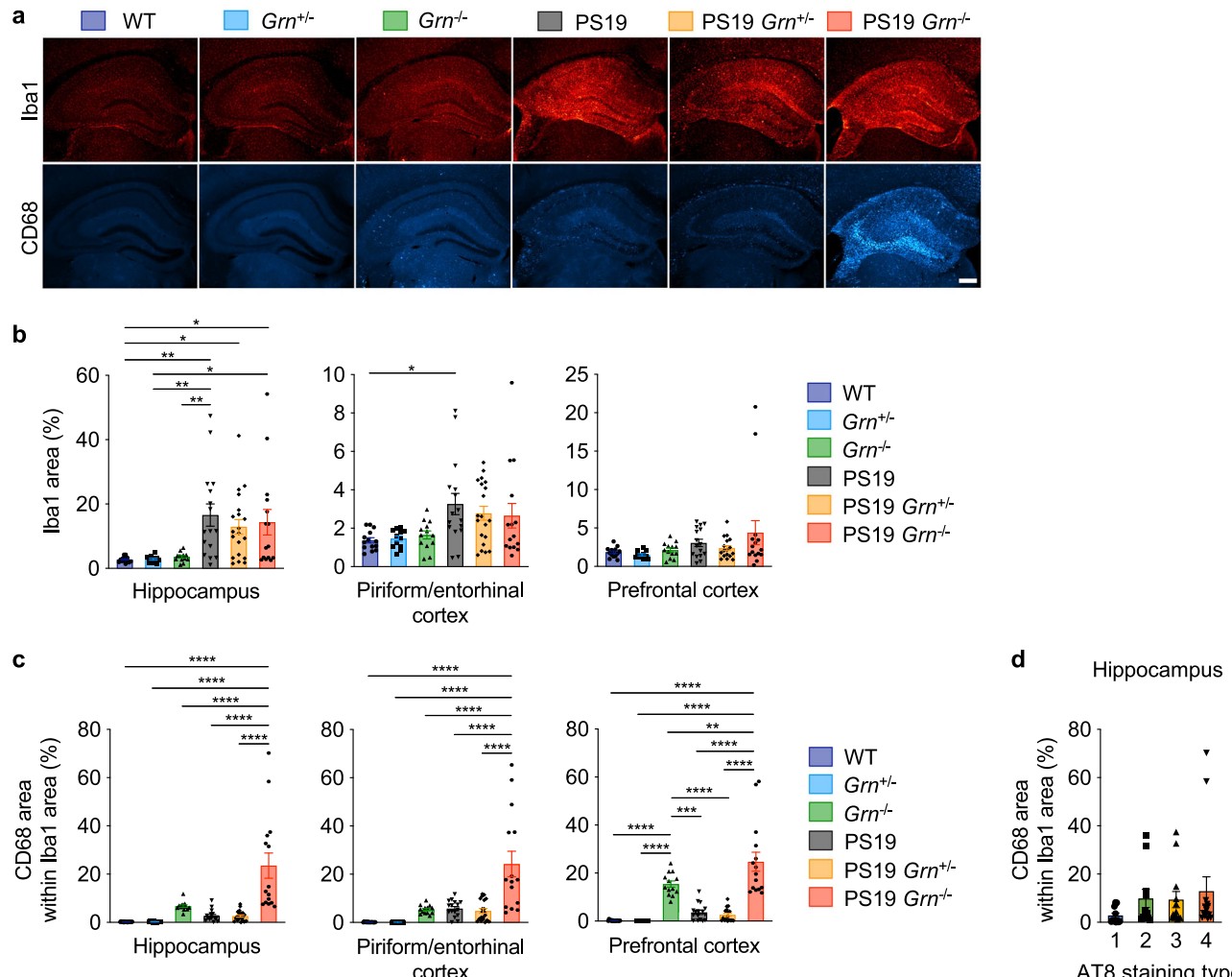

**Fig. 4 | CD68-positive microglia are not associated with increased tau inclusions in PS19 mice with PGRN reduction. a** Representative images of Iba1 and CD68 double staining of the hippocampus of 9–12-month-old 6 genotypes. Bar, 300 μm. **b** Quantification of Iba1 area and in 9–12-month-old 6 genotypes. Mean ± SEM, $n = 14$ mice (WT), $n = 12$ mice (Grn$^{+/-}$), $n = 13$ mice (Grn$^{-/-}$), $n = 16$ mice (PS19), $n = 20$ mice (PS19 Grn$^{+/-}$), $n = 15$ mice (PS19 Grn$^{-/-}$), **$p = 0.0032$ (hippocampus, WT vs. PS19), *$p = 0.0445$ (hippocampus, WT vs. PS19 Grn$^{+/-}$), *$p = 0.0254$ (hippocampus, WT vs. PS19 Grn$^{-/-}$), **$p = 0.0057$ (hippocampus, Grn$^{+/-}$ vs. PS19), *$p = 0.0375$ (hippocampus, Grn$^{+/-}$ vs. PS19 Grn$^{-/-}$), **$p = 0.0082$ (hippocampus, Grn$^{-/-}$ vs. PS19), *$p = 0.0240$ (piriform/entorhinal cortex, WT vs. PS19); One-way ANOVA with Tukey's post hoc test. **c** Quantification of CD68 area within Iba1 area (%) in 9–12-month-old 6 genotypes. Mean ± SEM, $n = 14$ mice (WT), $n = 12$ mice (Grn$^{+/-}$), $n = 13$ mice (Grn$^{-/-}$), $n = 16$ mice (PS19), $n = 20$ mice (PS19 Grn$^{+/-}$), $n = 15$ mice (PS19 Grn$^{-/-}$), ****$p = 0.000000001$ (hippocampus, WT vs. PS19 Grn$^{-/-}$), ****$p = 0.000000004$ (hippocampus, Grn$^{+/-}$ vs. PS19 Grn$^{-/-}$), ****$p = 0.00001$ (hippocampus, Grn$^{-/-}$ vs. PS19 Grn$^{-/-}$), ****$p = 0.00000001$ (hippocampus, PS19 vs. PS19

Grn$^{-/-}$), ****$p = 0.000000002$ (hippocampus, PS19 Grn$^{+/-}$ vs. PS19 Grn$^{-/-}$), ****$p = 0.000000001$ (piriform/entorhinal cortex, WT vs. PS19 Grn$^{-/-}$), ****$p = 0.000000004$ (piriform/entorhinal cortex, Grn$^{+/-}$ vs. PS19 Grn$^{-/-}$), ****$p = 0.000002$ (piriform/entorhinal cortex, Grn$^{-/-}$ vs. PS19 Grn$^{-/-}$), ****$p = 0.000001$ (piriform/entorhinal cortex, PS19 vs. PS19 Grn$^{-/-}$), ****$p = 0.00000007$ (piriform/entorhinal cortex, PS19 Grn$^{+/-}$ vs. PS19 Grn$^{-/-}$), ****$p = 0.000003$ (prefrontal cortex, WT vs. Grn$^{-/-}$), ****$p < 1e$-15 (prefrontal cortex, WT vs. PS19 Grn$^{-/-}$), ****$p = 0.00001$ (Grn$^{+/-}$ vs. Grn$^{-/-}$), ****$p < 1e$-15 (prefrontal cortex, Grn$^{+/-}$ vs. PS19 Grn$^{-/-}$), ***$p = 0.0003$ (prefrontal cortex, Grn$^{-/-}$ vs. PS19), ****$p = 0.00004$ (prefrontal cortex, Grn$^{-/-}$ vs. PS19 Grn$^{+/-}$), **$p = 0.0087$ (prefrontal cortex, Grn$^{-/-}$ vs. PS19 Grn$^{-/-}$), ****$p = 0.000000000005$ (prefrontal cortex, PS19 vs. PS19 Grn$^{-/-}$), ****$p < 1e$-15 (prefrontal cortex, PS19 Grn$^{+/-}$ vs. PS19 Grn$^{-/-}$); One-way ANOVA with Tukey's post hoc test. **d** No association of CD68 area within Iba1 area with AT8 staining patterns in the hippocampus. Mean ± SEM, $n = 51$ mice. Four groups in each graph are not significantly different by Kruskal-Wallist test. Source data are provided as a Source Data file.

dysfunctional lysosomes might be enriched in fractions #3 and #4 from Grn$^{-/-}$ cortices.

GCase activity assay using fractions #2 and #3 revealed that PGRN deficiency causes a 20% decrease in total and GBA1-specific activities in fraction #2, while increasing those activities in fraction #3 (Supplementary Fig. 15a, b). Consistent with the results from 5-month-old animals (Supplementary Fig. 9d), GCase activity assay using cortical lysates from the other hemisphere of the same animals showed 5–10% decrease of total and GBA1-specific GCase activity in Grn$^{-/-}$ cortices. The presence of taurocholate, which presumably mimics the function of Sap C, in the assay buffer had no significant impacts on the changes in GCase activity by PGRN deficiency, suggesting that the effects of

PGRN deficiency are not due to altering subcellular localization of Sap C. Indeed, immunoblot analysis showed no significant difference in Sap C levels of fractions #2 or #3 between WT and Grn$^{-/-}$, while PGRN deficiency caused a decrease of GCase levels in fraction #2 of Grn$^{-/-}$ cortices (Supplementary Fig. 15b, c). As expected, in fractions #2 and #3, GCase levels by immunoblot correlated highly with GBA1 activity (Supplementary Fig. 15d). Similar results were obtained in 11.5-month-old animals (Supplementary Fig. 16). Taken together, these results suggest that PGRN deficiency alters the appropriate subcellular localization of GCase, but not Sap C. Accumulation of GCase into dysfunctional lysosomes likely contributes to the increase in GlcCer levels.

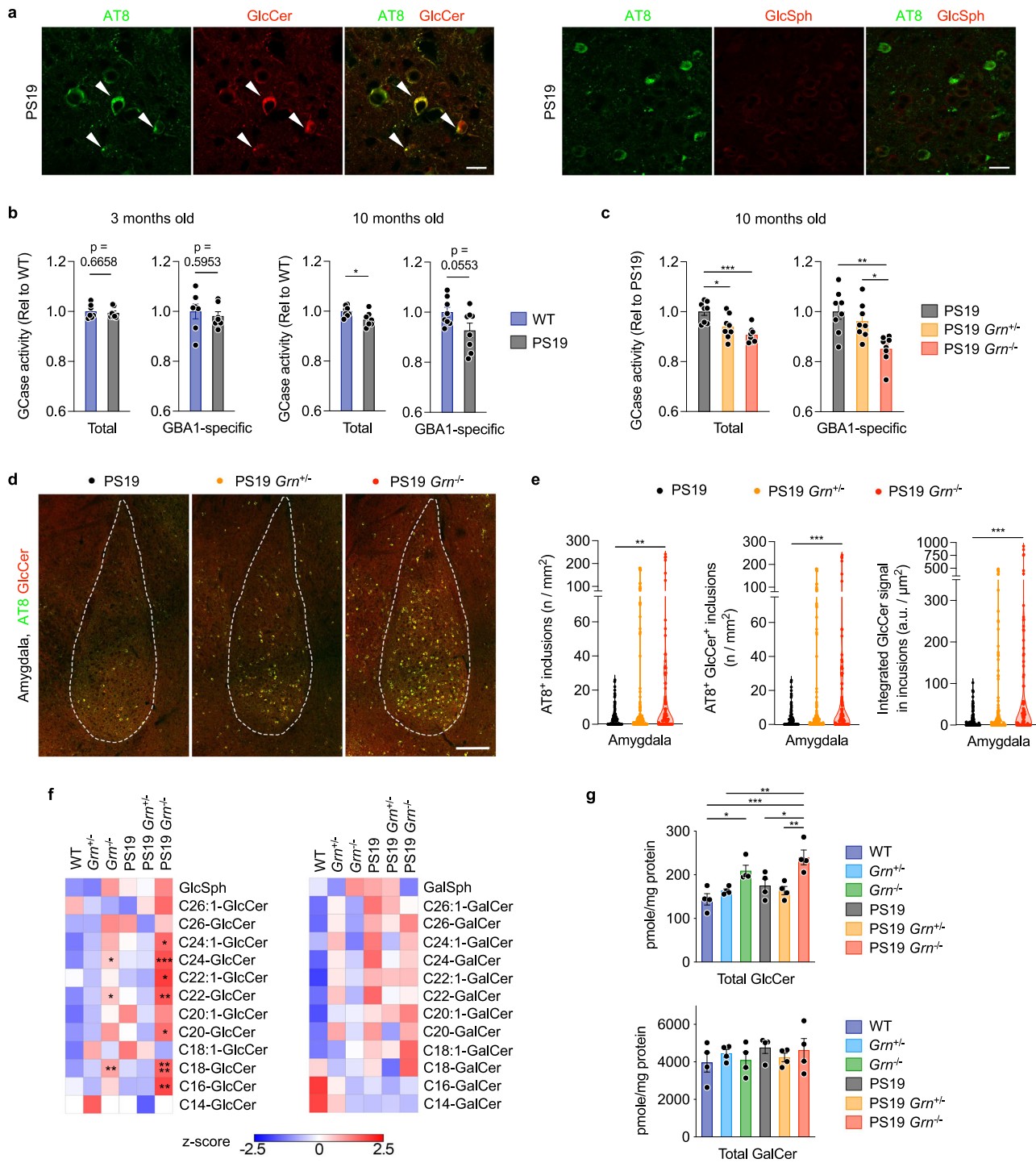

## PGRN reduction increases co-accumulation of p-α-syn in PS19 mice

Given the known prominent association of GCase activity with synucleinopathy[45,46], we also examined α-syn pathology in 3 genotypes (PS19, PS19 *Grn*[+/−] and PS19 *Grn*[−/−]). Immunohistochemical analysis using AT8 and anti-phospho-Ser129 (p-α-syn) antibodies revealed a sparse p-α-syn immunoreactivity that is colocalized with AT8-positive inclusions in PS19 brains (Fig. 6a, c). Importantly, PGRN reduction caused a significant increase in AT8 and p-α-syn double positive inclusions in the CA2 + CA3, amygdala, and piriform cortex of PS19 mice (Fig. 6a, c and Supplementary Fig. 17a, c, d). No AT8 and p-α-syn double positive inclusions were observed in WT, *Grn*[+/−], and *Grn*[−/−] brains without PS19 background (Supplementary Fig. 18). Furthermore,

using immunohistochemistry with anti-GCase antibody we found a significant incorporation of GCase in the p-α-syn double positive inclusions of PS19 mice with PGRN reduction (Fig. 6b and Supplementary Fig. 17b). These co-aggregates were not immunoreactive for TDP-43 (Supplementary Fig. 17e). Thus, PGRN reduction exacerbated α-synuclein and GCase but not TDP-43 co-accumulation in PS19 mice.

## GlcCer and GCase promote tau aggregation in vitro

To further investigate the role of GCase and GlcCer in tau aggregation, we examined the effect of CBE in the well-established tau seeding assay using primary cultured neurons[72–75]. In this assay, seeded tau fibrils extracted from AD brains (AD-tau) cause aggregation of endogenous mouse tau in both MAP2-positive dendrites and neurofilament light

**Fig. 5 | PGRN reduction attenuates GCase activity while tau inclusions are immunoreactive for GlcCer. a** Representative confocal images of AT8 and GlcCer or GlcSph co-staining in the amygdala region of PS19 mice. Bar, 20 μm. **b** Total and GBA1-specific GCase activity of WT and PS19 cortices at 3 and 10 months of age. Mean ± SEM, $n$ = 7 mice (3 months old) or 8 mice (10 months old) per genotype. $*p$ = 0.0159; Two-tailed unpaired $t$-test. **c** Total and GBA1-specific GCase activity of PS19, PS19 $Grn^{+/-}$, and PS19 $Grn^{-/-}$ cortices at 10 months of age. Mean ± SEM, $n$ = 8 mice per genotype. $*p$ = 0.0192 (total, PS19 vs. PS19 $Grn^{+/-}$), $***p$ = 0.0005 (total, PS19 vs. PS19 $Grn^{-/-}$), $**p$ = 0.0014 (GBA1-specific, PS19 vs. PS19 $Grn^{-/-}$), $**p$ = 0.0159 (GBA1-specific, PS19 $Grn^{+/-}$ vs. PS19 $Grn^{-/-}$); One-way ANOVA with Tukey's post hoc test. **d** Representative confocal images of AT8 and GlcCer co-staining in the amygdala region of PS19, PS19 $Grn^{+/-}$ or PS19 $Grn^{-/-}$ mice. Bar, 200 μm. **e** The number of AT8-positive inclusions, AT8 and GlcCer double-positive inclusions and integrated GlcCer intensity in AT8 and GlcCer double-positive inclusions in the amygdala of 3 genotypes at 9–12 months of age. $n$ = 15–20 mice per genotype,

6 slices per mouse, $**p$ = 0.0019 (AT8$^+$ inclusions, PS19 vs. PS19 $Grn^{-/-}$), $***p$ = 0.0007 (AT8$^+$ GlcCer$^+$ inclusions, PS19 vs. PS19 $Grn^{-/-}$), $***p$ = 0.0001 (integrated GlcCer signal in inclusions, PS19 vs. PS19 $Grn^{-/-}$), $***p$ < 0.001; Kruskal-Wallis test with Dunn's post hoc test. **f** Heatmaps showing levels of GlcCer and GalCer species in the cortex of 6 genotypes at 10 months of age. $n$ = 4 mice per genotype. $*p$ = 0.0269 (C24:1-GlcCer), $*p$ = 0.0258 (C24-GlcCer), $***p$ = 0.0003 (C24-GlcCer), $*p$ = 0.0324 (C22:1-GlcCer), $*p$ = 0.0492 (C22-GlcCer), $**p$ = 0.0012 (C22-GlcCer), $*p$ = 0.0179 (C20-GlcCer), $**p$ = 0.0014 (C18-GlcCer), $****p$ < 0.0001 (C18-GlcCer), $**p$ = 0.0046 (C16-GlcCer); One-way ANOVA with Dunnett's post hoc test comparing to WT. **g** Total GlcCer or GalCer levels in the cortex of 6 genotypes at 10 months of age. Mean ± SEM, $n$ = 4 mice per genotype. $*p$ = 0.0169 (total GlcCer, WT vs. $Grn^{-/-}$), $***p$ = 0.0004 (total GlcCer, WT vs. PS19 $Grn^{-/-}$), $**p$ = 0.0046 (total GlcCer, $Grn^{+/-}$ vs. PS19 $Grn^{-/-}$), $*p$ = 0.0188 (total GlcCer, PS19 vs. PS19 $Grn^{-/-}$), $**p$ = 0.0048 (total GlcCer, PS19 $Grn^{+/-}$ vs. PS19 $Grn^{-/-}$); One-way ANOVA with Tukey's post hoc test. Source data are provided as a Source Data file.

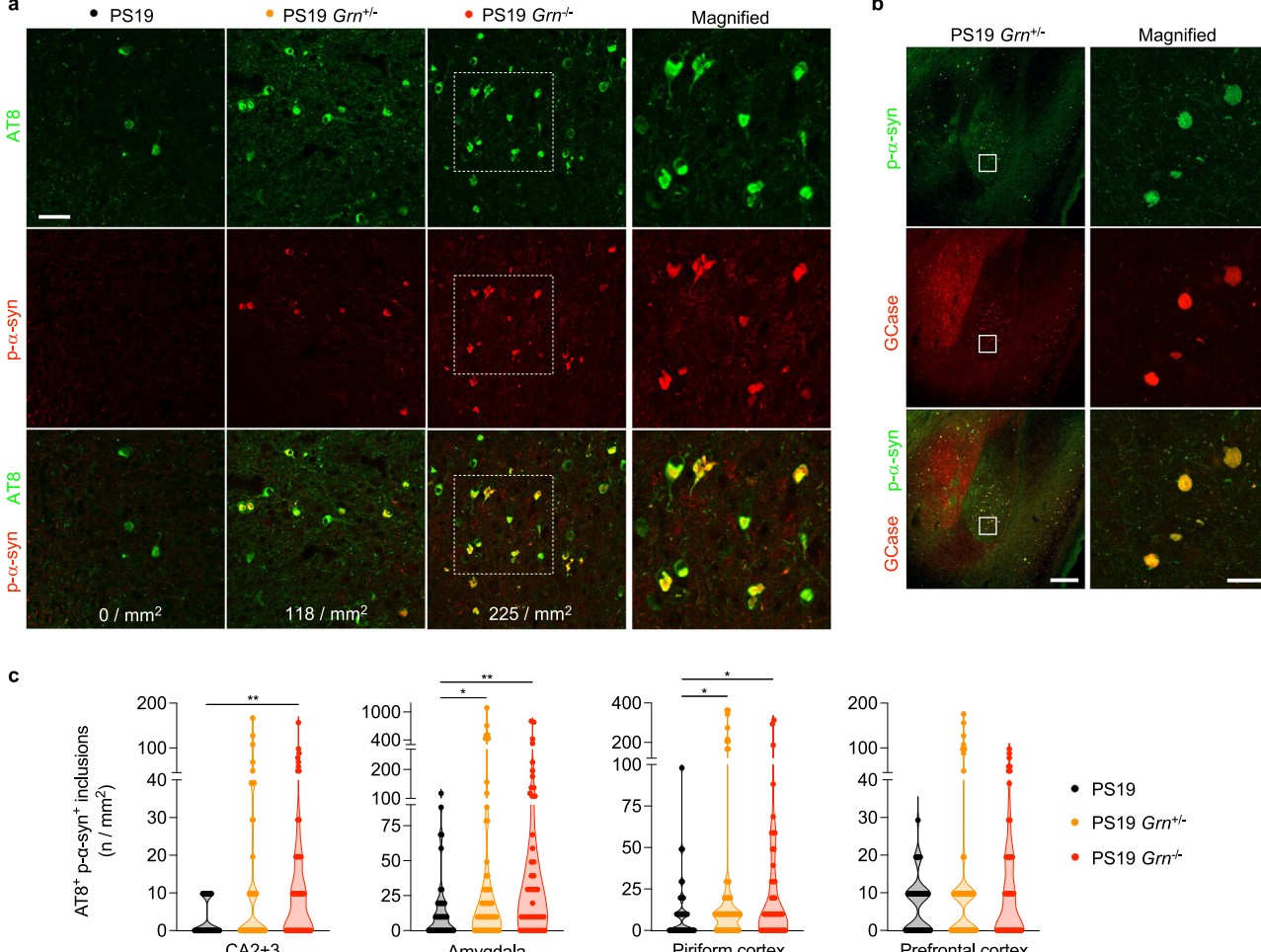

**Fig. 6 | PGRN reduction exacerbates α-syn co-accumulation in PS19 mice.**
**a** Representative confocal images of AT8 and p-α-syn co-staining in the amygdala region of PS19, PS19 $Grn^{+/-}$ or PS19 $Grn^{-/-}$ mice. Square dotted boxes indicate regions where magnified images are presented. The number of AT8 and p-α-syn double positive inclusions was shown on the bottom of each panel. Bar, 50 μm. **b** Representative confocal images of p-α-syn and GCase co-staining in the amygdala region of PS19 and PS19 $Grn^{+/-}$ mice. Bar, 200 μm. The right panels show a high magnification of white square area in the left panels. Bar, 20 μm. **c** Quantification of

the number of AT8 and p-α-syn double positive inclusions in the CA2 + 3, amygdala, piriform cortex, and prefrontal cortex of 3 genotypes (PS19, PS19 $Grn^{+/-}$, and PS19 $Grn^{-/-}$ mice) at 9–12 months of age. Mean ± SEM, $n$ = 15–20 mice per genotype, 3 ROIs per mouse, $**p$ = 0.0012 (CA2 + 3, PS19 vs. PS19 $Grn^{-/-}$), $*p$ = 0.0353 (amygdala, PS19 vs. PS19 $Grn^{+/-}$), $**p$ = 0.0020 (amygdala, PS19 vs. PS19 $Grn^{-/-}$), $*p$ = 0.0350 (piriform cortex, PS19 vs. PS19 $Grn^{+/-}$), $*p$ = 0.0270 (piriform cortex, PS19 vs. PS19 $Grn^{-/-}$), $**p$ < 0.01; Kruskal-Wallis test with Dunn's post hoc test. Source data are provided as a Source Data file.

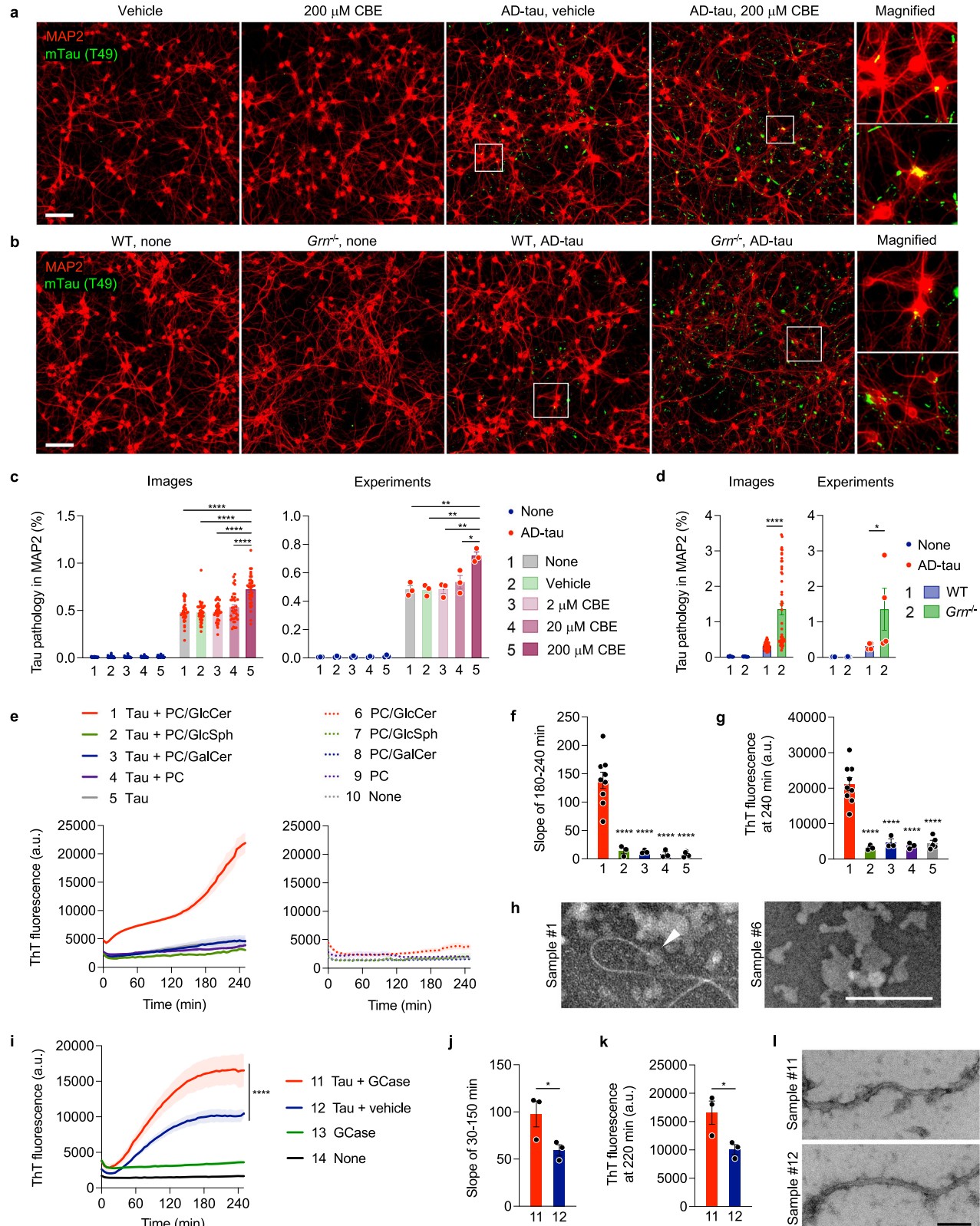

chain-positive axons of primary neurons[74]. The aggregation of endogenous mouse tau can be specifically detected by a mouse tau-specific antibody (T49) in methanol-fixed neurons[73,74]. We seeded AD-tau onto neurons with CBE or vehicle treatment at DIV7 and incubated for two weeks. Immunofluorescence using T49 antibody revealed that a high concentration of CBE significantly promotes tau aggregation induced by AD-tau in MAP2-positive dendrites (Fig. 7a, c). Similarly, we also observed that PGRN deficiency increases tau aggregation induced by AD-tau in neurons although no tau aggregation was detected in *Grn*[−/−] neurons without AD-tau seeding (Fig. 7b, d). Interestingly, we also observed an increase in AT8 mean intensity in the piriform cortex of WT mice treated with CBE for 21 days although AT8-positive tau inclusions were not detected during this short duration experiment (Supplementary Fig. 9j).

**Fig. 7 | GlcCer and GCase promote tau aggregation in vitro. a** Representative images of mouse tau (T49) and MAP2 co-staining of WT neurons at DIV21 after AD-tau seeding in the presence of vehicle or 200 µM CBE. Bar, 100 µm. **b** Representative images of mouse tau (T49) and MAP2 co-staining of WT and $Grn^{-/-}$ neurons at DIV21 after seeding AD-tau. Bar, 100 µm. **c** Quantification of endogenous mouse tau-positive area within MAP2 area. Neurons were seeded with AD-tau in the absence or presence of vehicle, 2 µM CBE, 20 µM CBE or 200 µM CBE. Experiments were performed in triplicate, and four images were unbiasedly taken from each well. Mean ± SEM, *n* = 36 images or 3 experiments, ****$p$ = 0.000000000006 (images, none vs. 200 µM CBE), ****$p$ = 0.000000000003 (images, vehicle vs. 200 µM CBE), ****$p$ = 0.000000000007 (images, 2 µM CBE vs. 200 µM CBE), ****$p$ = 0.00000008 (images, 20 µM CBE vs. 200 µM CBE), **$p$ = 0.0024 (experiments, none vs. 200 µM CBE), **$p$ = 0.0021 (experiments, vehicle vs. 200 µM CBE), **$p$ = 0.0024 (experiments, 2 µM CBE vs. 200 µM CBE), *$p$ = 0.0129 (experiments, 20 µM CBE vs. 200 µM CBE); One-way ANOVA with Tukey's post hoc test in 4 groups treated with AD-tau. **d** Quantification of endogenous mouse tau-positive area within MAP2 area. Experiments were performed in triplicate, and four images were unbiasedly taken from each well. Mean ± SEM, *n* = 48 images or 4 experiments, *$p$ = 0.0286, ****$p$ = 0.000000000001; Two-tailed Mann-Whitney test in 2 groups treated with AD-tau. **e** ThT assay of heparin-induced aggregation of P301S tau in the presence or absence of PC, PC/GlcCer, PC/GlcSph, or PC/GalCer lipid dispersions. Heparin was included in all samples except for the sample #10. Mean ± SEM, *n* = 9 (for the samples #1 and 6), 5 (for the samples #5 and 10), or 3 (for the other samples) experiments, each preformed in triplicate. **f** Slopes determined by linear regression from 180 to 240 min in (**e**). ****$p$ = 0.00005 (1 vs. 2), ****$p$ = 0.00004 (1 vs. 3), ****$p$ = 0.00003 (1 vs. 4), ****$p$ = 0.000002 (1 vs. 5); One-way ANOVA with Tukey's post hoc test. **g** ThT fluorescence at 240 min in (**e**). ****$p$ = 0.00001 (1 vs. 2), ****$p$ = 0.00004 (1 vs. 3), ****$p$ = 0.00002 (1 vs. 4), ****$p$ = 0.000003 (1 vs. 5); One-way ANOVA with Tukey's post hoc test. **h** Representative negative-stain EM images of the samples #1 and #6. An arrowhead indicates a tau fibril elongated from PC/GlcCer lipid dispersions. Bar, 200 nm. **i** ThT assay of heparin-induced aggregation of P301S tau in the presence or absence of GCase. Instead of GCase, the same volume of 50 mM sodium citrate pH 5.5 was added as a vehicle control to the samples #12 and #14. Heparin was included in all samples except for the sample #14. Mean ± SEM, *n* = 3 experiments, each preformed in triplicate. ****$p$ < 1e-15; nonlinear regression with a sigmoidal model followed by two-sided extra sum-of-squares *F* test. **j** Slopes determined by linear regression from 30 to 150 min in (**i**). *$p$ = 0.0438; Two-tailed paired *t*-test. **k** ThT fluorescence at 220 min in (**i**). *$p$ = 0.0338; Two-tailed paired *t*-test. **l** Representative negative-stain EM images of samples #11 and #12. Bar, 100 nm. Source data are provided as a Source Data file.

To examine whether accumulation of GlcCer by GCase inhibition directly affects tau aggregation, we next performed in vitro Thioflavin T (ThT) assay using recombinant P301S tau and lipid dispersions made of purified phosphatidylcholine (PC) with or without GlcCer, GlcSph, or GalCer. We found that heparin-induced aggregation of P301S tau was dramatically accelerated in the presence of PC/GlcCer lipid dispersions, but not the other lipid dispersions, while the lipid dispersions have no effects on ThT fluorescence (Fig. 7e–g). Negative-stain EM analysis showed the presence of tau fibrils that appear to be elongated from the PC/GlcCer lipid dispersions (Fig. 7h). Together, these results suggest a possibility that GlcCer directly accelerates tau aggregation.

Given the incorporation of GCase in tau inclusions in PS19 mice with PGRN reduction, we also examined whether GCase affects tau aggregation in vitro. Similar to GlcCer, in vitro ThT assay revealed that recombinant GCase significantly accelerates heparin-induced P301S tau aggregation, while GCase alone did not spontaneously aggregate at the concentration we tested (Fig. 7i–k). We confirmed that a control protein (BSA) has no effects on the tau aggregation, excluding a possibility that total protein concentration modulates tau aggregation in this in vitro experimental setting (Supplementary Fig. 19). Negative-stain EM analysis showed that there is no significant morphological difference between fibrils with and without GCase (Fig. 7l). These results suggest that GCase protein itself could also promote tau aggregation.

**NFTs in human tauopathy brains are immunoreactive for GlcCer**
Based on these mouse model and in vitro experiments, we investigated the relationship between tau, PGRN, GCase, and α-syn in human autopsy brain with tauopathy. We first examined PGRN expression in tau pathology of AD. Similar to the results obtained from PS19 mice, we found no or very weak PGRN immunoreactivity in AT8-positive neurofibrillary tangles (NFTs) and neuritic plaques (NPs) of AD brains, respectively, even though PGRN signal was detected in neurons and significantly increased in dystrophic neurites near amyloid plaques (Fig. 8a and Supplementary Fig. 20b), which is largely consistent with a recent publication[76].

We then examined whether the co-aggregation of GlcCer and GCase with tau found in PS19 mice with PGRN reduction is also observed in tau pathologies in anterior hippocampal and middle frontal gyrus sections from 5 AD brains, 4 with moderate AD pathological changes (AH2), and 1 frontotemporal dementia (FTD)-*GRN*. Strikingly, we detected high GlcCer positivity in NFTs of the anterior hippocampus in all of five AD brains and four AH2 brains we tested (Fig. 8b, d and Supplementary Fig. 20c). In addition, GlcCer signal was partially colocalized with NPs in AD brains (Fig. 8b and Supplementary Fig. 20d). The GlcCer immunoreactivity was also detected in NFTs of DLB brains, while Lewy bodies of the DLB brains were largely negative for aggregated GlcCer (Fig. 8c, d). Furthermore, we observed a partial colocalization of GlcCer and GCase with NPs in the middle frontal gyrus of an FTD-*GRN* brain with AD pathology (Fig. 8e). We further examined GCase and p-α-syn immunoreactivities using another set of paraffin-embedded hippocampal sections of 4 AD and 2 FTD-*GRN* brains. While mature NFTs showed low GCase positivity, immature NFTs and NFTs found in FTD-*GRN* exhibited higher GCase positivity (Fig. 8f, g). In 2 AD brains, we found dot-like aggregates of p-α-syn in a small subset of AT8-positive NFTs (Supplementary Fig. 20e). Similar dot-like aggregates of p-α-syn were also observed in our PS19 Grn cohort (Supplementary Fig. 17c, d). Of the AT8 and p-α-syn double positive neurons, more than 60% showed GCase immunoreactivity (Supplementary Fig. 20f, g). Together, these results suggest GlcCer involvement and potential co-aggregation of GCase and/or α-syn in human tauopathy.

## Discussion
### The role for a lysosomal PGRN-GCase complex in tau aggregation
A striking observation of the present study is that tau inclusions in PS19 neurons were consistently immunoreactive for the glycosphingolipid GlcCer, a substrate of GCase. This was unexpected because GCase has been previously associated with synucleinopathy[45,46] rather than tauopathy. However, there are several recent studies showing concomitant tau pathology in the brains of GCase mutant mice and Gaucher disease patients in which GCase is mutated[59–61,77]. Importantly, in the present study, NFTs and some of NPs of human AD, DLB, and FTLD-*GRN* brains were also immunoreactive for GlcCer, although the human pathology should be confirmed using additional cohorts. Consistent with our observation, previous studies have revealed the presence of glycolipids or cerebrosides in paired helical filaments of AD[78,79]. In neuronal culture, we showed that GCase inhibition exacerbates tau aggregation induced by AD-tau. In vitro, PC/GlcCer lipid dispersions dramatically promoted heparin-induced aggregation of P301S tau. Together, these results clearly suggest a critical role of GlcCer in tau aggregation. In future studies, it will be interesting to examine the prevalence of tau pathology in a large cohort of human subjects with GCase mutations.

We then found that both complete loss and haploinsufficiency of PGRN impair GCase activity while increasing GlcCer levels and GlcCer-positive tau inclusions in PS19 brains. Importantly, GlcCer was the only

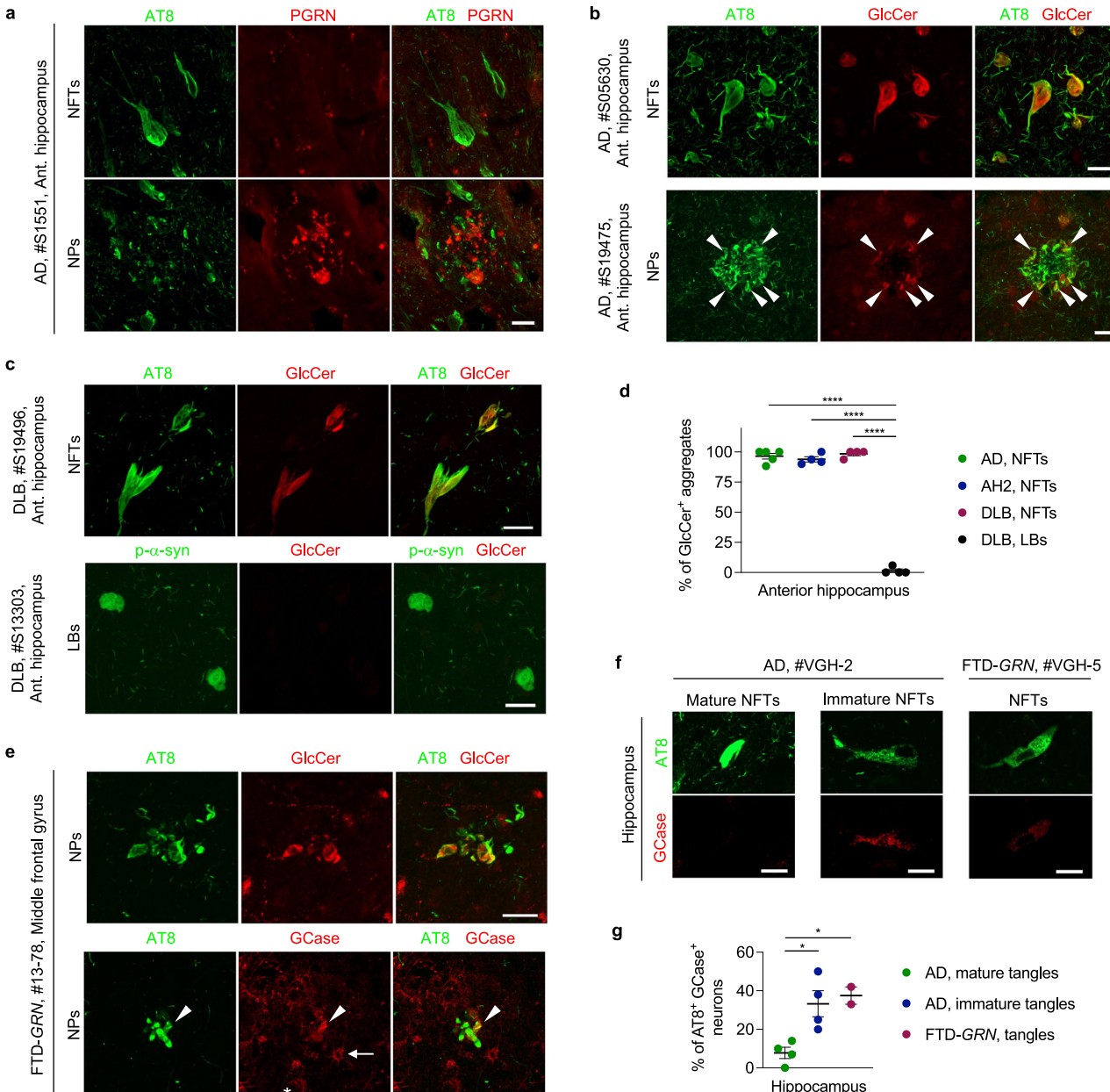

**Fig. 8 | Accumulation of GlcCer and GCase to tau aggregates in AD, DLB, and FTD-*GRN* patients. a** Representative confocal images of AT8 and PGRN co-staining of NFTs and NPs in the anterior hippocampus region of AD brain. Bar, 20 μm. Similar results were obtained in at least two other free-floating AD samples (S15147 and S19475) and one paraffin-embedded AD sample (VGH-2, shown in Supplementary Fig. 20b). **b** Representative confocal images of AT8 and GlcCer co-staining of NFTs and NPs in the anterior hippocampus region of AD brains. Arrowheads indicate GlCer labeling of some of NPs. Bar, 20 μm. **c** Representative confocal images of AT8 and GlcCer co-staining of NFTs and LBs in the anterior hippocampus region of DLB brains. Bar, 20 μm. **d** Quantification of percentage of GlcCer+ aggregates of NFTs of AD, AH2, DLB brains and LBs of DLB brain tissue. At least 10 NFTs were examined in each sample except for two DLB samples due to the low numbers of NFTs (2 and 6 NFTs). The mean number of NFTs examined in each group is as follows: AD = 15 ± 1.924, AH2 = 13.75 ± 1.652, DLB = 8.5 ± 2.986. At least 11 LBs were examined for each DLB brain, and the mean number is 14 ± 1.472. Mean ± SEM, $n = 5$ brains (AD, NFTs), $n = 4$ brains (the other groups) ****$p = 0.000000000004$; One-way ANOVA with Tukey's post hoc test.

**e** Representative confocal images of AT8 and GlcCer or GCase of co-staining of NPs in the middle frontal gyrus of an FTD-*GRN* brain. Bar, 20 μm. An arrowhead indicates co-localization of GCase with AT8-positive neuritic plaques. An arrow indicates glial localization of GCase. An asterisk indicates lysosomal GCase in neurons. Bar, 20 μm. **f** Representative confocal images of AT8 and GCase of co-staining of NFTs in the hippocampus of AD and FTD-*GRN* brain. Bar, 20 μm. **g** Quantification of percentage of AT8+ GlcCer+ aggregates of NFTs in the hippocampus of AD and FTD-*GRN* brain. NFTs of AD were divided into two groups: mature and immature NFTs, based on their morphology. At least 48 NFTs for each group were examined in each sample except for one FTD-*GRN* sample due to the low number of NFTs (15 NFTs). The mean number of NFTs examined in each group is as follows: AD, mature NFTs = 56.25 ± 3.75, AH2, immature NFTs = 58.00 ± 4.673, FTD-*GRN* NFTs = 35.5 ± 20.50. Mean ± SEM, $n = 4$ patients (AD, mature NFTs and AD, immature NFTs), $n = 2$ (FTD-*GRN*, NFTs), *$p = 0.0202$ (AD, mature tangles vs. AD, immature tangles), *$p = 0.0253$ (AD, mature tangles vs. FTD-*GRN*, tangles); One-way ANOVA with Tukey's post hoc test. Source data are provided as a Source Data file.

lipid class that was significantly increased by PGRN deficiency in our comprehensive lipidomic analysis. These results suggest that PGRN regulates formation of tau inclusions via GCase by limiting accumulation of GlcCer. Several models have been proposed for the molecular mechanism by which PGRN affects GCase activity[11,56–58,70,80]. Those include direct chaperone activity of PGRN, lysosomal delivery of GCase, prosaposin, or Sap C, and interaction with BMP. In the present study, our co-IP assay reproduced interaction between PGRN and GCase. We also confirmed that PGRN reduction causes ~10% decrease in cortical brain lysate. Our GCase activity assay and immunoblot analysis using lysosome-enriched fractions showed that PGRN deficiency alters subcellular localization of GCase but not Sap C. Additionally, a decrease in BMP levels was observed only in PGRN-null brains while both complete loss and haploinsufficiency of PGRN decreased GCase activity. Therefore, it appears that our results support the hypothesis that PGRN binds to GCase and assists its maturation and lysosomal localization, although other possibilities cannot be excluded.

GCase protein itself was also incorporated into tau inclusions in PS19 mice with PGRN reduction. In vitro, similar to GlcCer, recombinant GCase protein promoted heparin-induced aggregation of P301S tau. PGRN deficiency reportedly causes insolubility or aggregation of GCase in both humans and rodents[57,58]. Thus, PGRN binding to GCase may also be crucial to stabilize GCase protein thereby preventing its co-aggregation with tau. Recent studies have shown a role for microglial activation in driving tau pathology[39,52–54], and PGRN plays an important role in microglial activation[5,6,55]. However, in the present study, the extent of CD68+ microgliosis was not associated with either AT8 staining types or the number of tau inclusions. It is thus unlikely that PGRN reduction accelerates formation of tau inclusions via microglial activation.

Interestingly, we observed that neurons with tau inclusions exhibit little or no PGRN expression in both PS19 mice and human AD brains. Consistent with a previous study[76], PGRN immunoreactivity was absent in NFTs. As previously reported[8,76,81], our immunohistochemistry showed that PGRN was highly expressed in dystrophic neurites near amyloid plaques in AD. However, the PGRN-positive dystrophic neurites were poorly colocalized with AT8-positive NPs. We have found that GCase activity is a slightly but significantly attenuated in the cortex of PS19 mice at 10 months of age. Thus, there may be a positive feedback mechanism by which pathological tau affects PGRN expression and GCase activity to exacerbate its accumulation. Further investigation will be required to explore this possibility.

## Implications of increased α-syn co-pathology in PS19 mice with PGRN reduction

In the present study, PGRN reduction also increased α-syn co-accumulation in PS19 mice. Concomitant proteinopathies are frequently observed in aged brain with neurodegenerative diseases and influence their clinical course[2–4], suggesting that therapies targeting pathways that affect co-pathologies may be more effective than monotherapies focusing on a single disease-associated protein. Our results in the present study suggest that PGRN reduction may cause tau and α-syn accumulation by impairing GCase activity and increasing GlcCer levels in FTLD-GRN and in other neurodegenerative diseases, consistent with a large body of literature showing an association between GCase and synucleinopathy[45,46]. This also suggests that a lysosomal PGRN–GCase complex may be a common therapeutic target for comorbid proteinopathies in neurodegenerative disease. Similar to the effects of GlcCer on tau in the present study, a previous study has shown direct effects of GlcCer on α-syn aggregation[64]. Destabilization and aggregation of GCase by PGRN reduction may also contribute to the tau and α-syn co-accumulation, since incorporation of GCase into Lewy bodies is reported in patients with parkinsonism[46].

Interestingly, in addition to the increased co-accumulation of α-syn, PS19 mice with PGRN deficiency exhibited synergistic body weight loss and increased mortality at least in part due to hindlimb paresis. A recent genome-wide association study has suggested that GRN variants also increase PD risk[25]. Therefore, it will be important to further understand how PGRN reduction influences PD phenotypes using animal and cellular models of synucleinopathy.

## Differential effects of PGRN deficiency on symptoms of tauopathy

In the present study, our results revealed a differential role of PGRN in memory and disinhibited behaviors in the setting of tauopathy. PGRN reduction worsens disinhibited behaviors while improving a spatial memory deficit in PS19 mice. The improvement was coupled with attenuated neurodegeneration and transcriptomic changes in the hippocampus. PS19 mice, a tauopathy model expressing human P301S tau under the control of the mouse prion protein promoter, display a wide range of behavioral deficits observed in human tauopathies such as AD and FTLD[40,41]. In FTD patients, memory function is preserved until late stage of the disease, while disinhibition is a key clinical symptom of behavioral variant FTD patients[82]. Similarly, in the presence of tau pathology, reduction of PGRN, an FTD-associated protein, might attenuate tau-driven memory impairment while exacerbating FTD-like disinhibited behavior in PS19 mice.

Intriguingly, in PS19 mice with PGRN reduction, the rescue of memory impairment and neurodegeneration was accompanied by an increase in tau inclusions. This finding seems somewhat contradictory but is consistent with recent studies using PS19 mice. Type 2 AT8 staining pattern, which is characterized by frequent tangle-like staining, has been shown to correlate with less brain atrophy, while type 4, strong but diffuse staining without large somatic inclusions, was frequently observed in brains with severe atrophy[36]. Another published study showed a rescue of neurodegeneration and memory deficits in parallel with an increase in NFTs[83]. Thus, formation of somatic tau inclusions might protect against hippocampal neurodegeneration and functions, although it is important to validate the protective role of tau inclusions using different approach and tauopathy models.

How PGRN reduction and tauopathy synergistically worsen disinhibited behaviors is less clear. In the present study, we also observed a trend toward an increase in co-accumulation of tau and α-syn in the prefrontal cortex, a brain region implicated in executive functions including inhibitory control[84]. An increase in co-pathology of tau and α-syn might perturb neuronal networks regulating inhibitory control in the prefrontal cortex despite rescue of neurodegeneration in the hippocampus.

The differential effects on symptoms have a significant implication for PGRN's role in AD. While improvement of memory in PS19 mice with PGRN reduction might not explain the mechanism of increased AD risk by GRN rs5848 T allele, exacerbation of disinhibited behaviors in combination with increased tau inclusions could do. It will be interesting to examine whether similar alteration of symptoms is observed in AD patients with the GRN rs5848 T allele.

## The effects of PGRN haploinsufficiency in PS19 mice

The present study shows clear effects of PGRN haploinsufficiency in PS19 mice, closely recapitulating those of the GRN mutations or rs5848 variant in AD. This is in contrast to most, but not all, previous studies describing no obvious effects of PGRN haploinsufficiency in WT mice[5,6]. Attenuated behavioral deficits and neurodegeneration, increase in tau and p-α-syn inclusions, reduced GCase activity were all observed not only in PS19 Grn⁻/⁻ mice but also in PS19 Grn⁺/⁻ mice. As described above, our data suggest that phenotypes observed in the present study are due, at least in part, to reduced GCase activity and stability in PS19 mice

with PGRN reduction. Thus, direct and primary effects of loss of lysosomal PGRN-GCase complex after PGRN reduction could explain obvious phenotypes in PS19 $Grn^{+/-}$ mice. This raises an intriguing possibility that phenotypes due to reduced GCase activity or levels might be observed even in $Grn$ heterozygous mice, although it also requires further investigation.

## Transcriptomic analysis of PGRN deficient mice

Bulk RNA-seq/transcriptomic analysis using PGRN deficient mice from our previous study and others has shown up-regulation of lysosomal genes and microglial TYROBP gene network including C1q genes even at early stage of mouse lifetime[47,85,86]. Our snRNA-seq detected only limited changes in such genes for $Grn^{-/-}$ nuclei without the PS19 transgene in comparison to WT. The statistically significant up-regulation of lysosomal genes in $Grn^{-/-}$ mice was in most instances less than two-fold in our Bulk RNA-seq[47]. We did confirm lysosomal and microglial changes (e.g., CD68) at protein levels in the hippocampus of $Grn^{-/-}$ mice by immunohistochemistry (Fig. 4), consistent with our previous studies[26,47]. The less prominent transcriptomic changes for these pathways in our snRNA-seq may relate to the significant pools of RNA outside the nucleus being discarded during nuclei isolation, similar to a recent study on a different topic[87].

In conclusion, the present study revealed an unexpected role of glycosphingolipid GlcCer in tau aggregation and demonstrated that a lysosomal PGRN-GCase complex titrates accumulation of GlcCer to regulate formation of tau and α-syn inclusions, which alters symptoms and neurodegeneration in tauopathy (Supplementary Fig. 20a). Our findings provide insights into the roles that lysosomes play in proteinopathy and advance our understanding of PGRN as a risk factor for AD and PD. It is likely that the PGRN-GCase complex plays an important role not only in Gaucher disease and FTLD but also in other neurodegenerative proteinopathies.

## Methods

### Animals

All protocols were approved by Yale Institutional Animal Care and Use Committee (IACUC) (2023-07281).

Human P301S tau transgenic (PS19) mice with B6C3F1 background[34] were obtained from the Jackson Laboratory. The mice express mutant human *MAPT* gene, which results in a five-fold greater amount of human mutant tau protein (P301S 1N4R tau) than endogenous mouse tau. $Grn^{-/-}$ mice with C57BL/6 J background[35] were obtained from the RIKEN Bioresource Center. To generate 6 genotypes used in this study (WT, $Grn^{+/-}$, $Grn^{-/-}$, PS19, PS19 $Grn^{+/-}$, and PS19 $Grn^{-/-}$) with minimal differences in genetic background, PS19 and $Grn^{-/-}$ mice were first crossed to generate PS19 $Grn^{+/-}$ and $Grn^{+/-}$ mice and then these mice were crossed. The resulting littermates with WT, $Grn^{+/-}$, $Grn^{-/-}$, PS19, PS19 $Grn^{+/-}$, and PS19 $Grn^{-/-}$ genotypes born with the ratio 1:2:1:1:2:1 were used in the experiments. For behavioral tests, both male and female mice were used with similar ratio between genotypes at 10–11 months of age. For volumetric, immunohistochemical, and biochemical analyses, only male were used at 9–12 months of age (mean ages are not significantly different between genotypes in all analyses) as previous studies have reported significantly more tau pathology and neurodegeneration in male versus female PS19 mice[36,38,42,43].

Animals were housed in groups with 2–5 animals per cage and maintained on 12 h light-dark schedule with access to food and water *ad libitum*. The ambient temperature and humidity of the mouse room were set to 21–23 °C and 50 ± 20%, respectively.

### Conduritol B epoxide (CBE) treatment in mice

GCase inhibitor CBE (MedChemExpress #HY-100944) was reconstituted at 10 mg/mL in DPBS and stored at −80 °C. Eight-months-old male WT mice with B6C3F1 background were injected intraperitoneally with 50 mg CBE per kg body weight or vehicle (DPBS) per day, for 21 days, alternating sides of injection. The dose was determined based on previous publications[88,89] and a pilot experiment.

### Human brain tissue

Samples of pre-existing human autopsy brain with no personal identifiers were obtained from Harvard Brain Tissue Resource Center, Banner Sun Health Research Institute, and Vancouver General Hospital (Supplementary Tables 1 and 2) and deemed exempt from human subjects regulation by the Yale Institutional Review Board.

### Primary neuronal culture

Primary mouse hippocampal and cortical neurons were prepared from E16-18 embryos (both male and female) as described previously[72], plated at 75,000 cells/well onto poly-D-lysine (PDL)-coated 96 well plates (CORNING #354461) and cultured in Neurobasal-A medium (GIBCO #10888-022) supplemented with B27 supplement (GIBCO #17504-044), 1 mM sodium pyruvate (GIBCO #11360-070), GlutaMAX-I (GIBCO #35050-061), and 100 U/mL penicillin and 100 mg/mL streptomycin (GIBCO #15140-122) at 37 °C in 5% $CO_2$.

### Cell culture and transfection

HEK293T cells (ATCC #CRL-11268) were maintained in DMEM (GIBCO #11965-092) supplemented with 10% fetal bovine serum (GIBCO #16000-044), 100 U/mL penicillin and 100 mg/mL streptomycin (GIBCO #15140-122) at 37 °C in 5% $CO_2$. Transient transfection was performed with Lipofectamine 2000 (Invitrogen #11668-019) according to the manufacturer's protocol.

### Mouse behavioral tests

All behavioral tests were conducted in a blinded manner. Both male and female were used at 10–11 months of age. Prior to behavioral tests, each mouse was handled for 5 min for 4 days to reduce anxiety. In all behavioral tests, mouse behavior was recorded on a JVC Everio G-series camcorder (Yokohama, Japan) and tracked by Panlab SMART software (Barcelona, Spain).

### Open field test

Open field test was performed as previously described[47] with slight modifications. Briefly, the apparatus for the test consists of a gray, 50 cm wide, 50 cm long, 40 cm high acrylamide box, with an open top. Two light sources were put by east and west sides of the box. Individual mice were placed at the corner of the box and monitored for 10 min. The bottom of box was cleaned with 70% ethanol after every trial.

### Elevated plus maze

Elevated plus maze (EPM) was performed as previously described[47]. Briefly, EPM was set at a height of 65 cm and consisted of two open white Plexiglas arms, each arm 8 cm wide × 30 cm long and two enclosed arms (5 cm × 30 cm) with black 15 cm high walls that were connected by a central platform (5 cm × 5 cm). Individual mice were placed at the central platform and monitored for 5 min. All arms were cleaned with 70% ethanol after every trial.

### Morris water maze

Morris water maze (MWM) was performed as previously described[26,72] with slight modifications. MWM was performed at room temperature and water temperature was kept at 20–25 °C. Mice were placed in an -1 meter diameter pool with a hidden, clear platform filled with water to 1 inch above the submerged platform. The hidden platform was placed in one of the 4 quadrants of the pool with the 4 drop zones directly across from the platform. A symbol, such as a plus or a cross, was placed at each of the 4 cardinal directions as possible recognition flags. The learning trial consisting of 6 trail blocks was performed two times

per day for 3 consecutive days. Mice were dropped off facing the wall at 4 different drop zones (four trials in each trial block). Each trial block was performed by alternating two mice (e.g. A, B, A, B, A, B, A, B, C, D, C, D...). Latency was measured as the time that it took for the mouse to find and spend 1 s on the hidden platform. If there was a failure to reach the platform in 60 s, the mouse was guided to the platform and allowed to rest on it for 15 s. The probe trial was performed ~24 h after finishing the learning trail. During the probe trail, the platform was removed from the pool and mice were started from a location along the pool wall diagonally opposed to the location of the platform in the learning trail and allowed to swim in the pool for 60 s. Latencies for the learning and probe trials were measured with the Panlab SMART Video Tracking Software (ver 2.5.21).

After the probe trial, the visible platform trial was performed, placing a flag (50 mL tube) atop of the hidden platform. The trial was performed by alternating two mice and mice were repeatedly placed in the pool. Time taken to reach the visible platform was manually recorded. When a consistent time for a mouse was reached, the last three times were averaged.

## Brain tissue collection and processing

Mice were euthanized with $CO_2$ and perfused with ice-cold PBS for 80 s. The brains were dissected out and the hemispheres were divided. The left hemisphere was immediately snap frozen in liquid nitrogen or the hippocampus and cortex were dissected from the left hemisphere and were individually snap frozen for biochemical analysis. The right hemisphere was fixed in 4% paraformaldehyde (PFA) (SIGMA #158127) in PBS for 1 day at 4 °C and then embedded in 10% gelatin (SIGMA #G1896) and placed in 4% PFA for another 3 days at 4 °C. Fifty μm coronal sections of the right hemisphere were cut with a Leica VT1000S Vibratome and stored in PBS with 0.05% sodium azide at 4 °C for volumetric and immunohistochemical analyses.

## Volumetric analysis and neuronal layer thickness measurement

Volumetric analysis and neuronal layer thickness measurement were performed as previously described[36] with modifications. Every sixth section (0.3 mm between sections) from bregma −1.1 mm to −3.9 mm was mounted for each mouse. After drying overnight, all sections were stained with 0.1% cresyl violet (SIGMA #C5042) for 10 min at 50 °C. The slides were then rinsed with $H_2O$ for 1 min, sequentially dehydrated in 95% ethanol for 15 min (twice) and 100% ethanol for 5 min (twice), cleared in xylene for 5 min (twice), and coverslipped in CYTOSEAL60 (ThermoFisher #8310-4). The stained sections were imaged using Aperio CS2 scanner (Leica) at 20X magnification and analyzed using ImageScope software. For 7-month-old WT and PS19 mice, the sections were imaged using VS200 system (Olympus) at 20X magnification and analyzed using OlyVIA software (V4.1). For hippocampus and posterior lateral ventricle, sections between bregma −1.1 mm and −3.9 mm were used for quantification. For piriform/entorhinal cortex, sections between bregma −2.3 mm and −3.9 mm were used for quantification. The volume was calculated using the following formula: (sum of area) × 0.3 mm.

Neuronal layer thickness measurement was performed using three sections (bregma −1.4, −1.7, and −2.0 mm) that were stained with cresyl violet as described above. The thickness was measured using the Distance Measurement tool in ImageScope software. All staining and data analysis were performed by a researcher who was blinded to the genotype.

## Thalamic cell density

Thalamic cell density was examined using two ROIs (200-μm squares) cropped from two cresyl-violet (CV)-stained coronal sections with Bregma −1.7 mm and −2.0 mm. CV-positive area and integrated CV density of the ROIs were measured using Fiji (ImageJ) software.

## Immunohistochemistry

Immunohistochemistry was performed as previously described[26] with slight modifications. For AT8/GFAP co-staining, five sections (approximately bregma 2.0, and 1.7 mm for prefrontal cortex, −1.4, −1.7, and −2.0 mm for hippocampus and piriform/entorhinal cortex) were used. For Iba1/CD68 co-staining, four sections (approximately bregma 2.0 and 1.7 mm for prefrontal cortex, −1.4 and −1.7 mm for hippocampus and piriform/entorhinal cortex) were used. For PGRN/Iba1 co-staining, two sections (approximately bregma −1.4 and −1.7 mm) were used. For AT8/p-α-syn co-staining, two sections (approximately bregma 1.9 and −2.4 mm) were used. Free-floating sections were permeabilized and blocked with 10% normal donkey serum, 0.2% Triton X-100 in PBS for 1 h at room temperature. The sections were then incubated in primary antibody in 1% normal donkey serum, 0.2% Triton X-100 in PBS overnight at room temperature. The primary antibodies that were used include: mouse PGRN (R&D #AF2557, 1:400), human PGRN (R&D #AF2420, 1:200), AT8 (Invitrogen #MN1020, 1:500), CD68 (Bio-Rad #MCA1957, 1:900), Iba1 (FUJIFILM Wako #019-19741, 1:500), GFAP (Abcam #ab7260, 1:1000), GlcCer (Glycobiotech #RAS_0011, 1:100), GCase (SIGMA #G4171, 1:150), phospho-α-synuclein (BioLegend #MMS-5091, 1:250), TDP-43 (proteintech #10782-2-AP, 1:250). BMP (Echelon #Z-PLBPA, 1:100), MC1 (gift from Dr. Peter Davies, Albert Einstein College of Medicine, 1:250). The sections were then washed three times with PBS and then incubated for 2–3 h at room temperature in either donkey anti-rabbit or donkey anti-mouse fluorescent secondary antibodies (Invitrogen Alexa Fluor 1:500) in 1% normal donkey serum, 0.2% Triton X-100 in PBS. For AT8/p-α-syn co-staining, normal goat serum was used for blocking and dilution of antibodies, and goat anti-mouse IgG2a, Alexa Fluor 568 (Invitrogen #A-21134) and goat anti-mouse IgG1, Alexa Fluor 488 (Invitrogen #A-21121) secondary antibodies were used. For AT8/p-α-syn/TDP-43 triple staining, normal goat serum was used for blocking and dilution of antibodies, and goat anti-mouse IgG2a, Alexa Fluor 488 (Invitrogen #A-21131), goat anti-rabbit IgG, Alexa Fluor 568 (Invitrogen #A11036), and goat anti-mouse IgG1, Alexa Fluor 647 (Invitrogen #A-21240) secondary antibodies were used. After incubation, the sections were washed three times with PBS. In Supplementary Fig. 18, Nuclei were stained using SYTOX Deep Red Nucleic Acid Stain (Invitrogen #S11380, 1:2000 in PBS) for 30 min at room temperature. To quench autofluorescence, sections were dipped briefly in $dH_2O$ and then incubated in 10 mM copper sulfate, 50 mM ammonium acetate, pH 5 for 15 min before dipping back into $dH_2O$ and then placed in PBS[90]. All sections were mounted onto glass slides (Superfrost plus, Fischer Scientific Company L.L.C.) and coverslipped with Vectashield antifade mounting medium with DAPI (Vector). All procedures were performed by a researcher who was blinded to the genotype.

## Immunohistochemistry with human brain samples

Formalin-fixed brains from Harvard and Banner were cut with Leica VT1000S Vibratome, and free-floating sections (50 μm) were stored in PBS with 0.05% sodium azide at 4 °C. For the free-floating sections, antigen retrieval was performed by autoclaving at 120 °C for 10 min in 10 mM sodium citrate buffer, pH 6.0 before performing the immunohistochemical procedure described above. Paraffin-embedded sections from Vancouver were deparaffinized, rehydrated, and incubated in preheated 10 mM sodium citrate buffer, pH 6.0 for 30 min at 95 °C before performing the immunohistochemical procedure described above. SUPER PAP PEN (EMS #71312) was used to perform the procedure with minimum volume of antibody solution. Humidity was kept by using a storage container (Rosti Mepal, 500 mL) with wet paper towel on the bottom during the procedure. For AT8/p-α-syn/GCase triple staining, normal goat serum was used for blocking and dilution of antibodies, and goat anti-mouse IgG2a, Alexa Fluor 488 (Invitrogen #A-21131), goat anti-rabbit IgG, Alexa Fluor 568 (Invitrogen #A11036), and goat anti-mouse IgG1, Alexa Fluor 647 (Invitrogen #A-21240)

secondary antibodies were used. To check background and bleed-through signals, a control staining without the primary antibody was performed side by side for each sample and antigen. Note that a low confocal laser power that does not detect autofluorescent lipofuscin signal in any section without primary antibody was used. In some cases, the low laser power could not detect normal GlcCer signal although it did detect GlcCer accumulated in the NFTs.

## Imaging

For AT8/GFAP, Iba1/CD68, and PGRN/Iba1 staining, all images of hippocampus, piriform/entorhinal cortex, and prefrontal cotex were taken using a Zeiss AxioImager Z1 fluorescent microscopy with a 5X objective lens. For AT8/PGRN, AT8/GlcCer, AT8/p-α-syn, and AT8 or p-α-syn/GCase staining, images were taken using Zeiss LSM800 confocal microscopy with 10X, 20X, 40X or 63X objective lens. For AT8/p-α-syn co-staining, three confocal images with higher number of AT8 and p-α-syn double positive inclusions were taken from each region (prefrontal cortex, CA2 + 3, amygdala, piriform cortex) with 20X objective lens. For p-α-syn/GCase images in Fig. 5 the maximal intensity projection function was used with z-stack confocal images.

## Nuclei isolation from the mouse hippocampus

Nuclei were isolated with Nuclei EZ Prep Kit (SIGMA #NUC-101) as previously reported[91] with modifications. Frozen hippocampus was homogenized using a glass dounce tissue grinder (SIGMA #D8938) 25 times with pastel A and 25 times with pastel B in 2 mL of ice-cold lysis buffer (Nuclei EZ lysis buffer with Complete Mini EDTA-free and Protector RNase Inhibitor (ROCHE)) and transferred to 15 mL tube with additional 2 mL of ice-cold lysis buffer, and then vortexed and incubated on ice for 5 min. Nuclei were centrifuged at $500 \times g$ for 5 min at 4 °C, washed with 4 mL of ice-cold lysis buffer, briefly vortexed and incubated on ice for 5 min. After centrifugation at $500 \times g$ for 5 min at 4 °C, the nuclei were washed with 4 mL of ice-cold nuclei suspension buffer (0.5 mg/mL BSA (ThermoFisher #AM2616) and 0.1% Protector RNase inhibitor in DPBS (CORNING #21-031-CV)) once, resuspended in 1 mL of nuclei suspension buffer, filtered through a 35 μm cell strainer (FALCON #352235) twice, and counted with trypan blue. The samples were diluted to a final concentration of ~1000 nuclei per μL and used for subsequent analyses. Nuclei of three samples ((WT, $Grn^{+/-}$, and $Grn^{-/-}$) or (PS19, PS19 $Grn^{+/-}$, and PS19 $Grn^{-/-}$)) were isolated and submitted for the sequencing at a time and the process was repeated six times to obtain three biological replicates for each genotype.

## Single-nucleus RNA sequencing (snRNA-seq)

Libraries were prepared using Chromium Single Cell 3′ GEM, Library & Gel Bead Kits v3 (10X Genomics #PN-1000075) according to the manufacturer's protocol. The libraries underwent sequencing with a sequence depth of 200 million reads per sample using NovaSeq instrument (Ilumina) according to 10X Genomics recommendations. Sample demultiplexing, alignment, and gene counting were performed using Cell Ranger software (10X Genomics, version 3.0.2). Gene counts were obtained by aligning reads to the mm10 reference genome.

## Quality control and pre-processing of snRNA-seq datasets

Subsequent data processing was performed using the Seurat package (version 3.1.4)[92]. In all 18 datasets, genes expressed in <3 nuclei and nuclei with >5% mitochondrial counts were removed. Nuclei that express <200 or > 2500 genes were also removed for downstream analysis.

Doublets were detected and removed using the DoubletFinder package (version 2.0.3) prior to dataset integration[93]. All datasets were pre-processed with SCTransform[94], RunPCA (principal component analysis), and RunUMAP (Uniform Manifold Approximation and Projection) functions using the Seurat package. The proportion of artificial doublets (pN) was set to 0.25 (authors' recommendation) in all datasets, and the neighborhood size parameter (pK) was optimized for each dataset with the mean-variance-normalized bimodality coefficient (BCMVN) maximization as described previously[93]. The doublet rate was set to 0.075 based on 10x Genomics protocol without homotypic doublet adjustment to remove all potential heterotypic doublets.

## Integration, visualization, and clustering of snRNA-seq datasets

In order to reduce potential batch effects, "Reference-based" integration was performed with 18 SCTransform-normalized and doublet-removed datasets using the Seurat package as shown in online vignettes (https://satijalab.org/seurat/v3.1/integration.html).

A reference was selected from each genotype and total 6 references were used for the integration. For visualization and clustering, the integrated dataset was processed with RunPCA and RunUMAP (with the top 50 PCs) functions, followed by FindNeighbors and FindClusters (with a resolution of 0.3) functions of the Seurat package. These analyses identified 33 pre-clusters in the integrated dataset. Cell types were then assessed for each pre-cluster based on the expression of known maker genes and previously identified signatures using the hippocampal samples[50,91,95]. Pre-clusters that were defined to be the same cell type were grouped and a pre-cluster with nuclei with low UMI counts was removed for downstream analysis. Due to the low number of nuclei, cell type could not be defined in one pre-cluster (cluster #13). The resulting 12 cell type clusters were used for differential expression and enrichment analyses.

## Differential expression and gene overlap of snRNA-seq dataset

Differential expression analysis after dataset integration was performed on the "RNA" assay after normalization, which is suggested in Satija lab's website (https://satijalab.org/seurat/) using the Wilcoxon-rank-sum test of the Seurat package. The adjusted $p$-value was calculated using Bonferroni correction. Genes with |log(FC)| > 0.25 and adjusted $p$-value < 0.01 were selected as differentially expressed. To estimate the significance of overlap of two gene lists, overlapping $p$-value was calculated using the Fisher's exact test of the GeneOverlap package (version 1.23.0) (https://bioconductor.org/packages/release/bioc/html/GeneOverlap.html). The list of all genes expressed in a given cell type was used as the background. The lists of DAM- and DAA-upregulated genes were taken from Supplementary Table 2 in Keren-Shaul et al.[49] and from Supplementary Table 2 in Habib et al.[50], respectively.

## Enrichment analysis

Enrichment analysis was performed using Metascape[96] with default setting, which includes ontology catalogs of KEGG Pathway, GO Biological Processes, Reactome Gene Sets, and CORUM. Terms with −log10(FDR) > 2 were selected as significantly enriched. The heatmap of −log10(FDR) of all enriched terms was generated using Morpheus (https://software.broadinstitute.org/morpheus/). Hierarchal clustering was performed with 1-Pearson correction and average agglomeration method.

## Brain extraction

Sequential fractionation using RAB, RIPA, 70% formic acid (FA) buffer was performed as described previously[36] with modifications. Briefly, cortices from left hemisphere were weighted and homogenized using a dounce homogenizer (DWK Life Science #357422) for 25 strokes in 10-fold volume of ice-cold RAB buffer (100 mM MES, 1 mM EGTA, 0.5 mM $MgSO_4$, 750 mM NaCl, 20 mM NaF, 1 mM $Na_3VO_4$, pH 7.0, supplemented with PhosSTOP, cOmplete-mini (Roche)). After ultra-centrifugation for 20 min at $50,000 \times g$ at 4 °C, the supernatant was

collected and saved as the RAB-soluble fraction and the pellet was resuspended in 10-fold volume of ice-cold RIRA buffer (25 mM Tris, 150 mM NaCl, 1% NP40, 0.5% deoxycholic acid, 0.1% SDS, 20 mM NaF, 1 mM Na$_3$VO$_4$, pH 8.0, supplemented with PhosSTOP, cOmplete-mini (Roche)) and nutated for 30 min at 4 °C. After ultracentrifugation for 20 min at 50,000 × g at 4 °C, the supernatant was collected and saved as the RIPA-soluble fraction and the pellet was resuspended in 3-fold volume of ice-cold 70% FA and nutated for 30 min at 4 °C. After ultracentrifugation for 20 min at 50,000 × g at 4 °C, the supernatant was collected and saved as the FA-soluble fraction. All fractions were stored at −80 °C.

### Co-immunoprecipitation assay
Co-immunoprecipitation (co-IP) assay was performed as previously reported[47] with modifications. Plasmids encoding non-tagged human GBA were purchased from OriGene Technologies, Inc. FLAG-PGRN was described previously[97]. One day after transfection, HEK293T cells were harvested and lysed with ice-cold 50 mM Tris-HCl pH 7.5, 150 mM NaCl, 0.5% Triton X-100 supplemented with PhosSTOP, cOmplete-mini (Roche). After centrifugation for 30 min at 100,000 × g at 4 °C, the supernatant was incubated with anti-FLAG M2 Affinity Gel (SIGMA #A2220) overnight at 4 °C. The immunoprecipitates were washed five times with ice-cold 50 mM Tris-HCl pH 7.5, 150 mM NaCl, 0.1% Triton X-100 and boiled with 2 x Laemmli sample buffer (Bio-Rad) with βME.

### Immunoblot
Immunoblot was performed as previously reported[26], with slight modifications. For the FA-soluble fraction, the samples were diluted with 16-fold volume of 1 M Tris. The protein samples were electrophoresed using precast 4–20% Tris-glycine gels (Bio-Rad) and transferred with an iBlot 2 Transfer Device onto nitrocellulose membranes (Invitrogen IB23001). The membranes were incubated in blocking buffer (Rockland MB-070) for 1 h at room temperature and then incubated overnight at 4 °C in blocking buffer with primary antibodies: GCase (SIGMA #G4171, 1:1000) and FLAG (SIGMA #F7425, 1:1000), tau (DAKO #A0024, 1:5000), HT7 (Invitrogen #MN1000, 1:1000), AT8 (Invitrogen #MN1020, 1:1000), PHF1 (gift from Dr. Peter Davies, Albert Einstein College of Medicine, Bronx, NY, 1:1000), phospho-tau (Ser199, Ser202) (Invitrogen #44-768 G, 1:1000), phospho-tau (Ser356) (Invitrogen #44-751 G, 1:1000), β-actin (Cell Signaling Technology #3700, 1:2000), LIMPII (Novus Biologicals #NB400-129, 1:1000), cathepsin B (R&D #AF965, 1:500), cathepsin D (R&D #AF1029, 1:500), Hsp60 (Cell signaling #4870, 1:1000), calreticulin (Novus Biologicals #681233, 1:1000), Rab5 (Cell signaling #3547, 1:1000), Sap C (Santa Cruz #sc-347118, 1:200), mouse PGRN (R&D #AF2557, 1:400), The next day, the membranes were washed three times with TBST for 3 min and incubated in secondary antibodies (Li-Cor, IR Dye 680 or 800, all 1:10,000) for 1 h at room temperature. After washing three times with TBST for 3 min, proteins were visualized with an Odyssey Infrared imaging system (Li-Cor). The immunoreactive bands were quantified using ImageJ software.

### GCase activity assay
GCase activity assay was performed as previously reported[56,57] with modifications. Cortices from the left hemispheres were homogenized in 10-fold volume of ice-cold citrate-phosphate buffer pH 5.2, 1% Triton X-100 with PhosSTOP and cOmplete-mini (Roche) using a dounce homogenizer (DWK Life Science #357422) for 20 strokes and nutated for 30 min at 4 °C. After ultracentrifugation for 30 min at 100,000 × g at 4 °C, the supernatants were collected and stored at −80 °C. On the day of the experiment, the supernatants were thawed and 5 µL of each sample was pre-incubated in a black 96-well plate (Costar #3916) for 30 min on ice with 10 µL of 2.5 mM Conduritol B Epoxide (CBE) in GCase activity assay buffer (citrate-phosphate

buffer pH 5.2, 1% BSA (SIGMA #A9647), 4 mM sodium taurocholate (SIGMA #T4009), 150 µM EDTA) and then mixed with 10 µL of 10 mM 4-Methylumbelliferyl β-D-glucopyranoside (SIGMA #M3633) in GCase activity assay buffer for a total reaction volume of 25 µL. All assays were performed in duplicate. Reactions were performed on ice for 30 min with brief shaking every 5 min and then at 37 °C for 60 min. Reactions were stopped by adding 200 µL of precooled 0.5 M glycine-NaOH, pH 10.6 stop solution on ice. Measurements were taken with the Victor 3 plate reader (PerkinElmer) or with VICTOR Nivo® multimode plate reader (PerkinElmer) at excitation of 355 nm and emission of 460 nm. Protein concentrations were determined using Protein Assay Dye Reagent Concentrate (BIO-RAD #5000006) and used for normalization.

### Preparation of lysosome-enriched fractions from mouse cortex
Lysosome-enriched fractions were isolated using Lysosome Isolation Kit (SIGMA #LYSISO1). Freshly dissected left cerebral cortex from WT or Grn$^{−/−}$ mice was homogenized in 1.6 mL of 1x Extraction Buffer supplemented with PhosSTOP, cOmplete-mini (Roche) using a dounce homogenizer (DWK Life Science #357422) for 25 strokes on ice. The homogenate was centrifuged at 1000 × g for 10 min at 4 °C to remove nuclei and other cell debris. The supernatant was transferred into a new tube and centrifuged at 18,000 × g for 30 min at 4 °C to collect the Crude Lysosomal Fraction (CLF). The CLF pellet was resuspended in 1x Extraction Buffer and diluted to prepare 16% Diluted Optiprep Fraction. After preparing a discontinuous density gradient with 27% Optiprep solution at the bottom and 8% Optiprep at the top of the tube (Beckman #331374), the tube was ultracentrifuged for 4 h at 150,000 × g in an SW 40 Ti rotor. Four bands (the fractions #1-4) floating in the gradient were collected using 1 mL syringes with 18 G x 1 needles (1 mL for each). Protein concentration of each fraction was determined using Bio-Rad Protein Assay Dye Reagent Concentrate (BIO-RAD #5000006). β-N-acetylglucosaminidase activity of each fraction was measured using β-N-acetylglucosaminidase assay kit (SIGMA #CS0780).

For GCase activity assay and immunoblots for GCase, Sap C, and PGRN, 0.9 mL of the fractions #2 or #3 was mixed with 2.5 mL of 1x Extraction Buffer supplemented with PhosSTOP, cOmplete-mini (Roche) to dilute Optiprep and split into two 2 mL tubes. The tubes were then centrifuged at 18,000 × g for 30 min at 4 °C to precipitate lysosomes. After removing the supernatant, the pellet was surface-washed with 1 mL/tube of ice-cold citrate-phosphate buffer pH 5.2 with PhosSTOP and cOmplete-mini (Roche) and centrifuged at 18,000 × g for 30 min at 4 °C. The lysosome pellet was solubilized with 100 µL (for the fraction #2) or 50 µL (for the fraction #3) of ice-cold citrate-phosphate buffer pH 5.2, 1% Triton with PhosSTOP and cOmplete-mini (Roche) for 30 min at 4 °C. After ultracentrifugation at 100,000 x g for 30 min at 4 °C, the supernatant was collected and stored at −80 °C until use. Right cortex was also dissected out and immediately snap frozen. The frozen right cortices were then processed the next day with ice-cold citrate-phosphate buffer pH 5.2, 1% Triton with PhosSTOP and cOmplete-mini (Roche) as described above. Protein concentrations of the lysosome-enriched fractions were determined using Pierce™ BCA protein assay kit (ThermoFisher SCIENTIFIC #23225).

### Lipidomics (for GalCer and GlcCer)
Separation of GlcCer and GalCer species was performed at the Lipidomics Shared Resource at Medical University of South Carolina. Cortices were weighed and homogenized in 10-fold volume of ice-cold Tissue Homogenization Buffer (0.25 M sucrose, 25 mM KCl, 0.5 mM EDTA, 50 mM Tris-HCl, pH 7.4). Lipids were extracted from homogenate containing 1 mg of protein, and levels of GlcCer and GalCer species were measured with supercritical fluid chromatography-tandem mass spectrometry analysis.

## Lipidomics (for general lipid panels, BMP, and GM1)

Multidimensional mass spectrometry-based shotgun lipidomic analysis was performed as previously reported[98]. Briefly, cerebral cortex from right hemisphere was homogenized in 0.1X PBS and the cortical homogenate containing 1.0 mg of protein, which was determined with a Pierce™ BCA protein assay kit (ThermoFisher SCIENTIFIC #23225), was accurately transferred to a disposable glass culture test tube. A premixture of lipid internal standards was added prior to conducting lipid extraction for quantification of the targeted lipid species. The following internal standards purchased from Avanti Polar Lipids were used: 26.25 nmol of PC(14:1/14:1), 23.58 nmol of PE(16:1/16:1), 3.08 nmol of PG(15:0/15:0), 19.724 nmol of PS(14:0/14:0), 0.482 nmol of PA(14:0/14:0), 8.008 nmol of GalCer(d18:1/15:0), 2.732 nmol of SM(d18:1/12:0), 1.053 nmol of Cer(d18:1/17:0), 3.08 nmol of ST(d18:0/16:0), 1.257 nmol of LPE(14:0), 1.232 nmol of LPC(17:0), 1.331 nmol of CL(14:0/14:0/14:0/14:0). Lipid extraction was performed using a modified Bligh and Dyer procedure[98], and each lipid extract was reconstituted in chloroform:methanol (1:1, $v{:}v$) at a volume of 400 μL/mg protein. Phosphoethanolamine (PE) was derivatized as described previously[99,100] before lipidomic analysis. Quantitative analysis of isomeric bis(monoacylglycerol)phosphate (BMP) and phosphatidylglycerol (PG) was performed as reported[101]. Gangliosides (GM1) were extracted using the mixture of chloroform, methanol, and water (1:2:0.65). Gangliosides in water phase were enriched using C18 SPE. After washing with methanol/water (1:1), the gangliosides were eluted with methanol, dried, and reconstituted in methanol for lipidomic analysis. GM1(d18:1/C18:0)-d3 was used for the internal standard and 1.6 nmol/mg was added to each samples prior to conducting lipid extraction.

For shotgun lipidomics, lipid extract was further diluted to a final concentration of ~500 fmol total lipids per μL. Mass spectrometric analysis was performed on a triple quadrupole mass spectrometer (TSQ Altis, Thermo Fisher Scientific, San Jose, CA) and a Q Exactive mass spectrometer (Thermo Scientific, San Jose, CA), both of which were equipped with an automated nanospray device (TriVersa NanoMate, Advion Bioscience Ltd., Ithaca, NY) as previously described[102]. The triple quadrupole and orbitrap systems were run separately for mutual validation and identification. Identification and quantification of lipid species were performed using an automated software program[103,104]. Data processing (e.g., ion peak selection, baseline correction, data transfer, peak intensity comparison and quantitation) was performed as previously described[104]. The result was normalized to the protein content (nmol lipid/mg protein).

## AD-tau seeding assay

AD-tau seeding assay was performed as previously reported[72,73,75], with slight modifications. Tau fibrils were extracted from human AD brains as previously reported[75]. Brain A and B were used for PGRN-deficient and CBE-treated neurons, respectively. Primary cultured neurons were prepared from mouse E16-18 WT (C57BL6) or $Grn^{-/-}$ embros (both male and female). Dissociated neurons from the hippocampus and cortex were plated at 75,000 cells/well onto poly-D-lysine-coated 96-well plates (Corning #354461). Experiments were performed in triplicate. At DIV7, the tau fibrils extracted from human AD brains (AD-tau) were seeded onto neurons. Conduritol B Epoxide (CBE) or vehicle (ultrapure water) was also treated with AD-tau at DIV7. At DIV21, the neurons were fixed with ice-cold methanol for 30 min on ice, and blocked with 10% normal donkey serum, 0.2% Triton X-100 in PBS for 1 h at room temperature. Neurons were then incubated overnight at 4 °C in 1% normal donkey serum, 0.2% Triton X-100 in PBS with primary antibodies: Mouse tau (T49) (SIGMA, #MABN827, 1:500) and MAP2 (Cell signaling, #4542, 1:150). The next day, neurons were washed with PBS two times and incubated for 1 h at room temperature with secondary antibodies anti-mouse IgG

Alexa Fluor 488 (Invitrogen, 1:500) and anti-rabbit IgG Alexa Fluor 568 (Invitrogen, 1:500) and DAPI (0.5 μg/mL) diluted in 1% normal donkey serum, 0.2% Triton X-100 in PBS. Finally, neurons were washed four times with PBS and imaged automatically and unbiasedly (4 images/well) using ImageXpress Micro XLS with 20X objective lens (Molecular Devices).

## Preparation of lipid dispersions

Lipid dispersions were prepared as previously reported[64] with modifications. Brain phosphatidylchorine (PC) (Avanti Polar Lipids #840053 P), Glucosyl (β) Ceramide (GlcCer) (Avanti Polar Lipids #860547 P), Glucosyl (β) Sphingosine (Avanti Polar Lipids #860535 P), and Galactosyl (β) Ceramide (Avanti Polar Lipids #860844 P) were dissolved in chloroform at 10 mg/mL. PC was aliquoted and stored at −20 °C in glass vials with nitrogen gas overlay. The other lipids was aliquoted and used immediately or stored at −20 °C in glass vials with nitrogen gas overlay and used the next day. For lipid dispersions, lipids were mixed at 1:3 molar ratio and vortexed thoroughly in glass tubes before drying under a nitrogen stream. The lipid film was hydrated in fibrillization buffer (10 mM Hepes, 100 mM NaCl, 1 mM DTT, pH 7.4), vortexed thoroughly, and sonicated in a water bath for 15 min. After transferring to polypropylene tube, the samples were bath sonicated for 15 min, put through 4 freeze/thaw cycles, and bath sonicated for another 30 min. The lipid dispersions were vortexed and then used immediately.

## Thioflavin T assay

A final concentration of 6.6 μM recombinant human P301S tau 2N4R (StressMarq #SPR-327) was mixed with 4.5 μM heparin (SIGMA #H3393) and 25 μM Thioflavin T (ThT; SIGMA #T3516) in fibrillization buffer (10 mM Hepes, 100 mM NaCl, 1 mM DTT, pH 7.4), and then with 1.2 μM recombinant human GCase (R&D system #7410-GHB), which was dissolved in 50 mM sodium citrate pH 5.5, BSA, or lipid dispersions described above (787 μM monomer equivalent). Forty μL volumes were added to black 384-well plates (Greiner 781900). All experiments were performed in triplicate. The plate was incubated for 255 min at 37 °C in the Victor 3 plate reader (PerkinElmer). ThT fluorescence (at excitation of 440/8 nm and emission of 486/10 nm) was measured every 5 min with 15 s shaking in a double-orbital fashion before the measurement.

## Transmission EM

Three micro liters of diluted samples after ThT assay were placed on carbon-coated TEM grids (Electron Microscopy Sciences CF300-CU) that were glow discharged for 45 seconds. Excess material was blotted with filter paper and the grid was washed and stained in 2% uranyl acetate for 1 minute before blotting and drying. The grids were imaged either in an FEI 120 kV Tecnai TEM with an Ultrascan 4000 CCD Camera or an FEI 200 kV TF20 with an AMT Nanosprint1200 CMOS Camera. At least two independent image sessions were conducted for each sample to confirm the reproducibility.

## Image quantification of immunohistochemistry

All quantitative analyses of images taken with Zeiss AxioImager Z1 fluorescent microscopy were done using ImageJ (National Institutes of Health). For Iba1/CD68 co-staining, after background subtraction (Rolling ball radius: 200 pixels), all images were uniformly thresholded and binarized. Iba1 and CD68 areas were calculated using the "analyze particles" of ImageJ. The mean of two or three sections was used to represent each mouse. For AT8/p-α-syn co-staining, three images taken from each region (prefrontal cortex, CA2 + 3, amygdala, piriform cortex) were used to manually count AT8 and p-α-syn positive inclusions. All imaging and data analyses were performed by a researcher who was blinded to the genotype.

## Image quantification of AD-tau seeding assay

All quantitative analyses were performed as previously reported[72] using macro of ImageJ (National Institutes of Health). For the graphs with data points of "Experiments", the mean of 12 images (3 wells × 4 images) from each sample was used to represent each experiment. Data analyses were performed by a researcher who was blinded to the group or genotype.

## Data analysis of GCase activity assay

After subtraction of background obtained from wells with GCase activity buffer containing only 4 mM 4-Methylumbelliferyl β-D-gluco-pyranoside, the mean values of duplicate wells were used to represent each sample. The values from samples without CBE were used as total GCase activity and normalized to the mean value of WT or PS19 samples. For GBA1-specific activity, values from samples with CBE were subtracted from those from sample without CBE and then the subtracted values were normalized to the mean values of WT or PS19 samples.

## Statistics and reproducibility

One-way ANOVA, two-way ANOVA, or Kruskal-Wallis test with multiple comparisons tests (for >3 groups) and two-tailed unpaired $t$-test (for 2 groups) were performed as specified in the figure legends using GraphPad Prism 9. All data were shown as mean ± SEM. Data were considered to be significant if $p < 0.05$. Sample size was determined based on previous studies[26,36,39,42,47,56,64,70,72,75,105]. No statistical method was used to predetermine sample size. Assumption of whether or not data follow normal distribution was based on previous studies[26,36,47]. In EPM test, outliers detected by ROUT (Q = 1%) were excluded because they were deemed to fail to be adapted to the test. In MWM, mice that took more than 30 s to reach the visible platform were excluded because they are deemed to have eye problems rather than memory problems or lack the motivation[72,105]. No data were excluded in the other analyses. Primary cultured neurons for AD-tau seeding assay and tau protein for ThT assay were randomly allocated into experimental groups. All behavioral tests, immunohistochemical, lipidomic, and EM data collection and analyses were performed by investigators who were blinded to the genotypes. Investigators were also blinded to the genotypes during data collection for snRNA-seq experiments. Images of AD-tau seeding assay were automatically and unbiasedly collected using ImageXpress and the analysis was performed by an investigator blinded to the samples. Investigators were not blinded during nuclei isolation and lysosomal fractionation to minimize batch effects because the nuclei or lysosome preparation was repeated several days to complete all samples. Investigators were not blinded during the snRNA-seq analysis using the Seurat package, because the data were analyzed based on the genotypes. In immunoblot and co-IP assay, investigators were unblinded to determine the loading order of the samples. Investigators were not blinded to the sample groups in GCase activity assay and ThT assay because the data were automatically collected with Victor 3 or VICOTR Nivo plate reader. For mouse experiments, all the biological replicates to support reproducibility are described in each figure legend. MWM and EPM tests were performed with two independent cohorts and similar results were obtained. Immunohistochemical findings were verified by repeating the staining at least once with different antigen combinations in different experiments. Similar results were obtained in the replication staining experiments. Co-IP and GCase activity assays were performed to replicate previous publications[56–58]. An increase in GlcCer levels in $Grn^{-/-}$ mice was confirmed by both lipidomics and immunohistochemistry. At least three independent experiments were performed in co-IP, AD-tau seeding, and ThT assays. At least two independent EM image sessions were conducted using independently prepared samples to confirm reproducibility. All attempts at replication were successful.

## Reporting summary

Further information on research design is available in the Nature Portfolio Reporting Summary linked to this article.

## Data availability

snRNA-seq raw data have been deposited at GEO under accession code GSE180672. Mouse reference 3.0.0, mm10 was used (https://www. 10xgenomics.com/support/software/cell-ranger/downloads/cr-ref-build-steps). Full results of Glc/GalCer analysis and lipidomic analysis for general lipid panels, BMP, and GM1 are provided in the Supplementary Data 1 and Data 2, respectively. Source data are provided with this paper.

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

## Acknowledgements

We thank Dr. Marc Llaguno and Dr. Kimberley Gibson of the Yale School of Medicine Electron Microscopy Core Facility for expert technical assistance. We also thank Dr. Peter Davies (Albert Einstein College of Medicine), Dr. Philippe Marambaud and Dr. Jeremy Koppel (The Feinstein Institutes for Medical Research) for providing MC1 antibody. We are grateful to the Banner Sun Health Research Institute Brain and Body Donation Program of Sun City, Arizona for the provision of human brain tissues. The Brain and Body Donation Program has been supported by the National Institute of Neurological Disorders and Stroke (U24 NS072026 National Brain and Tissue Resource for Parkinson's Disease and Related Disorders), the National Institute on Aging (P30 AG19610 Arizona Alzheimer's Disease Core Center), the Arizona Department of Health Services (contract 211002, Arizona Alzheimer's Research Center), the Arizona Biomedical Research Commission (contracts 4001, 0011, 05-901 and 1001 to the Arizona Parkinson's Disease Consortium) and the Michael J. Fox Foundation for Parkinson's Research. Human brain tissues were also obtained from the NIH NeuroBioBank. S.H.N. received a PhD fellowship from Boehringer Ingelheim Fonds. This work was supported in part by the Lipidomics Shared Resource, Hollings Cancer Center, Medical University of South Carolina (P30 CA138313 and P30 GM103339). This work was also supported by National Institute on Aging of the National Institutes of Health under grant number RF1AG061872 to X.H. and grant number R01AG034924 and R01AG066165 to S.M.S. The content is solely the responsibility of the authors and does not necessarily represent the official views of the National Institutes of Health.

## Author contributions

H.T. and S.M.S. planned and oversaw all aspects of the study. H.T. performed and analyzed most of the experiments. S.B. performed and analyzed PGRN/Iba1 immunostaining and analyzed AD-tau seeding assay. S.H.N. performed extraction of AD-tau. H.Y. and X.H. performed lipidomic analysis for general lipid panels, BMP, and GM1. M.T.C helped process mouse brain tissues and performed some immunostaining. G.W. performed sample preparation of snRNA-seq and provided instructions for the snRNA-seq analysis. I.R.M. provided human brain samples. H.T. and S.M.S. wrote the manuscript with input and revisions from all authors.

## Competing interests

The authors declare no competing interests.
