## [Peer Review File · Nature Communications]

Reduced progranulin increases tau and α -synuclein inclusions and alters mouse tauopathy phenotypes via glucocerebrosidaseEditorial Note: Parts of this Peer Review File have been redacted as indicated to remove third-party material where no permission to publish has been obtained.

Reviewers' Comments:

Reviewer #1 (Remarks to the Author):

In this manuscript, Takahashi et al demonstrate that reducing progranulin (PGRN) levels in PS19 tau mutant mice leads to enhanced a-synuclein and tau proteinopathies, with the latter typified by an elevation of GluCer (a GCCase substrate) -positive inclusions. Surprisingly, PGRN reduction ameliorated spatial memory defects and hippocampal degeneration in PS19 mice, while worsening alpha-synuclein pathology and premature mortality. The authors confirm that PGRN reduction impairs GCCase activity and promotes the accumulation of the GCCase substrates GlcCer and GlcSph, in agreement with a recent study (Logan et al. Cell 2021). Moreover, the authors show that GlcCer can directly promote tau aggregation, similar to what has previously been reported for a-synuclein, suggesting that the PGRN-GCCase axis may be a commonly dysregulated pathway across neurodegenerative disease, and perhaps hinting at a mechanism through which impairments in PGRN levels can increase disease risk across diverse neurodegenerative diseases (AD, PD, FTD), as has been highlighted by recent GWAS studies. As is such, the findings in this manuscript should be of especially high interest to the AD, PD and FTD fields.

While I am generally supportive of publication of this manuscript, there are a few outstanding questions that I believe could be addressed to increase the overall impact of the findings:

- While the finding that PGRN reduction leads to GlcCer-positive tau inclusions is interesting, the major weakness of the paper is the questionable disease relevance of PGRN reduction on the mutant tau mouse background. Indeed, it has been conclusively established that FTD patients with PGRN mutations do not present with tau pathology, but rather TDP-43 pathology.
- How specific is the GCCase substrate accumulation in the Grn KO x PS19 brain? Although the authors did not observe any effects on GalCer, the lipids measured still represent a narrow fraction of the total glycosphingolipids. For instance what happens to sphingomyelin, ceramide, sulfatide, and ganglioside levels? Similarly, what happens to levels of the glycerophospholipid BMP, recently demonstrated by two groups to be deficient in the brains of Grn KO mice (Logan et al. Cell 2021, Boland et al. bioRxiv 2022)?
- Similar to the question above, how specific is the effect on in vitro tau aggregation seeding to GlcCer?
- Along these lines, does GlcCer decorating tau aggregates indicate an active process, or do other potentially co-accumulating lipids similarly label insoluble tau? Is it possible to biochemically purify the tau aggregates and probe their lipid composition in an unbiased manner?
- What effect, if any, does reducing progranulin levels in vitro have on the AD-tau seeding assay?
- While understanding that the hippocampal region was chosen to assess neurodegeneration and for the sc-RNAseq study with tau pathology in mind, what happens in regions known to be preferentially

affected in the Grn KO mice alone (e.g. frontal cortex or thalamus)?

- Though understandably beyond the scope of this manuscript, do PS19 mice crossed with GBA mutant (e.g. D409V), null or haploinsufficient mice show similarly enhanced tau and alpha-synuclein pathology?

Reviewer #2 (Remarks to the Author):

The manuscript examines the role of Grn in tauopathies using PS19 mouse models and human brain. The general topic is of critical importance, and while some of the data are interesting, there are some significant weaknesses throughout the paper that dramatically reduce enthusiasm for publication. The data showing that Grn reduction can rescue hippocampal volume is interesting, but the correlation with increase tau tangles could be stronger and not completely supported by the data. Primary data is not shown in this readout, and there is only a single assay used (AT8 IHC) to make this conclusion. There are essential controls missing and primary data images omitted throughout. There is a lack of information on how the mouse crosses were done, if the background is mixed, additional technical issues in many of the assays, and the statistical tests change throughout without reason. The data also contradicts well established data from human disease that clearly links reduced Grn and GCase to increase neurotoxicity. While this can be of great interest, it is unfortunately done in a confusing way and without a clear explanation. Specific points are below.

1) The hippocampal and memory rescue by Grn $-/-$ is potentially interesting overall, and a strength of the study is the high n number used in the design for most of the studies. However, the authors should address the potential of genetic background differences contributing to the phenotype of memory rescue in the PS19+ / Grn $-/-$ mice (Fig. 1E-I). It is essential to confirm that the rescue doesn't occur from some unknown factor that results from crossing the PS19 mice into the B6 -RIKEN strain. The studies should have been done on a pure background by several rounds of backcrossing, to eliminate confounding genetic factors, however these seem to be all hybrids. It also wasn't clear if littermates were used for comparisons, since parentals will likely have a different background mix compared to progeny. This should be clarified in the methods section. Perhaps a SNP panel could be done to determine if the PS19 group has an identical background to the PS19 Grn $+/-$ and PS19 Grn $-/-$ groups that appear in the study.

2) In figure 1 c,d, the changes in these behavioral tests seem minor – are there differences between the PS19 and PS19 Grn $-/-$ groups? Justification for a Dunnett test as opposed to a Tukey test is required. If the study was designed to determine the role Grn in PS19 mice, then the PS19 and PS19 Grn $-/-$ groups should be directly compared through an ANOVA-Tukey test. It's also inconsistent with figure 3.

3) If the authors wish to correlate pathological changes with behavior, the genders of both studies should match. It isn't clear why both females and males were used for behavior analysis, if the males exhibit more pronounced pathology. Adding in a female group with reduced pathology and

neurodegeneration confounds the behavioral data and makes it very difficult to determine the basis of the behavioral changes.

4) Previous studies have shown that microgliosis and neurodegenerative changes occur prior to 7 months of age (Yoshiyama et al 2007); therefore, it would be important to confirm that no cell loss or gliosis occurs at 7 months of age for Fig 2 by IHC or other methods.

5) Fig 3 b, f, g— please show the primary data (IHC images) of all three genotypes, showing all 4 subtypes of pathology. It is not possible to tell from the graph what is significantly different from what.

6) Higher resolution confocal or electron microscopy is required to confirm the conclusion that PGRN reduction induces tangle like pathology in figure 3. Thioflavin staining may also help here.

7) What is the significance of the increase CD68 staining in the ps19 Grn^{-/-} group shown in Fig 4A? Could it play a role in rescue phenotype observed in Fig 1, or by redistributing tau pathology from Type 4 into Types 2 or 3?

8) In the text, the GC / PGRN pull down is Ext. fig 6 (not 5).

9) The GCase activity decrease shown in extended fig 6D and Fig 5b,c, is mild. Typically, for the substrate to accumulate as shown in Fig 5, a decrease in activity greater than 50% is required. The GCase activity assay used here doesn't report on problems that could occur from GCase trafficking or environment. The activity assay buffer includes taurocholate, which presumably mimics the function of Sap C, and therefore the enzyme is artificially activated in vitro. If Sap C loss was involved in reduced GCase function in vivo, this assay would not detect it. The same is true for trafficking deficit. The authors should consider a lysosomal fraction to determine GCase content within lysosomes, with and without detergent.

10) Figure 5A requires an image of the all of the genotypes quantified in the graph. A counter stain or nuclear stain is required, otherwise it is difficult to judge the comparability of the sections.

11) Regarding this statement “there was a linear trend in the increase of total GlcCer levels between WT, Grn^{+/-} and Grn^{-/-} genotypes, although unexpectedly we did not observe the trend between PS19, PS19 Grn^{+/-} and PS19 Grn^{-/-} genotypes (Fig. 5g,h and Extended Data Fig. 7).”

The study may be powered sufficiently to determine differences between groups, but more mice are required to draw conclusions about trends. The lipid levels in brain can be variable and typically require around 10-12 mice per group. Suggestion is to revise the text here or repeat the study with the appropriate number of mice.

12) In fig 6, does GluCer colocalize with AT8-a-Syn inclusions? Please explain why the PS19 Grn^{+/-} and ^{-/-} groups are combined in the quantification. In every other data set they are separate. They should be separated here because they are two different groups of mice. As in other IHC images, some counterstain is required to determine the comparability of the sections between the genotypes.

13) Synuclein pathology should be examined in Grn^{-/-} mice (without p301 tau).

- 14) In fig 6D and extended fig 8b, the specificity of the GCCase antibody should be demonstrated. No commercially available GCCase antibodies are known to work at all in fixed tissue/cultures.
- 15) The data in Fig 7 A, it appears that the tau signal is outside of neurons. A magnified image would help, along with a nuclei stain. For AD-derived tau, how are tau aggregates purified away from other insoluble species using this protocol? No details are included in the methods section.
- 16) It would be critical to determine if CBE without AD-tau can induce AT8 staining in primary neurons.
- 17) Figure 7i requires an additional control protein to take the place of GCCase, to exclude the possibility that aggregation occurs by increasing total protein/ molecular crowding.
- 18) The GluCer staining in the human brain appears unusual. There is no healthy control image shown, and the DLB section should show at least some normal staining pattern because there is a lot of GluCer in the brain, and this image indicates there is none which is highly unlikely.
- 19) The manuscript is written for a specialist and not written for a general audience of Nature communications. More descriptions of the assays and previous PS19 phenotypes would be helpful throughout.

Reviewer #3 (Remarks to the Author):

In this manuscript, Takahashi et al showed the effect of PGRN deficiency on promoting tau inclusions, concomitant accumulation of α -synuclein, mortality rate and disinhibited behaviors in tauopathy mice. They proposed a PGRN-GCCase pathway that contributes to the tau inclusion. The authors further showed that the reduction of PGRN protects against a spatial memory deficit and hippocampal neurodegeneration and transcriptomic change in PS19 mice. The results are interesting and data are of good quality in general. However, some of the major conclusions are not supported by the results. Another major weakness is the lack of mechanistic link of their various observations, which limit the impact of the study, especially in the context of the current state of the field.

Major concerns:

1. One major conclusion is that PGRN deficiency elevates tau inclusions (Fig. 3). The authors only showed the percentage of type 2 appeared to be elevated, while total AT8 signal was reduced. It is absolutely critical to show the total tau inclusions (as characterized as type 2) is elevated by PGRN deficiency. Fig. 3d is not comprehensible. As it stands, it does not support the relationship between hippocampal atrophy with the different types of AT8 staining.
2. The authors showed that PGRN deficiency in tauopathy results in a decrease in GCCase activity and

therefore accumulate the GLcCer. The authors propose the model that the increase in GlcCer accumulation may promote formation of tau inclusions in PS19 mice. However, the reduction in GCase activity is quite modest. To establish directly the effect of GCase, the authors are encouraged to overexpress GCase in PS19Grn^{-/-} mouse line to determine if it can reverse the phenotypes observed in PS19Grn^{-/-} mice, including tau and alpha-synuclein inclusions, functional improvements etc. Authors can also cross PS19 with GBA^{+/-} or GBA^{-/-} mouse line to make sure it can recapitulate the phenotypes found in PS19Grn^{-/-} mouse line.

3. The protective function of progranulin deficiency against MWM and neurodegeneration is very interesting, but is unfortunately the least studied. The connection between higher tau and alpha-synuclein inclusion, lower Gcase activity with rescue of brain atrophy and memory improvement remains completely unexplored.

Other comments:

1. Fig. 2. The single nuclei analyses are very limited and the striking normalization effects of progranulin haploinsufficiency (Grn^{+/-}) for DAM and DAA is unexplained.

Fig. 5, it is important to plot Gcase activity in all six genotypes, including PS19⁻ on the same plot, and at younger age,

2. Fig. 7a. It is impossible to identify tau pathology in MAP2⁺ neurons given the lack of overlap of the red and green signal. It is unclear how the exogenous seeds (AD-tau) vs. endogenous tau inclusion is differentiated by the staining.

3. Brain PK of CBE needs to be established.

4. Does CBE inhibition occlude the effects of PGRN deficiency on tau inclusion in vitro and in vivo?

5. Fig. 6, the plot needs to separate PSGrn^{+/-} and PSGrn^{-/-}, as in other graphs.

6. The number of human brains is too small to make reliable conclusions,

We have now extensively revised the manuscript by adding new data, re-editing the existing results, and providing detailed description to address each point made by the reviewers. The changes that we made in text are highlighted in red in the revised manuscript.

Most importantly, we have now included lipidomic analysis measuring total 253 lipid species with 16 lipid classes using cortices of 6 genotypes, detailed analysis of GCCase activity using the lysosome-enriched fractions by Optiprep density gradient centrifugation from 6- and 11.5-month-old WT and *Grn*^{-/-} cortices, and in-depth characterization of 7-month-old WT and PS19 brains. Full results of the lipidomic analysis have been provided as supplementary information. In response to the reviewers' comments, we have now provided 10 completely new figures with total 38 panels in the revised manuscript (Supplementary Fig. 3-5, 10, 12-16, 18).

Below is a point-by-point response to the reviewers' comments.

Reviewer #1:

In this manuscript, Takahashi et al demonstrate that reducing progranulin (PGRN) levels in PS19 tau mutant mice leads to enhanced a-synuclein and tau proteinopathies, with the latter typified by an elevation of GluCer (a GCCase substrate) -positive inclusions. Surprisingly, PGRN reduction ameliorated spatial memory defects and hippocampal degeneration in PS19 mice, while worsening alpha-synuclein pathology and premature mortality. The authors confirm that PGRN reduction impairs GCCase activity and promotes the accumulation of the GCCase substrates GlcCer and GlcSph, in agreement with a recent study (Logan et al. Cell 2021). Moreover, the authors show that GlcCer can directly promote tau aggregation, similar to what has previously been reported for a-synuclein, suggesting that the PGRN-GCCase axis may be a commonly dysregulated pathway across neurodegenerative disease, and perhaps hinting at a mechanism through which impairments in PGRN levels can increase disease risk across diverse neurodegenerative diseases (AD, PD, FTD), as has been highlighted by recent GWAS studies. As is such, the findings in this manuscript should be of especially high interest to the AD, PD and FTD fields.

While I am generally supportive of publication of this manuscript, there are a few outstanding questions that I believe could be addressed to increase the overall impact of the findings:

Response: We appreciate the reviewer's positive assessment of our manuscript.

• While the finding that PGRN reduction leads to GlcCer-positive tau inclusions is interesting, the major weakness of the paper is the questionable disease relevance of PGRN reduction on the mutant tau mouse background. Indeed, it has been conclusively established that FTD patients with PGRN mutations do not present with tau pathology, but rather TDP-43 pathology.

Response: We agree with the reviewer that PGRN mutations were initially identified in tau-negative ubiquitin-positive familial FTD cases. However, tau and α -synuclein pathologies have been subsequently also found in many FTLD cases with different PGRN mutations (e.g. please see references #1, #2). Additionally, *GRN* variations including rs5848 T allele have been associated with increased risk for AD (references #3, #4) and PD (reference #5). Our previous study found that *GRN* rs5848 T allele increases CSF tau levels in humans, while having no significant effects on A β PET or CSF A β (reference #6). These studies clearly indicate that in addition to TDP-43, PGRN is involved in tauopathy and synucleinopathy. Thus, FTD-GRN pathophysiology is not exclusively TDP-43 based.

1. Hosokawa et al. Sci Rep. 2017 May 4;7(1):1513. doi: 10.1038/s41598-017-01587-6
2. Leverenz et al. Brain. 2007 May;130(Pt 5):1360-74. doi: 10.1093/brain/awm069
3. Wightman et al. Nat Genet. 2021 Sep;53(9):1276-1282. doi: 10.1038/s41588-021-00921-z
4. Bellenguez et al. Nat Genet. 2022 Apr;54(4):412-436. doi: 10.1038/s41588-022-01024-z
5. Nalls et al. Lancet Neurol 2019 Dec;18(12):1091-1102. doi: 10.1016/S1474-4422(19)30320-5.
6. Takahashi et al. Acta Neuropathol. 2017 May;133(5):785-807. doi: 10.1007/s00401-017-1668-z

• How specific is the GCCase substrate accumulation in the Grn KO x PS19 brain? Although the authors did not observe

any effects on GlcCer, the lipids measured still represent a narrow fraction of the total glycosphingolipids. For instance what happens to sphingomyelin, ceramide, sulfatide, and ganglioside levels? Similarly, what happens to levels of the glycerophospholipid BMP, recently demonstrated by two groups to be deficient in the brains of Grn KO mice (Logan et al. Cell 2021, Boland et al. bioRxiv 2022)?

Response: In response to the reviewer's comment, we have performed multidimensional mass spectrometry-based shotgun lipidomic analysis and measured the levels of additional 16 lipid classes including the ones the reviewer suggested, which are total 253 lipid species, in 6 genotypes in the revised manuscript (Supplementary Fig. 12 and 13). Even without adjustment with the false discovery rate, only a few lipid species were found to be increased (unpaired two-tailed t-test, $p < 0.05$) in *Grn*^{-/-} or PS19 *Grn*^{-/-} cortex compared to WT or PS19, respectively, and none of them was increased in both *Grn*^{-/-} and PS19 *Grn*^{-/-} cortex (Supplementary Fig. 12b,c). We also did not observe an increase of GM1 ganglioside species. The age of mice used might explain the discrepancy between the Boland et al. and our results (18-20 versus 10 months old). Consistent with the previous studies (Laogan et al. Cell 2021, Boland et al. Nat Commun 2022), we observed a significant decrease in BMP species in both *Grn*^{-/-} and PS19 *Grn*^{-/-} cortex. However, the decrease in BMP was not seen in heterozygous *Grn*^{+/-} or PS19 *Grn*^{+/-} cortices even though tau changes were seen in the heterozygotes. In addition, immunohistochemical analysis showed no accumulation of BMP into tau inclusions in PS19 mice (Supplementary Fig. 13b), as opposed to GlcCer. Note that full results of the lipidomic analysis are provided as supplementary information.

• *Similar to the question above, how specific is the effect on in vitro tau aggregation seeding to GlcCer?*

Response: In the in vitro ThT assay, we had examined the specificity of GlcCer using PC, PC/GlcCer, and PC/GlcSph lipid dispersions (Fig. 7e-g). In addition, as described above, none of the lipid classes tested except for GlcCer was increased in PS19 mice with PGRN reduction (Supplementary Fig. 12 and 13a). Therefore, additional lipid controls as aggregation cofactors do not seem to be required here. BMP species were significantly decreased in PS19 *Grn*^{-/-} cortex compared to PS19, but the decrease was not seen in PGRN-haploinsufficient mice, and accumulation of BMP into tau inclusions was not observed in PS19 mice (Supplementary Fig. 13b), suggesting an indirect, if any, role of BMP in tau aggregation. For these reasons, we see no rationale to examine the effect of BMP in tau aggregation in vitro.

• *Along these lines, does GlcCer decorating tau aggregates indicate an active process, or do other potentially co-accumulating lipids similarly label insoluble tau? Is it possible to biochemically purify the tau aggregates and probe their lipid composition in an unbiased manner?*

Response: Previous studies have performed such experiments using human AD brains and found cerebroside in the PHFs (paired helical filaments) (Gellermann et al 2006 and Goux et al 2001, ref. 78 and 79 in the revised manuscript). While other lipids were also found in the PHFs of the previous studies, none of them was increased in either *Grn*^{-/-} or PS19 *Grn*^{-/-} cortex in the present study.

• *What effect, if any, does reducing progranulin levels in vitro have on the AD-tau seeding assay?*

Response: The effect of PGRN deficiency on the AD-tau seeding is assessed in Fig. 7c and 7d in the original manuscript (Now Fig. 7b and 7d in the revised manuscript).

• *While understanding that the hippocampal region was chosen to assess neurodegeneration and for the sc-RNAseq study with tau pathology in mind, what happens in regions known to be preferentially affected in the Grn KO mice alone (e.g. frontal cortex or thalamus)?*

Response: In response to the reviewer's comment, we have examined cresyl violet density in the ventral thalamus of 6 genotypes in the revised manuscript (Supplementary Fig. 3). We found no significant difference between 6 genotypes. Therefore, tau-mediated neurodegeneration appears to be region-specific.

• *Though understandably beyond the scope of this manuscript, do PS19 mice crossed with GBA mutant (e.g. D409V), null or haploinsufficient mice show similarly enhanced tau and alpha-synuclein pathology?*

Response: We thank the reviewer for the great suggestion. Although this would be an interesting experiment, we feel that it is beyond the scope of the present study, requiring more than three years. We hope to address this question in future studies.

However, one thing we would like to note is that aggregation of both α -synuclein and tau has been already observed in *Gba1*^{D409V/D409V} mice without the PS19 background, supporting the conclusion of the present study (please see a figure on the right side and reference #1-4). This point is described in the Discussion. In addition, in the present study, we have used the GCase inhibitor CBE and found that CBE treatment increases tau aggregation induced by AD-tau in primary neurons (Fig. 7a,c) and AT8 p-tau signal in the brain of WT mice (Supplementary Fig. 9j).

Sardi et al. (2013) PNAS, Fig. 1

1. Sardi et al. Proc Natl Acad Sci U S A. 2013 Feb 26;110(9):3537-42. doi: 10.1073/pnas.1220464110.
2. Sardi et al. Proc Natl Acad Sci U S A. 2017 Mar 7;114(10):2699-2704. doi: 10.1073/pnas.1616152114.
3. Clarke et al. Biomedicines. 2021 Apr 21;9(5):446. doi: 10.3390/biomedicines9050446.
4. Viel et al. Sci Rep. 2021 Oct 22;11(1):20945. doi: 10.1038/s41598-021-00404-5.

Reviewer #2:

The manuscript examines the role of Grn in tauopathies using PS19 mouse models and human brain. The general topic is of critical importance, and while some of the data are interesting, there are some significant weaknesses throughout the paper that dramatically reduce enthusiasm for publication. The data showing that Grn reduction can rescue hippocampal volume is interesting, but the correlation with increase tau tangles could be stronger and not completely supported by the data. Primary data is not shown in this readout, and there is only a single assay used (AT8 IHC) to make this conclusion. There are essential controls missing and primary data images omitted throughout. There is a lack of information on how the mouse crosses were done, if the background is mixed, additional technical issues in many of the assays, and the statistical tests change throughout without reason. The data also contradicts well established data from human disease that clearly links reduced Grn and GCase to increase neurotoxicity. While this can be of great interest, it is unfortunately done in a confusing way and without a clear explanation. Specific points are below.

Response: We greatly appreciate the reviewer' thorough review that helped us to improve our manuscript. Please note that our data do not contradict any of previous publications. In the present study, we showed that PGRN reduction worsened mortality and disinhibited behaviors in PS19 mice while attenuating hippocampal neurodegeneration and a spatial memory deficit. Thus, we believe that PGRN reduction is not simply better or worse for tauopathy, but rather it modifies the phenotypes, accelerating FTD-like behavioral deficits while ameliorating hippocampus-associated deficits. This had been elaborated in the Discussion section.

With respect to tau inclusions, we have performed not only AT8 staining (Fig. 3), but also MC1 staining (Supplementary Fig. 10), AT8 and GlcCer co-staining (Fig. 5) and AT8 and p-syn co-staining (Fig. 6) to draw our conclusions.

1) *The hippocampal and memory rescue by Grn -/- is potentially interesting overall, and a strength of the study is the high n number used in the design for most of the studies. However, the authors should address the potential of genetic background differences contributing to the phenotype of memory rescue in the PS19+ / Grn-/- mice (Fig. 1E-I). It is essential to confirm that the rescue doesn't occur from some unknown factor that results from crossing the PS19 mice into the B6 -RIKEN strain. The studies should have been done on a pure background by several rounds of backcrossing,*

to eliminate confounding genetic factors, however these seem to be all hybrids. It also wasn't clear if littermates were used for comparisons, since parentals will likely have a different background mix compared to progeny. This should be clarified in the methods section. Perhaps a SNP panel could be done to determine if the PS19 group has an identical background to the PS19 *Grn*^{+/-} and PS19 *Grn*^{-/-} groups that appear in the study.

Response: We thank the reviewer for raising an important issue. As described in the method, to minimize the effects of genetic background, we first crossed PS19 into *Grn*^{-/-} mice to generate PS19 *Grn*^{+/-} and *Grn*^{+/-}, and then crossed the PS19 *Grn*^{+/-} and *Grn*^{+/-} to generate littermates with 6 genotypes WT, *Grn*^{+/-}, *Grn*^{-/-}, PS19, PS19 *Grn*^{+/-}, PS19 *Grn*^{-/-} at the same generation. These littermates were used in the present study (Please also see the figure on the right side). We agree with the reviewer that it would be optimal to backcrossed PS19 mice more than 10 times then cross the PS19 with a pure C57BL/6J backgrounds into *Grn*^{-/-} mice. However, the backcrossing takes years and there are many studies using littermates of PS19 crosses without backcrossing (e.g., reference #1-4). Our crossing strategy appears to be the same as one in Apicco et al. paper (reference #2). We inadvertently forgot to describe the sentence "littermates were used" in the original manuscript, but we have used littermates at the same generation in the present study and this is now described in the revised manuscript.

1. Shi et al. Nature. 2017 Sep 28;549(7673):523-527. doi: 10.1038/nature24016.
2. Apicco et al. Nat Neurosci. 2018 Jan;21(1):72-80. doi: 10.1038/s41593-017-0022-z
3. Litvinchuk et al. Neuron. 2018 Dec 19;100(6):1337-1353.e5. doi: 10.1016/j.neuron.2018.10.031.
4. Wang et al. Nat Commun. 2022 Apr 12;13(1):1969. doi: 10.1038/s41467-022-29552-6.

2) In figure 1 c,d, the changes in these behavioral tests seem minor – are there differences between the PS19 and PS19 *Grn*^{-/-} groups? Justification for a Dunnett test as opposed to a Tukey test is required. If the study was designed to determine the role *Grn* in PS19 mice, then the PS19 and PS19 *Grn*^{-/-} groups should be directly compared through an ANOVA-Tukey test. It's also inconsistent with figure 3.

Response: In response to the reviewer's comments, we have consistently used one-way ANOVA with Tukey's test (or Kruskal-Wallis test with Dunn's test for nonparametric tests when the data are not normally distributed) for multiple comparisons in the revised manuscript. Figures where statistical tests were changed include Fig. 1a,c,d, 4b, 5h, Supplementary Fig. 1a, 2c,e, 8b, and 11.

For tau pathology measurement such as Fig. 3, 5e, 6, and 7c,d, no tau pathology was expected in WT, *Grn*^{+/-}, and *Grn*^{-/-} brains or AD-tau untreated samples, while clear transgene or AD-tau effects were expected. Thus, the negative-expectation groups were pre-excluded from statistical analysis of tau pathology in this study.

In contrast, for mouse behavioral tests such as Fig. 1, we believe that it is important to include WT (and *Grn*^{+/-} and *Grn*^{-/-}) animals in the analyses and see whether the mouse model (in this case PS19) show the expected phenotypes (deficits) compared to the WT control. In Fig. 1c,d, we have not observed a significant difference between the PS19 and PS19 *Grn*^{-/-} groups likely because of the age tested when PS19 began to show minor disinhibited behaviors. We might have detected a difference if we had tested them at an earlier age. However, as described in the text, PS19 *Grn*^{-/-} displayed significant deficits compared to WT mice while PS19 was yet to show such deficits in Fig. 1c,d. We think that these results still suggest that PGRN deficiency exacerbated PS19 phenotypes. Please note that we also performed MWM and saw a deficit in PS19 and a rescue by PGRN reduction in the memory test. It was not feasible to test all mice in all behavioral analyses at multiple ages to cover every aspect of disease development, though we have provided a comprehensive view of many mice at a specific time point.

3) If the authors wish to correlate pathological changes with behavior, the genders of both studies should match. It isn't clear why both females and males were used for behavior analysis, if the males exhibit more pronounced pathology. Adding in a female group with reduced pathology and neurodegeneration cofounds the behavioral data and makes it very difficult to determine the basis of the behavioral changes.

Response: We have also obtained similar results from behavior tests using only male, as detailed in Supplementary Fig. 1.

4) Previous studies have shown that microgliosis and neurodegenerative changes occur prior to 7 months of age (Yoshiyama et al 2007); therefore, it would be important to confirm that no cell loss or gliosis occurs at 7 months of age for Fig 2 by IHC or other methods.

Response: In response to the reviewer's comment, we have examined brain atrophy, hippocampal neurodegeneration, microgliosis, and tau pathology in 7-month-old PS19 mice in the revised manuscript (n = 9, Supplementary Fig. 4 and 5). Consistent with our snRNA-seq results, we did not observe significant brain atrophy, changes of DG and CA1 cell layers, and Iba1-positive area and cell number in 7-month-old PS19 brains, although there was a moderate increase in AT8 p-tau signal, which was mostly due to the type 1 staining pattern. Importantly, our results were also consistent with many previous studies showing delayed phenotypes (e.g. reference #1-2), compared to the original paper from Virginia Lee's group (Yoshiyama et al 2007). Please note that even in PS19 mice with ApoE4 knock-in background (TE4 mice), neurodegeneration and microgliosis are not observed at 6 months of age (reference #3). This issue is also described in JAX website (<https://www.jax.org/strain/008169>). The reason the delayed phenotypes are observed is currently unknown and requires further investigation.

1. Woerman et al. 2017, JAMA Neurol. 74, 1464-1472
2. DeVos et al. 2017 Sci Transl Med. 9, eaag0481
3. Chen et al. Nature. 2023 Mar;615(7953):668-677 (Fig. 1a,b and Fig. 2a,d)

5) Fig 3 b, f, g— please show the primary data (IHC images) of all three genotypes, showing all 4 subtypes of pathology. It is not possible to tell from the graph what is significantly different from what.

Response: The classification of AT8 p-tau pathology in the PS19 hippocampus has been established by David Holtzman's group and used in many previous studies accepted by major journals (references #1-6). Importantly, the rationale to use the classification is that the subtypes are well correlated with hippocampal neurodegeneration (reference #1 and please also see a figure c on the right side). We have also found (reproduced) the correlation, irrespective of *Grn* genotypes (Fig. 3b). Therefore, we believe that representative images of the 4 subtypes (Fig. 3a) appropriately shows the subtype classification paradigm independently of genotype. Please note that all previous studies (references #1-6) have presented only the 4 subtypes, irrespective of genotypes and treatments in their figures.

Shi et al. (2017) Nature, Figure 2

Importantly, with regard to genotype differences, we provide higher magnification views of AT8-positive tau inclusions and representative images of the different genotypes from multiple brain regions in Fig. 3g,h. For entorhinal cortex and prefrontal cortex, the representative intermediate-magnification images for each genotype, which were used to measure AT8 mean intensity and area (Fig. 3e,f), are now shown in the revised manuscript (Fig. 3d). We have re-organized Fig. 3 and shown Fig 3c (AT8 staining types) as qualitative data, removing the confusing contingency statistics from the revised manuscript. We have added a new AT8 area measurement, which is decreased in PS19 *Grn*^{-/-} mice (Fig. 3f). Our results from both the AT8 mean intensity and AT8+ area suggest an importance of the changes in AT8 staining patterns by PGRN reduction. Additionally, we have also included high resolution confocal image analysis of AT8-positive inclusions in multiple brain regions (CA2+3, amygdala, piriform cortex and prefrontal cortex) in the revised manuscript (Fig. 3g,h).

1. Shi et al. Nature. 2017 Sep 28;549(7673):523-527. doi: 10.1038/nature24016.
2. Martini-Stoica et al. J Exp Med. 2018 Sep 3;215(9):2355-2377. doi: 10.1084/jem.20172158.
3. Shi et al. J Exp Med. 2019 Nov 4;216(11):2546-2561. doi: 10.1084/jem.20190980.
4. Shi et al. Neuron. 2021 Aug 4;109(15):2413-2426.e7. doi: 10.1016/j.neuron.2021.05.034.
5. Grantuze et al. Neuron. 2023 Jan 18;111(2):202-219.e7. doi: 10.1016/j.neuron.2022.10.022.
6. Chen et al. Nature. 2023 Mar;615(7953):668-677. doi: 10.1038/s41586-023-05788-0.

6) Higher resolution confocal or electron microscopy is required to confirm the conclusion that PGRN reduction induces tangle like pathology in figure 3. Thioflavin staining may also help here.

Response: In response to the reviewer's comment, we have examined whether tau inclusions in PS19 mice with PGRN reduction are stained by MC1 antibody in the revised manuscript. MC1 is a commonly-used conformation-dependent antibody specific for PHF-tau (reference #1,2). We found that GlcCer-positive tau inclusions in PS19 mice with PGRN reduction are also consistently labeled by MC1 antibody, suggesting that the PGRN reduction increases pathological tangle-like pathology (Supplementary Fig. 10).

1. Jicha et al. J Neurosci Res 1997 Apr 15;48(2):128-32.
2. Weaver et al. Neurobiol Aging 2000 Sep-Oct;21(5):719-27. doi: 10.1016/s0197-4580(00)00157-3.

7) *What is the significance of the increase CD68 staining in the ps19 Grn^{-/-} group shown in Fig 4A? Could it play a role in rescue phenotype observed in Fig 1, or by redistributing tau pathology from Type 4 into Types 2 or 3?*

Response: As described in the results section, since there are numerous publications showing a role of microglial activation in tauopathy, we explored the possibility that microglial activation might be associated with increased tau inclusions in PS19 mice with PGRN reduction. However, we observed no significant correlation between microglial activation and tau inclusions in Fig. 4d (specifically, significant microglial activation was not seen in type 2 brain samples). Therefore, we focused on GCase and GlcCer to explore the mechanism by which PGRN reduction increases tau inclusions.

8) *In the text, the GC / PGRN pull down is Ext. fig 6 (not 5).*

Response: We thank the reviewer for identifying the error. We have corrected the mismatch in the revised manuscript.

9) *The GCase activity decrease shown in extended fig 6D and Fig 5b,c, is mild. Typically, for the substrate to accumulate as shown in Fig 5, a decrease in activity greater than 50% is required. The GCase activity assay used here doesn't report on problems that could occur from GCase trafficking or environment. The activity assay buffer includes taurocholate, which presumably mimics the function of Sap C, and therefore the enzyme is artificially activated in vitro. If Sap C loss was involved in reduced GCase function in vivo, this assay would not detect it. The same is true for trafficking deficit. The authors should consider a lysosomal fraction to determine GCase content within lysosomes, with and without detergent.*

Response: We thank the reviewer for the great suggestion. In the revised manuscript, we have performed lysosomal fractionation using WT and *Grn^{-/-}* cortices (Supplementary Fig. 14-16). We found two lysosome-enriched fractions (#2 and #3) in the cortex of WT mice after Optiprep density gradient centrifugation (Supplementary Fig. 14c-f). Interestingly, we also found that PGRN deficiency increases accumulation of lysosomal markers in the fraction #3 (Supplementary Fig. 14c-f). We observed ~20% decrease in GCase activity in *Grn^{-/-}* fraction #2, although PGRN deficiency increased GCase activity in the fraction #3 (Supplementary Fig. 15). Similar results were obtained between the assays with and without taurocholate and between 6- and 11.5-month-old animals (Supplementary Fig. 15 and 16). Given the results of lipidomic analysis showing an increase in GlcCer (Fig. 5g,h) and that lysosomal markers were accumulated in the *Grn^{-/-}* fraction #3, we hypothesize that malfunctioning lysosomes are accumulated in the fraction #3 and that lysosomal enzymes in the fraction #3 are not functional *in vivo*. Please note that consistent with the results with and without taurocholate, we did not see a change in the lysosomal localization of saposin C (Sap C), an activator of GCase (Supplementary Fig 15b,c and 16c,d). Taken together, in addition to total GCase activity being decreased, subcellular mislocalization of GCase is likely to contribute to the increase in GlcCer levels in PGRN-deficient mice (Supplementary Fig. 15d).

10) *Figure 5A requires an image of the all of the genotypes quantified in the graph. A counter stain or nuclear stain is required, otherwise it is difficult to judge the comparability of the sections.*

Response: In response to the reviewer's comment, we have shown images of all genotypes in the revised manuscript (Fig. 5d). In Fig. 5a, all sections were co-stained with AT8 antibody, providing evidence of comparable tau inclusions and intact sections in all images. Therefore, we believe that the comparability is not an issue in this case and that an additional nuclear stain is not necessary. There are numerous previous studies accepted by major journals with similar immunofluorescent images without nuclear stain. For example, please see references #1-10.

1. Chen et al. Nature 2023 Mar;615(7953):668-677. Fig 2a, Fig 4e,f,g, Fig 5e,f,g
2. McNamara et al. Nature 2023 Jan;613(7942):120-129. Fig 1c,d,f, Fig 6b
3. Udeochu et al. Nat Neurosci 2023 May;26(5):737-750 Fig 1g,
4. Gratuze et al. Neuron 2023 Jan 18;111(2):202-219.e7. Figure 5C,E,G, Figure 6A,C, Figure 8A,F

5. Lee et al. Nat Neurosci 2022 Jun;25(6):688-701. Fig.1d, Fig2a,e
6. Ennerfelt et al. Cell 2022 Oct 27;185(22):4135-4152.e22. Figure 1E, Figure 2A,C, Figure 3A,E, Figure 5A,C
7. Wang et al. Cell 2022 Oct 27;185(22):4153-4169.e19. Figure 2C, Figure 3F, Figure 4D,F,H, Figure 5I,J, Figure 7F,H
8. Litvinchuk et al. Neuron 2018 Dec 19;100(6):1337-1353.e5. Figure 2D,F, Figure 4A,C,E, Figure 8E
9. Leyns et al. Proc Natl Acad Sci U S A 2017 Oct 24;114(43):11524-11529. Fig. 4B
10. Hong et al. Science 2016 May 6;352(6286):712-716 Fig 2A,C,D

11) Regarding this statement “there was a linear trend in the increase of total GlcCer levels between WT, *Grn*^{+/-} and *Grn*^{-/-} genotypes, although unexpectedly we did not observe the trend between PS19, PS19 *Grn*^{+/-} and PS19 *Grn*^{-/-} genotypes (Fig. 5g,h and Extended Data Fig. 7).”

The study may be powered sufficiently to determine differences between groups, but more mice are required to draw conclusions about trends. The lipid levels in brain can be variable and typically require around 10-12 mice per group. Suggestion is to revise the text here or repeat the study with the appropriate number of mice.

Response: In response to the reviewer’s comment, we have revised the text and essentially deleted these statements from the revised manuscript.

12) In fig 6, does GlcCer colocalize with AT8- α -Syn inclusions? Please explain why the PS19 *Grn*^{+/-} and *-/-* groups are combined in the quantification. In every other data set they are separate. They should be separated here because they are two different groups of mice. As in other IHC images, some counterstain is required to determine the comparability of the sections between the genotypes.

Response: As described in the revised text, AT8-positive tau inclusions were consistently immunoreactive for GlcCer (Fig. 5e). Specifically, 94.0% (322 out of 343), 98.2% (2006 out of 2043), 99.7% (2015 out of 2020) of AT8+ inclusions were immunoreactive for GlcCer in PS19, PS19 *Grn*^{+/-} and PS19 *Grn*^{-/-} brains, respectively. Therefore, essentially all AT8 and p- α -syn double positive inclusions are also immunoreactive for GlcCer (please see images on the right side).

In response to the reviewer’s comment, the two groups (PS19 *Grn*^{+/-} and PS19 *Grn*^{-/-}) have been separately shown in Fig. 6c of the revised manuscript. Please note that while the PS19 model is extremely useful in studying tau pathology and tau-mediated neurodegeneration, it is also well-known that the number of tau inclusions in the PS19 brain is highly variable between animals, and do not follow Gaussian distribution (reference #1, please see eFigure 1D, reference #2), and that is partly why the classification of AT8 p-tau pathology has been used in several previous publications. Therefore, to account for local variability of AT8-positive inclusions, we sampled multiple higher resolution images from each mouse to provide a statistically valid survey of tau inclusion density, as in a previous study (reference #2).

As for the counterstaining, similar to Fig. 5a, all sections were co-stained (counterstained) with AT8 antibody in Fig. 6a, providing evidence of comparable tau inclusions and intact sections in all images. Therefore, we believe that the comparability is not an issue in this case and that an additional nuclear stain is not necessary. There are numerous accepted publications with similar immunofluorescent images without nuclear stain. Please see references #1-10 listed in response to the comment #10 above.

1. Woerman et al, JAMA Neurol, 2017, 74, 1464-1472. doi:10.1001/jamaneurol.2017.2822.
2. Wang et al. Nat Commun. 2022 Apr 12;13(1):1969. doi: 10.1038/s41467-022-29552-6.

13) Synuclein pathology should be examined in *Grn*^{-/-} mice (without p301 tau).

Response: In response to the reviewer’s comment, we have added the data in the revised manuscript (Supplementary Fig. 18). Neither tau inclusions nor p- α -syn-and AT8-positive inclusions were observed in *Grn*^{+/-} and *Grn*^{-/-} mice without PS19 background, although we confirmed the staining protocol using contemporaneous positive controls (PS19 *Grn*^{-/-} samples).

In the bottom panels, the section was incubated only with anti-GlcCer antibody, followed by incubation with Alexa Fluor 488 and 568 secondary antibodies. All images were taken using the same setting.

14) In fig 6D and extended fig 8b, the specificity of the GCCase antibody should be demonstrated. No commercially available GCCase antibodies are known to work at all in fixed tissue/cultures.

Response: The antibody has been used in immunostaining of a previous study (Arrant et al. 2019 Acta Neuropathol Commun 7:218). We have observed a clear correlation between the immunoreactivity of this antibody and GCCase activity, validating the antibody by immunoblot (Supplementary Fig. 15 and 16) and a lysosome-consistent distribution of GCCase by immunostaining in Fig. 8e.

15) The data in Fig 7 A, it appears that the tau signal is outside of neurons. A magnified image would help, along with a nuclei stain. For AD-derived tau, how are tau aggregates purified away from other insoluble species using this protocol? No details are included in the methods section.

Response: The AD-tau seeding assay was developed by Virginia Lee's group (references #1 and 2). In this assay, the mouse tau pathology has been observed not only in MAP2-positive dendrites but also in NFL-positive axons (reference #2, Fig.1, please see a figure on the right side), which explains the partial overlap between red and cyan in the present study. Please also note that MAP2 is just a microtubule-associated protein, not a membrane-associated protein. So, even dendritic tau aggregates cannot always be completely within MAP2 area. We have described the detailed methodology in the text, increased the brightness of MAP2 signal, and added magnified images in the revised manuscript (Fig. 7a,b).

For AD-tau extraction, we have followed the protocol of the Virginia Lee's paper (reference #1) and the exact protocol has been written in our previous paper (reference #3), as cited in the Methods section. The AD-tau we used has been also characterized in detail in our newly published paper (reference #4), which is now also cited in the revised manuscript. We have also detailed this information in the revised manuscript. Based on the total protein and tau concentrations (reference #4), the AD-tau likely contains other proteins as well. However, we have shown that the induction of mouse tau pathology is dependent on tau concentrations in AD-tau, and found that no tau pathology was induced by parallel brain extracts from healthy controls (reference #4) *in vitro*. Both the AD-tau and its extraction protocol are commonly accepted and used in the field as a tool to induce tau pathology in both *in vitro* and *in vivo* (e.g. references #5-11).

Xu et al. (2021) Acta Neuropathol, Fig. 1

1. Guo et al. J Exp Med. 2016 Nov 14;213(12):2635-2654. doi: 10.1084/jem.20160833.
2. Xu et al. Acta Neuropathol. 2021 Feb;141(2):193-215. doi: 10.1007/s00401-020-02253-4.
3. Tang et al. Acta Neuropathol Commun. 2020 Jul 1;8(1):96. doi: 10.1186/s40478-020-00976-9.
4. Nies et al. J Biol Chem. 2021 Oct;297(4):101159. doi: 10.1016/j.jbc.2021.101159.
5. He et al. Nat Med. 2018 Jan;24(1):29-38. doi: 10.1038/nm.4443.
6. Leyns et al. Nat Neurosci. 2019 Aug;22(8):1217-1222. doi: 10.1038/s41593-019-0433-0.
7. Gratuze et al J Exp Med 2021 Aug 2;218(8):e20210542. doi: 10.1084/jem.20210542. Epub 2021 Jun 8.
8. Saroja et al. Proc Natl Acad Sci U S A 2022 Aug 23;119(34):e2108870119. doi: 10.1073/pnas.2108870119
9. Kim et al. Biol Psychiatry 2023 May 1;93(9):829-841. doi: 10.1016/j.biopsych.2022.10.015.
10. Wand et al. J Clin Invest. 2023 Jul 17;133(14):e169131. doi: 10.1172/JCI169131.
11. Zhao et al. Sci Transl Med 2023 Sep 13;15(713):eabo6889. doi: 10.1126/scitranslmed.abo6889.

16) *It would be critical to determine if CBE without AD-tau can induce AT8 staining in primary neurons.*

Response: In response to the reviewer's comment, we have added the data in the revised manuscript (Fig. 7a-d). In this assay system, CBE treatment or PGRN deficiency alone did not induce aggregation of endogenous mouse tau in mouse primary neurons.

17) *Figure 7i requires an additional control protein to take the place of GCCase, to exclude the possibility that aggregation occurs by increasing total protein/ molecular crowding.*

Response: We had performed the control experiment using BSA in Extended Data Figure S9 of the original manuscript (now Supplementary Fig. 19 in the revised manuscript).

18) *The GluCer staining in the human brain appears unusual. There is no healthy control image shown, and the DLB section should show at least some normal staining pattern because there is a lot of GluCer in the brain, and this image indicates there is none which is highly unlikely.*

Response: Human postmortem brain samples have strong autofluorescence. Thus, as described in the Methods, in addition to CuSO₄ treatment, we have used a serial section incubated only with secondary antibody as a negative control to check the background signal side-by-side and employed a constant low confocal laser power that does not detect autofluorescent lipofuscin signal from any sections lacking primary antibody. In some cases, the low laser power could not detect normal diffuse lower level GlcCer signal although it did detect GlcCer accumulated in the NFTs. We have added this explanation in Method section of the revised manuscript.

19) *The manuscript is written for a specialist and not written for a general audience of Nature communications. More descriptions of the assays and previous PS19 phenotypes would be helpful throughout.*

Response: We thank the reviewer for the suggestion. We have described known PS19 phenotypes and the methodology of AD-tau seeding assay in the Introduction and Results sections of the revised manuscript, respectively.

Reviewer #3:

In this manuscript, Takahashi et al showed the effect of PGRN deficiency on promoting tau inclusions, concomitant accumulation of a-synuclein, mortality rate and disinhibited behaviors in tauopathy mice. They proposed a PGRN-GCase pathway that contributes to the tau inclusion. The authors further showed that the reduction of PGRN protects against a spatial memory deficit and hippocampal neurodegeneration and transcriptomic change in PS19 mice. The results are interesting and data are of good quality in general. However, some of the major conclusions are not supported by the results. Another major weakness is the lack of mechanistic link of their various observations, which limit the impact of the study, especially in the context of the current state of the field.

Response: We greatly appreciate the reviewer's helpful comments to improve our manuscript.

Major concerns:

1. *One major conclusion is that PGRN deficiency elevates tau inclusions (Fig. 3). The authors only showed the percentage of type 2 appeared to be elevated, while total AT8 signal was reduced. It is absolutely critical to show the total tau inclusions (as characterized as type 2) is elevated by PGRN deficiency. Fig. 3d is not comprehensible. As it stands, it does not support the relationship between hippocampal atrophy with the different types of AT8 staining.*

Response: In response to the reviewer's comment, we have also shown the number of AT8-positive tau inclusions in several brain regions (CA2+3, amygdala, piriform cortex and prefrontal cortex) in the revised manuscript (Fig. 3g,h and Fig. 5d,e). Please note that while the PS19 model is extremely useful in studying tau pathology and tau-mediated neurodegeneration, it is also well-known that the number of tau inclusions in the PS19 brain is variable between animals, and do not follow Gaussian distribution (reference #1, please see eFigure 1D, reference #2), and that is partly why the classification of AT8 p-tau pathology has been used in several previous publications. Therefore, to account for local variability of AT8-positive inclusions, we sampled multiple higher resolution images from each mouse to provide a statistically valid survey of tau inclusion density, as in a previous study (reference #2).

Shi et al. (2017) Nature, Figure 2

The classification and association of AT8 p-tau pathology with brain atrophy in the PS19 hippocampus (please see the figures on the right side) have been established by David Holtzman's group and commonly used in multiple previous studies accepted by major journals (references #3-8). Our blinded experiment has also reproduced the correlation, irrespective of *Grn* genotypes (Fig. 3b), supporting the relationship between brain atrophy and the subtypes of AT8 staining patterns.

1. Woerman et al, JAMA Neurol, 2017, 74, 1464-1472. doi:10.1001/jamaneurol.2017.2822.
2. Wang et al. Nat Commun. 2022 Apr 12;13(1):1969. doi: 10.1038/s41467-022-29552-6.
3. Shi et al. Nature. 2017 Sep 28;549(7673):523-527. doi: 10.1038/nature24016.
4. Martini-Stoica et al. J Exp Med. 2018 Sep 3;215(9):2355-2377. doi: 10.1084/jem.20172158.
5. Shi et al. J Exp Med. 2019 Nov 4;216(11):2546-2561. doi: 10.1084/jem.20190980.
6. Shi et al. Neuron. 2021 Aug 4;109(15):2413-2426.e7. doi: 10.1016/j.neuron.2021.05.034.
7. Grantuze et al. Neuron. 2023 Jan 18;111(2):202-219.e7. doi: 10.1016/j.neuron.2022.10.022.
8. Chen et al. Nature. 2023 Mar;615(7953):668-677. doi: 10.1038/s41586-023-05788-0.

2. *The authors showed that PGRN deficiency in tauopathy results in a decrease in GCase activity and therefore accumulate the GLcCer. The authors propose the model that the increase in GlcCer accumulation may promote formation of tau inclusions in PS19 mice. However, the reduction in GCase activity is quite modest. To establish directly the effect of GCase, the authors are encouraged to overexpress GCase in PS19Grn^{-/-} mouse line to determine if it can reverse the phenotypes observed in PS19Grn^{-/-} mice, including tau and alpha-synuclein inclusions, functional improvements etc. Authors can also cross PS19 with GBA^{+/-} or GBA^{-/-} mouse line to make sure it can recapitulate the phenotypes found in PS19Grn^{-/-} mouse line.*

Response: We thank the reviewer for the excellent suggestions. In agreement with reviewer #1, we feel that studying PS19 crossing into *Gba1* mice is beyond the scope of our current study, taking more than three years. We hope to address this question in future studies. However, please note that aggregation of both α -synuclein and tau has been already observed in *Gba1*^{D409V/D409V} mice without PS19 background, supporting conclusion of the present study (please see a figure on the right side and reference #1-4). This point is described in the Discussion.

In addition, to establish the direct effect of GlcCer/GCase on tau aggregation, we have performed cell culture and in vitro experiments. We have found that GCase inhibitor CBE increases tau aggregation induced by AD-tau in primary cultured neurons (Fig. 7a,c) and AT8 p-tau signal in WT brains (Supplementary Fig. 9j). By using ThT assay, we also showed that purified GlcCer directly promotes aggregation of purified tau (Fig. 7e-h).

Sardi et al. (2013) PNAS, Fig. 1

1. Sardi et al. Proc Natl Acad Sci U S A. 2013 Feb 26;110(9):3537-42. doi: 10.1073/pnas.1220464110.
2. Sardi et al. Proc Natl Acad Sci U S A. 2017 Mar 7;114(10):2699-2704. doi: 10.1073/pnas.1616152114.
3. Clarke et al. Biomedicines. 2021 Apr 21;9(5):446. doi: 10.3390/biomedicines9050446.
4. Viel et al. Sci Rep. 2021 Oct 22;11(1):20945. doi: 10.1038/s41598-021-00404-5.

3. *The protective function of progranulin deficiency against MWM and neurodegeneration is very interesting, but is unfortunately the least studied. The connection between higher tau and alpha-synuclein inclusion, lower Gcase activity with rescue of brain atrophy and memory improvement remains completely unexplored.*

Response: In the present study, we have extensively investigated the protective function of PGRN reduction against a memory deficit and neurodegeneration in PS19 mice.

Critically, we performed snRNA-seq of the hippocampus. The extensive analysis revealed that PGRN reduction attenuates transcriptomic changes in PS19 hippocampus irrespective of cell types (Fig. 2d-g) and does not induce a significant transcriptomic change in WT hippocampus (Supplementary Fig. 6d, e). These results indicate that PGRN reduction may affect tau pathology upstream of global rescue of the transcriptomic changes.

Mechanistically, we examined hippocampal tau pathology. We found that PGRN reduction decreases AT8 signal intensity and area in general (Fig. 3e,f), more specifically causes a shift of AT8 staining subtypes from type 3, 4 to type 1, 2 in the hippocampus (Fig. 3c) and increases AT8-positive inclusions in the hippocampus and amygdala (Fig. 3g,h). These results suggest that PGRN reduction alters tau pathology and thereby attenuates transcriptomic changes, neurodegeneration, and a memory deficit.

Finally, we examined the mechanisms of increased tau inclusions by PGRN reduction. We found that PGRN reduction decreases GCase activity and increases GlcCer (Fig. 5) and thereby increases tau inclusions (Fig. 7).

Please note that these results had been extensively discussed and summarized in Extended Data Figure S10a (now Supplementary Fig. 20a in the revised manuscript).

Other comments:

1. *Fig. 2. The single nuclei analyses are very limited and the striking normalization effects of progranulin haploinsufficiency (*Grn*^{+/-}) for DAM and DAA is unexplained.*

Response: As described above, the normalization (transcriptome-wide rescue effect, reference #1) occurs not only in glial cells but also in other cell types (Fig. 2d). Therefore, we focused on the hypothesis that the effect of PGRN reduction was not unique to specific cell types, but rather altered tau pathology and thereby caused global rescue of transcriptomic changes in PS19 mice.

1. Lee et al. *Neuron* 2018 **97**, 1032-1048 e1035, doi:10.1016/j.neuron.2018.02.002

Fig. 5, it is important to plot Gcase activity in all six genotypes, including PS19- on the same plot, and at younger age,

Response: The GCase activity was measured in different batches as indicated in Fig. 5b, c, and Supplementary Fig. 9b) due to availability of mice and tissue, so it is not appropriate to combine these data. However, we have analyzed GlcCer levels in all six genotypes in parallel using SFC-MS/MS (Fig. 5f,g and Supplementary Fig. 11), which are a more direct and important measurement, since, as pointed out by the comment #9 from the reviewer #2, the GCase activity assay in Fig. 5 reports total GCase activity in the brain homogenates and does not accurately reflect activity of GCase localized at the lysosome. We found that several species of GlcCer and total GlcCer levels were increased in both *Gm^{-/-}* and PS19 *Gm^{-/-}* cortices (Fig. 5f,g and Supplementary Fig. 11).

Additionally, in the revised manuscript, we have performed Optiprep density gradient centrifugation and measured GCase activity using lysosome-enriched fractions as well as total lysates from WT and *Gm^{-/-}* cortices at both 6 and 11.5 months of age and similar results were obtained between 6 and 11.5 months (Supplementary Fig. 14-16).

2. Fig. 7a. It is impossible to identify tau pathology in MAP2+ neurons given the lack of overlap of the red and green signal. It is unclear how the exogenous seeds (AD-tau) vs. endogenous tau inclusion is differentiated by the staining.

Response: The AD-tau seeding assay was developed by Virginia Lee's group (references #1 and 2). A key point of this assay is that endogenous mouse tau pathology is detected by an antibody against mouse tau (T49), which does not detect exogenous human AD-tau. Soluble mouse tau was removed during methanol fixation and washing (please see Figures on the right side). In this assay, the mouse tau pathology has been observed not only in MAP2-positive dendrite, but also in NFL-positive axons (reference #2, Fig. 1, please see a figure on the next page), which explains the partial overlap between red and cyan in the present study. Please also note that MAP2 is just a microtubule-associated protein, not a membrane-associated protein. So, even dendritic tau aggregates cannot always be completely within MAP2 area. We have described the detailed methodology in the text, increased the brightness of MAP2 signal, and added magnified images in the revised manuscript (Fig. 7a,b).

1. Guo et al. J Exp Med. 2016 Nov 14;213(12):2635-2654. doi: 10.1084/jem.20160833.
2. Xu et al. Acta Neuropathol. 2021 Feb;141(2):193-215. doi: 10.1007/s00401-020-02253-4.

Gou et al. (2021) JEM, Figure 2

Xu et al. (2021) Acta Neuropathol, Fig. 1

3. Brain PK of CBE needs to be established.

Response: Effective doses of CBE for *in vivo* use have been established (please see reference #1). Now the reference has been cited in Methods section of the revised manuscript. We used older mice compared to those used in the publication, therefore carefully monitored their body weight (Supplementary Fig. 9g), which is known to be affected by CBE (reference #1), and euthanized animals when weight declined.

1. Vardi et al. J Pathol. 2016 Aug;239(4):496-509. doi: 10.1002/path.4751.

4. Does CBE inhibition occlude the effects of PGRN deficiency on tau inclusion *in vitro* and *in vivo*?

Response: Long-term treatments of mice with CBE that could recapitulate chronic effects of PGRN deficiency on GCase have not been established (reference #1). Most likely it will not be possible to achieve reproducible constitutive 10-20% reduction of GCase activity (matching PGRN deficiency) with CBE treatment. Genetic manipulation of GCase activity might be an alternative and more feasible approach (e.g. generation of PS19 *Grn*^{-/-} *Gba1*-KI mice), but we believe that such experiments are beyond the scope of our current study, as described above.

1. Vardi et al. J Pathol. 2016 Aug;239(4):496-509. doi: 10.1002/path.4751.

5. Fig. 6, the plot needs to separate PSGrn^{+/-} and PSGrn^{-/-}, as in other graphs.

Response: In response to the reviewer's comment, we have separated those two genotypes in the revised manuscript.

6. The number of human brains is too small to make reliable conclusions,

Response: The primary focus of this study is the mouse genetic analysis, and we sought a focused confirmation of specific observations in human autopsy tissue. We have collected human brain samples from three independent institutes. The results are clear and statistically significant. However, future larger scale pathology-focused surveys will be of interest. We have added this limitation in discussion of the revised manuscript.

REVIEWER COMMENTS

Reviewer #2 (Remarks to the Author):

The authors have provided a comprehensive response and some of the manuscript has been improved, but there remain points of confusion that should be clarified.

For example in figure 1 A, C, D, there does not appear to be any difference between the PS19 and PS19/Grn^{-/-} mice. But, the bold subtitle indicates that PGRN reduction worsens disinhibition of PS19 mice. In the next section, the morris water maze and correlated neurodegeneration analysis do show that GRN loss influences PS19 phenotype (panels e-i). Therefore the authors may want highlight this result better and perhaps place panels A, C, and D in a supplement, or remove the data and further develop it for another study. If disinhibition is worsened by GRN loss, there should be some statistical difference seen between PS19 and PS19/GRN^{-/-}. The authors could also examine the neural correlate for inhibited behaviors by analyzing for cell loss in this area (eg prefrontal cortex), comparing PS19 vs PS19/Grn^{-/-}.

Regarding the previous statement that the current data contradicts previous publications, the authors claim otherwise. Apologies if I was not clear but I was referring to genetic data from human patients and many molecular studies showing that loss of GCase protein and activity result in neurotoxicity in synucleinopathies. Loss of GCase and GlcCer and / or GlcSph accumulation are linked to cell loss in Gaucher disease. However, the authors claim here that loss of GCase and GlcCer improve neuron function(water maze) and rescue cell loss, which is interesting but in general at odds with decades of work on Gaucher disease and Parkinson's disease. The authors mention that progression may be closer to FTD (disinhibition first, memory impairments later) but there is no difference between PS19 and PS19/GRN^{-/-} mice in fig 1 C, D. Therefore, the authors could have done a better job at explaining the relationship of their data to this large body of work on Gaucher disease and Parkinson's disease.

Figure 3H – The figure legend indicates the data are from 15-20 mice using 3 ROIs. I am assuming the plots represent an individual ROI. However, the ROIs should be averaged so that one plot represents the value from one mouse, and statistics should be done in this way. It is more informative to plot the variation of individual mice as opposed to individual ROIs, which may influence statistical significance.

Figure 7i – This is an interesting result, but there is no molecular crowding control for the tau aggregation assay, and this is essential to include. Since the total protein concentration of the tau + GCase condition is increased compared to tau + vehicle, there is no way to determine if the effect comes from increased protein concentration alone.

The lysosomal fractionation seemed to work, given the presence of cathepsin in fractions 2 and 3, but it is not clear how the authors could know that fraction 3 represents non-functional lysosomes. It seems that fraction 3 contains lysosomal enzymes in both WT and Grn^{-/-} lines. Does this mean the WT mice have dysfunctional lysosomes? The enzyme activity assays indicate that F3 actually has more function

compared to F2, therefore this result caused some confusion. If GCase accumulates in non-functional lysosomes, the authors should find some way to directly test this, and not rely only on the correlation with GlcCer levels. This is because GlcCer steady-state levels come from multiple pathways including synthesis/ de novo pathway or recycling pathways. Overall it is difficult to connect this data with the increases in GlcCer.

In Supplemental Figure 15B, please explain the asterisks. Including a total lysate control would be helpful. It is normal to see multiple forms of GCase by western analysis, however the lower band is increasing in F2, while the upper band is decreasing by a subtle amount. Did the authors quantify the upper band only? There is a similar issue for F3. If the authors could digest the lysates with a glycosidase to remove glycans, this may simplify the analysis and help to determine if they quantified the correct band. Also, given that lysosomal forms of GCase migrate slower, it would be expected to see only high (glycosylated) MW bands in these fractions.

There remain many grammatical errors making the paper hard to read and understand in some places

Reviewer #3 (Remarks to the Author):

The authors have addressed this reviewers' comments. The revised manuscript is much improved.

Reviewer #4 (Remarks to the Author):

Based on assessment of the original reports from the three referees, the authors have substantially improved their manuscript over the course of the revision. There is generally great interest in understanding the mechanisms linking GRN genetics to diseases other than GRN-associated FTL, including AD and PD, particularly as novel therapies are being developed and tested in the clinic. I am generally supportive of publication. I would just recommend that the authors should capture a bit more comprehensively the various molecular mechanisms linking PGRN deficiency to decreased GCase activity, in their discussion. As discussed in-depth in a recent TICB review article (PMID 36244875), those potentially include: (1) loss of direct chaperone activity; (2) reduced PSAP levels, processing or reduced SapC levels; (3) reduced maturation and lysosomal delivery of GCase; and (4) reduced levels of BMP, which can stimulate GCase activity via electrostatic interactions with protonated SapC and/or GCase itself. It seems that the authors settled on (3) as a likely mechanism, but the other three cannot be excluded.

We are most grateful to the Reviewers for their very careful evaluation and helpful comments to improve our manuscript.

We have now revised the manuscript Results and Discussion to address each point made by the reviewers. All changes that we made are highlighted in red in the revised manuscript.

Please note that in response to the editorial requests, we have now separated Supplementary Information in a single file and provided uncropped blots in Supplementary Fig. 21. We have also provided full results of lipidomic analysis for Glc/GalCer as Supplementary Data1 with our revised manuscript. Finally, we have also shown exact p-values (except for $p < 0.0001$) in figure legends of the revised manuscript.

We trust that the revised form of our manuscript has addressed all the points raised by the reviewers, complied with the editorial guidelines and will now be acceptable for publication.

Below is a point-by-point response to the reviewers' comments.

Reviewer #2:

The authors have provided a comprehensive response and some of the manuscript has been improved, but there remain points of confusion that should be clarified.

Response: We thank the reviewer again for the thorough review and the positive comments.

For example in figure 1 A, C, D, there does not appear to be any difference between the PS19 and PS19/Grn^{-/-} mice. But, the bold subtitle indicates that PGRN reduction worsens disinhibition of PS19 mice. In the next section, the morris water maze and correlated neurodegeneration analysis do show that GRN loss influences PS19 phenotype (panels e-i). Therefore the authors may want highlight this result better and perhaps place panels A, C, and D in a supplement, or remove the data and further develop it for another study. If disinhibition is worsened by GRN loss, there should be some statistical difference seen between PS19 and PS19/GRN^{-/-}. The authors could also examine the neural correlate for inhibited behaviors by analyzing for cell loss in this area (eg prefrontal cortex), comparing PS19 vs PS19/Grn^{-/-}.

Response: In the present study, PS19 Grn^{-/-} mice showed significant body weight loss, increased mortality due to hindlimb paralysis, and disinhibited behaviors compared to WT mice even when PS19 or Grn^{-/-} mice did not show such deficits compared to WT in Fig. 1a-d. Thus, these results demonstrate that PGRN deficiency and tauopathy synergistically worsens motor deficits and disinhibited behaviors. There are numerous previous studies accepted by major journals with similar results and interpretation. For example, please see the figures in references #1-9. In the revised manuscript, we have edited the heading and text to describe the results more precisely.

We believe that it is critical to keep these data in the main figures of this manuscript to avoid reader's misunderstanding that PGRN reduction or decreased GCase activity is simply beneficial in tauopathy.

1. Depp et al. Nature 2023 Jun;618(7964):349-357. Extended Data Fig. 3p, EPM
2. Wang et al. Cell 2022 Oct 27;185(22):4153-4169.e19. Figure 2G
3. Brody et al Mol Neurodegener 2022 May 3;17(1):32. Fig. 5B
4. Trzeciakiewicz et al. Nat Commun 2020 Nov 2;11(1):5522. Fig. 5b,f
5. Zhu et al. Nat Neurosci 2020 May;23(5):615-624. Fig. 3f, 12 and 18 months
6. Chalermphanupap et al. J Neurosci 2018 Jan 3;38(1):74-92. Fig. 5H,I, and 6E
7. Min et al. J Neurosci. 2018 Apr 11; 38(15): 3680–3688. Figure 4B
8. Chakrabarty et al. Neuron 2015 Feb 4;85(3):519-33. Fig. 3D
9. Minami et al. Nat Med 2014 Oct;20(10):1157-64. Fig. 4a,c

Regarding the previous statement that the current data contradicts previous publications, the authors claim otherwise. Apologies if I was not clear but I was referring to genetic data from human patients and many

molecular studies showing that loss of GCCase protein and activity result in neurotoxicity in synucleinopathies. Loss of GCCase and GlcCer and / or GlcSph accumulation are linked to cell loss in Gaucher disease. However, the authors claim here that loss of GCCase and GlcCer improve neuron function (water maze) and rescue cell loss, which is interesting but in general at odds with decades of work on Gaucher disease and Parkinson's disease. The authors mention that progression may be closer to FTD (disinhibition first, memory impairments later) but there is no difference between PS19 and PS19/GRN^{-/-} mice in fig 1 C, D. Therefore, the authors could have done a better job at explaining the relationship of their data to this large body of work on Gaucher disease and Parkinson's disease.

Response: Please note that PGRN reduction causes only up to ~20% decrease in lysosomal GCCase activity. Therefore, the resultant phenotypes should not be as severe as ones seen in mouse models and patients of Gaucher disease, an autosomal recessive disease caused by biallelic mutations in *GBA1* gene.

Our results showed that the reduced GCCase activity by PGRN deficiency leads to an increase in GlcCer-positive tau inclusions, which was unexpectedly protective against hippocampal atrophy and a memory deficit in PS19 mice. However, we also found that PGRN reduction in tauopathy caused increased co-accumulation of p- α -syn, motor deficits, and disinhibited behaviors. These results are consistent with the previous genetic studies suggesting that *GRN* variants as well as heterozygous *GBA1* variants are associated with increased risk for PD with p- α -syn accumulation and motor deficits. Understanding the precise roles of PGRN in PD requires further investigation using animal and cellular models of PD. We have added this discussion in the revised manuscript.

Figure 3H – The figure legend indicates the data are from 15-20 mice using 3 ROIs. I am assuming the plots represent an individual ROI. However, the ROIs should be averaged so that one plot represents the value from one mouse, and statistics should be done in this way. It is more informative to plot the variation of individual mice as opposed to individual ROIs, which may influence statistical significance.

Response: Please note that while the PS19 model is extremely useful in studying tau pathology and tau-mediated neurodegeneration, it is also well known that the number of tau inclusions in the PS19 brain is highly variable between animals and between ROIs in one animal, and do not follow Gaussian distribution (reference #1, please see eFigure 1D, reference #2). Therefore, to account for local variability of AT8-positive inclusions and to provide a statistically valid survey of tau inclusion density, we sampled multiple higher resolution images from each mouse, as performed in previous studies (reference #2-4, please also see the figure on the right side).

1. Woerman et al, JAMA Neurol, 2017, 74, 1464-1472. doi:10.1001/jamaneurol.2017.2822.
2. Wang et al. Nat Commun. 2022 Apr 12;13(1):1969. doi: 10.1038/s41467-022-29552-6
3. Yan et al. Cell 2022 Oct 13;185(21):3913-3930.e19. doi: 10.1016/j.cell.2022.09.002.
4. Udeochu et al. Nat Neurosci 2023 May;26(5):737-750. doi: 10.1038/s41593-023-01315-6. Fig. 7ik

Figure 7i – This is an interesting result, but there is no molecular crowding control for the tau aggregation assay, and this is essential to include. Since the total protein concentration of the tau + GCCase condition is increased compared to tau + vehicle, there is no way to determine if the effect comes from increased protein concentration alone.

Response: The requested control experiment using BSA is provided in Supplementary Fig. 19. Addition of BSA had no significant effect on tau aggregation, while GCCase did affect tau aggregation.

Wang et al. (2022) Nat Commun
Figure 6a-c, n= 9 (+/+) or 12 (-/-),
8 sections/mouse

The lysosomal fractionation seemed to work, given the presence of cathepsin in fractions 2 and 3, but it is not clear how the authors could know that fraction 3 represents non-functional lysosomes. It seems that fraction 3 contains lysosomal enzymes in both WT and Grn^{-/-} lines. Does this mean the WT mice have dysfunctional lysosomes? The enzyme activity assays indicate that F3 actually has more function compared to F2, therefore this result caused some confusion. If GCCase accumulates in non-functional lysosomes, the authors should find some way to directly test this, and not rely only on the correlation with GlcCer levels. This is because GlcCer steady-state levels come from multiple pathways including synthesis/ de novo pathway or recycling pathways. Overall it is difficult to connect this data with the increases in GlcCer.

Response: Thank you for accepting that our lysosomal fractionation worked. In the present study, our co-IP assay confirmed physical interaction between PGRN and GCCase. We also found that PGRN deficiency causes up to ~10% decrease in GCCase activity in cortical brain lysates. These results are consistent with many previous studies (reference #1-5). Our experiments using lysosome-enriched fractions showed that PGRN deficiency alters the appropriate subcellular localization of GCCase, which is also consistent with a previous study demonstrating that PGRN is required for lysosomal localization of GCCase (reference #6). Finally, our comprehensive lipidomic analysis showed that GlcCer and BMP are the only lipid classes significantly affected by PGRN deficiency at 10 months of age. Therefore, it appears that the likeliest explanation for an increase in GlcCer in PGRN-deficient brains is a partial loss of total amounts and lysosomal localization of GCCase, although we agree that other possibilities cannot be excluded. We have added a paragraph of this discussion in the revised manuscript.

1. Jian et al. EBioMedicine 2016 Nov;13:212-224. doi: 10.1016/j.ebiom.2016.10.010.
2. Zhou et al. PLoS One 2019 Jul 10;14(7):e0212382. doi: 10.1371/journal.pone.0212382.
3. Arrant et al. Acta Neuropathol Commun 2019 Dec 23;7(1):218. doi: 10.1186/s40478-019-0872-6.
4. Logan et al. Cell 2021 Sep 2;184(18):4651-4668.e25. doi: 10.1016/j.cell.2021.08.002.
5. Reifschneider et al. EMBO J 2022 Feb 15;41(4):e109108. doi: 10.15252/emboj.2021109108.
6. Jian et al. EBioMedicine 2016 Sep;11:127-137.doi: 10.1016/j.ebiom.2016.08.004.

In Supplemental Figure 15B, please explain the asterisks. Including a total lysate control would be helpful. It is normal to see multiple forms of GCCase by western analysis, however the lower band is increasing in F2, while the upper band is decreasing by a subtle amount. Did the authors quantify the upper band only? There is a similar issue for F3. If the authors could digest the lysates with a glycosidase to remove glycans, this may simplify the analysis and help to determine if they quantified the correct band. Also, given that lysosomal forms of GCCase migrate slower, it would be expected to see only high (glycosylated) MW bands in these fractions.

Response: Thank you so much for pointing out this issue. We inadvertently forgot to explain the asterisks, but we believe that the lower bands with the asterisks are non-specific bands because it appears that the bands show up only when the membrane was overexposed with primary (anti-GCase) antibody in our experiments, while the upper bands were consistently detected irrespective of the conditions. For example, we didn't observe the lower bands in 11.5 mo F3 samples in Supplementary Figure 16. In addition, as shown in the right panel, the lower bands in F2 samples barely showed up when primary antibody was applied only for 2 hours (versus overnight in Supplementary Fig. 15b) in our pilot experiment using the same samples as ones used in Supplementary Fig. 15b (but just changing the loading order). Therefore, we have quantified only the upper bands. Please note that the upper band intensity highly correlated with *in vitro* GCase activity as shown in Supplementary Fig. 15d and 16e. We have included this explanation in figure legends of Supplementary Fig. 15b and 16c of the revised manuscript.

There remain many grammatical errors making the paper hard to read and understand in some places.

Response: We have corrected the errors in the revised manuscript.

Reviewer #3:

The authors have addressed this reviewers' comments. The revised manuscript is much improved.

Response: We thank the reviewer for the positive comments.

Reviewer #4:

Based on assessment of the original reports from the three referees, the authors have substantially improved their manuscript over the course of the revision. There is generally great interest in understanding the mechanisms linking GRN genetics to diseases other than GRN-associated FTL, including AD and PD, particularly as novel therapies are being developed and tested in the clinic. I am generally supportive of publication. I would just recommend that the authors should capture a bit more comprehensively the various molecular mechanisms linking PGRN deficiency to decreased GCase activity, in their discussion. As discussed in-depth in a recent TICB review article (PMID 36244875), those potentially include: (1) loss of direct chaperone activity; (2) reduced PSAP levels, processing or reduced SapC levels; (3) reduced maturation and lysosomal delivery of GCase; and (4) reduced levels of BMP, which can stimulate GCase activity via electrostatic interactions with protonated SapC and/or GCase itself. It seems that the authors settled on (3) as a likely mechanism, but the other three cannot be excluded.

Response: We appreciate the reviewer's positive assessment of our manuscript. In response to the reviewer's comment, we have added a paragraph for the molecular mechanism of PGRN's regulation of GCase activity in Discussion of the revised manuscript.

REVIEWERS' COMMENTS

Reviewer #2 (Remarks to the Author):

Thank you to the authors for responding to the concerns. All have been addressed adequately and i recommend publication.

Below is a point-by-point response to the reviewers' comments.

REVIEWERS' COMMENTS

Reviewer #2:

Thank you to the authors for responding to the concerns. All have been addressed adequately and i recommend publication.

Response: We thank the reviewer for the positive comments.